# *Pm57* from *Aegilops searsii* encodes a tandem kinase protein and confers wheat powdery mildew resistance

Yue Zhao [1,5], Zhenjie Dong[2,5], Jingnan Miao [1], Qianwen Liu [1], Chao Ma [1], Xiubin Tian[1], Jinqiu He [1], Huihui Bi[1], Wen Yao [1], Tao Li[1], Harsimardeep S. Gill [3], Zhibin Zhang[4], Aizhong Cao[2], Bao Liu [4], Huanhuan Li [1] ✉, Sunish K. Sehgal [3] ✉ & Wenxuan Liu [1] ✉

Powdery mildew is a devastating disease that affects wheat yield and quality. Wheat wild relatives represent valuable sources of disease resistance genes. Cloning and characterization of these genes will facilitate their incorporation into wheat breeding programs. Here, we report the cloning of *Pm57*, a wheat powdery mildew resistance gene from *Aegilops searsii*. It encodes a tandem kinase protein with putative kinase-pseudokinase domains followed by a von Willebrand factor A domain (WTK-vWA), being ortholog of Lr9 that mediates wheat leaf rust resistance. The resistance function of *Pm57* is validated via independent mutants, gene silencing, and transgenic assays. Stable *Pm57* transgenic wheat lines and introgression lines exhibit high levels of all-stage resistance to diverse isolates of the *Bgt* fungus, and no negative impacts on agronomic parameters are observed in our experimental set-up. Our findings highlight the emerging role of kinase fusion proteins in plant disease resistance and provide a valuable gene for wheat breeding.

Bread wheat (*Triticum aestivum* L.) is an important staple food crop that plays a vital role in global food security[1]. However, sustainable wheat production across the globe is threatened by its susceptibility to various pests and diseases. Among various diseases, powdery mildew caused by *Blumeria graminis* f. sp. *tritici* (*Bgt*) is a major wheat disease worldwide that affects grain yield and processing quality. This disease typically leads to yield losses higher than 1% globally and as high as 3.27% in China[2]. The development and deployment of resistant wheat varieties have been regarded as the most economical, effective, and sustainable way to mitigate the losses caused by powdery mildew.

To date, more than one hundred *Pm* resistance genes/alleles at approximately 60 loci in wheat and its wild relatives have been documented[3,4]; however, only a few *Pm* genes, including *Pm2*, *Pm4*, *Pm5*, *Pm6*, *Pm8*, *Pm21*, and *Pm52*, have been widely used in developing disease-resistant wheat varieties[5-8]. The majority of *Pm* genes cannot be utilized due to associated linkage drags causing adverse pleiotropism[9] or narrow-spectrum resistance that often becomes ineffective with the evolution of new *Bgt* races[10]. This necessitates the identification and cloning of new genes that confer broad-spectrum powdery mildew resistance with no adverse effects on wheat agronomic characteristics.

Owing to its large genome size, cloning a gene in wheat by map-based cloning was time-consuming and challenging[11]. However, the availability of recently released genome sequence resources and tools, such as the genome sequences of more than ten hexaploid wheat cultivars, has largely facilitated gene mapping and cloning in wheat.

[1]State Key Laboratory of Wheat and Maize Crop Science, College of Life Sciences, Henan Agricultural University, Zhengzhou 450002, China. [2]College of Agronomy, Nanjing Agricultural University, Nanjing 210000, China. [3]Department of Agronomy, Horticulture and Plant Science, South Dakota State University, Brookings, SD 57007, USA. [4]Key Laboratory of Molecular Epigenetics of the Ministry of Education (MOE), Northeast Normal University, Changchun 130024, China. [5]These authors contributed equally: Yue Zhao, Zhenjie Dong. ✉e-mail: lihuanhuanhappy@henau.edu.cn; sunish.sehgal@sdstate.edu; wxliu2003@hotmail.com

Since *Pm3* was isolated from wheat in 2004[12], 20 *Pm* resistance genes have been cloned. Thirteen of these 20 genes, i.e., *Pm3*[12], *Pm8*[13] (ortholog of *Pm3*), *Pm2*[14], *Pm60*[15], *Pm17*[16] (allele of *Pm8*), *Pm21*[17,18], *Pm41*[19], *Pm5e*[20], *Pm1a*[21], *MlIW172/MlWE18*[22] (allelic to *Pm60*), *Pm12*[23] (ortholog of *Pm21*), *Pm69*[24], and *PmS5*[25], encode nucleotide-binding leucine-rich repeat (NLR) immune receptors. Among the remaining seven *Pm* genes, *Pm24*, *WTK4*, and *Pm36* encode tandem kinases[26–28], *Pm4* and *Pm13* encode kinase fusion proteins[29,30], and the two broad-spectrum resistance genes *Pm38/Yr18/Lr34/Sr57* and *Pm46/Yr46/Lr67/Sr55* encode an ABC transporter and a hexose transporter, respectively[31,32].

Furthermore, wild relatives of hexaploid bread wheat serve as reservoirs of genetic diversity for important agronomic traits, including disease resistance genes[17,] and more than half of the currently designated *Pm* genes have been derived from secondary and tertiary gene pools of wheat[33,34]. Although *Pm* genes from wild relatives play an important role in breeding for disease resistance, it is very difficult to fine-map and clone alien genes from secondary or tertiary gene pools in the wheat background compared to those from common wheat due to homoeologous recombination suppression, lack of alien chromosome-specific markers and the unavailability of annotated reference genomes. To date, only six *Pm* genes have been cloned from wild relatives of wheat in secondary and tertiary gene pools including recently reported *Pm55*[25] (*Dasypyrum villosum* L.) and *Pm13*[30] (*Ae. longissima*); for the other four genes, *Pm12* (*Ae. speltoides*) and *Pm21* (*Dasypyrum villosum* L.) are orthologous, and *Pm8* and *Pm17* (*Secale cereale* L.) were cloned by *Pm3* homology-based cloning. The difficulty of gene cloning hinders a better understanding of the molecular basis of these genes and limits their deployment in wheat breeding.

The powdery mildew resistance gene *Pm57*[35] was derived from *Aegilops searsii* Feldman & Kislev ex Hammer (2n = 2x = 14, SˢSˢ), an S-genome species from the section *Sitopsis* (Jaub. & Spach) Zhuk in the secondary gene pool of wheat. Like disease resistance gene *Sr51*[36], *Pm57* has been transferred into bread wheat from *Ae. searsii*. In previous study, we screened a complete set of Chinese Spring (CS)-*Ae. searsii* chromosome addition lines and found that the chromosome 2Sˢ addition line (TA3581) confers resistance to powdery mildew[35]. We developed ten CS-*Ae. searsii* 2Sˢ translocation lines based on homoeologous recombination induced by the pairing homoeologous gene (*Ph1*) mutation (*ph1b*). Cytogenetic analyses and powdery mildew resistance assays of the translocation lines localized *Pm57* to the fragment length (FL) interval of FL0.75-0.87 on the long arm of 2Sˢ [35]. We further physically mapped *Pm57* to the long arm of 2Sˢ in a 5.13 Mb genomic region[37].

In this work, we report the map-based cloning of *Pm57*. *Pm57* encodes a tandem kinase protein with putative kinase-pseudokinase domains followed by a von Willebrand factor A (vWA) domain, being an ortholog of Lr9 (WTK6-vWA) that mediates resistance to wheat leaf rust[38]. Stable *Pm57* transgenic wheat lines and introgression lines exhibit high-level, all-stage powdery mildew resistance with no apparent adverse agronomic traits. Our results provide a valuable resistance gene for further elucidating the molecular mechanisms underlying wheat powdery mildew resistance and will enable the development of wheat varieties with broad-spectrum powdery mildew resistance.

## Results
### High-resolution mapping and map-based cloning of Pm57
Previously, we mapped *Pm57* to the long arm of 2Sˢ in a 5.13 Mb genomic region using an $F_2$ population generated by crossing recombinant line 89(6)88 containing *Pm57* with TA3809, a CS *ph1b* deletion mutant that can promote homoeologous chromosome pairing and recombination[37] (Fig. 1a). To fine-map *Pm57*, nine heterozygous recombinant plants harboring *Pm57* were self-pollinated to develop $F_3$ populations segregated for *Pm57*. A total of 3,380 $F_3$

individuals were subsequently screened using two *Pm57* flanking markers (*X67593* and *X62492*), resulting in the identification of 104 CS-*Ae. searsii* 2Sˢ recombinants (Fig. 1b). The plant responses to *Bgt* were subsequently assessed in the progeny of these 104 recombinants by inoculation with *Bgt* isolate E09 at the seedling stage in a growth chamber.

Next, 16 2Sˢ-specific markers were designed within the mapping interval of *Pm57* flanked by the markers *X67593* and *X62492* using the recently released genome sequence of *Ae. searsii* (TE01)[1] (Supplementary Data 1). Marker analysis of the 104 CS-*Ae. searsii* 2Sˢ recombinants revealed six different types (Types I to VI). By integrating the *Bgt* responses of the 104 recombinants with the 16-marker analysis, we mapped *Pm57* to the region between markers *X10* and *X13* (Fig. 1b). This region corresponds to a physical interval of 710 kb on the long arm of *Ae. searsii* chromosome 2Sˢ, which harbors 12 genes (referred to as *G1-G12*) based on the genome sequence of *Ae. searsii*[1]. It is worth noting that *G4* and *G5* encode proteins containing putative tandem kinase domains and represent wheat tandem kinase (WTK) genes (Fig. 1c)[39,40]. In addition, the *G4* and *G5* genes were absent in CS, while the other ten of the 12 genes were conserved and shared more than 80% amino acid sequence identity with the corresponding genes on Group 2 chromosomes of common wheat CS (Supplementary Table 1).

### Identification of Pm57 candidate genes by MutRNA-Seq
To identify *Pm57* candidates, we performed ethyl methane sulfonate (EMS) mutagenesis on the CS-*Ae. searsii* recombinant line 89(5)69. Line 89(5)69 contains a pair of recombined chromosomes (T2BS·2BL-2SˢL) in which the long-arm terminal segment carries *Pm57* of the *Ae. searsii* chromosome 2Sˢ that have been translocated to the long arms of CS 2B. Approximately 10,000 $M_0$ seeds were treated with 0.6% EMS, and $M_1$ seeds were harvested from approximately 1,598 surviving $M_0$ plants. At least ten plants from each of the randomly selected 300 $M_1$ families were screened for susceptible mutants using *Bgt* isolate E09. To eliminate the susceptibility caused by seed contamination or missing 2Sˢ, the screened susceptible mutants were verified using the 2Sˢ-specific molecular markers *X67593* and *X62492* flanking *Pm57*. Finally, 15 independent susceptible mutants from different $M_1$ families were identified and further validated by *Bgt* isolate E09 assays in the $M_2$ generation (Supplementary Fig. 1).

Furthermore, we performed mutant RNA-seq (MutRNA-seq) using five susceptible mutants (Mut51, Mut60, Mut141, Mut209 and Mut216) and the resistant wild-type parental line 89(5)69 (Fig. 2a). Alignment of the MutRNA-seq data to the 12 genes in the *Pm57* mapping interval revealed that the *G4* gene had EMS-type (G/C to A/T) mutations in all five mutants, whereas only one EMS-type mutation was found in genes *G2* (in Mut209) and *G8* (in Mut141), and none were found in *G5*, *G6*, *G7*, *G9*, *G10*, *G11* or *G12*. The expression levels of the remaining two genes (*G1* and *G3*) were too low to reliably call mutations in the transcriptome sequences (Fig. 2a and Supplementary Table 2). Therefore, the *G4* gene, which encodes a tandem kinase-vWA domain protein, emerged as the most likely candidate for *Pm57* among the 12 genes.

### Validation of G4 by EMS mutants, gene silencing, and transgenic assays
To validate the candidacy of *G4* for *Pm57*, we sequenced the genomic DNA and cDNA sequences of the *G4* gene and another *WTK* gene (*G5*) in the resistant 2Sˢ-2A recombinant line 89(5)69 as well as the 15 susceptible mutants for sequence comparison. In the resistant line 89(5)69, the *G4* gene was 9473 bp in length and contained 14 exons with a coding sequence of 3489 bp (Fig. 2b); moreover, the sequence of *G4* in 89(6)69 was identical to that in the reference genome TE01. Intriguingly, *G4* had at least seven alternative splicing variants, designated IF1-IF7; IF1 was the main isoform, representing 50.6% (39 out of 77 tested *G4* cDNA clones) at 0 h post-inoculation (hpi) with *Bgt* isolate E09 and

increasing to 80.9% at 24 hpi; and isoforms IF2-IF7 were much less abundant (1.3–27.3%). IF1 encodes a full-length intact G4 protein with Kin I, Kin II and vWA domains, while IF2 and IF3 encode proteins with truncated Kin I and Kin II domains, and IF4-IF7 encode proteins with only a truncated Kin I domain (Supplementary Fig. 2). Gene sequence comparison revealed that 11 of the 15 susceptible mutants had SNPs in *G4* that resulted in amino acid substitutions, premature stop codons, or relocation of the intron/exon splice sites (Fig. 2b). Specifically, a frameshift mutation was found in Mut216, with a G/A point mutation in the splice acceptor site of intron 9. Mut351 was the same as Mut60 in *G4*, both of which had a nonsense mutation that gave rise to a premature stop codon at amino acid position 1,081. The other eight mutants (G78D in Mut223, G177E in Mut141, G193R in Mut51, D209N in Mut210, G424D in Mut92, P747L in Mut121, R829W in Mut22, and G903D in Mut209) harbored missense mutations that occurred in the kinase I (Kin I), kinase II (Kin II) or vWA domains (Fig. 2b). In addition, no sequence variations in the *G5* gene were found among the susceptible mutants.

Next, virus-induced gene silencing (VIGS) was performed to validate candidate genes *G4* and *G5*. The resistant introgression line 89(5) 69 inoculated with G4-VIGS constructs lost resistance to powdery mildew, whereas the plants inoculated with G5-VIGS constructs and empty vector control constructs remained resistant (Supplementary Fig. 3). These results were consistent with the conclusions drawn from the MutRNA-Seq analyses and suggested that *G4* is the prime candidate for *Pm57*.

Transgenic complementation of the susceptible cv. Fielder was subsequently performed to confirm the role of *G4* in conferring resistance against powdery mildew. We inserted the CDS (isoform IF1) of *G4* into the expression vector pWMB110 driven by the maize *ubiquitin* (*Ubi*) promoter and transformed it into cv. Fielder by *Agrobacterium*-mediated transformation. A total of 36 $T_0$ plants (L1 to L36) were generated, and 33 were identified as positive transgenic plants (Supplementary Fig. 4). With the exceptions of L20 and L29, all the positive $T_0$ transgenic plants (+) were highly resistant to the *Bgt* isolate E09, whereas all the negative plants (−, L5, L23 and L27) were as susceptible as the WT Fielder (Supplementary Fig. 4). This difference in powdery mildew disease resistance was also observed for the $T_1$ transgenic lines and the WT (Supplementary Table 3); 16 representative lines are shown in Fig. 3a. In addition, qRT–PCR analyses revealed no expression of the *G4* gene in the positive transgenic lines L20 and L29, explaining why these two positive lines were susceptible to powdery mildew (Fig. 3b). Taken together, the results of genetic mapping, mutant analysis, gene silencing, and transgenic assays confirmed that *G4* is *Pm57*.

## Subcellular localization and structural analysis of the Pm57 protein

The subcellular localization of Pm57 was investigated through transient transformation of wheat leaf protoplasts. We constructed a fusion protein of Pm57 with green fluorescent protein (GFP), and the resulting Pm57-GFP construct and the GFP control were introduced

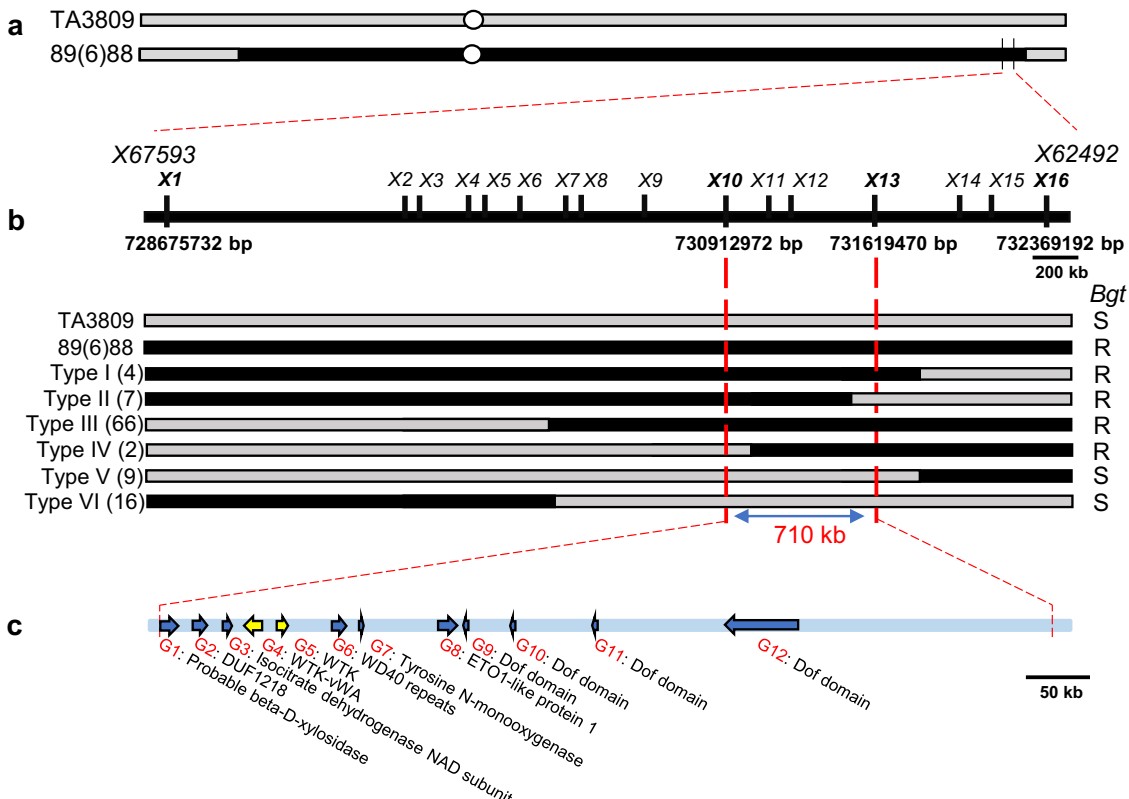

**Fig. 1 | High-resolution physical mapping of Pm57 on chromosome 2S[s]. a** Initial physical-mapping by comparing the chromosome recombination breakpoints of the resistant and susceptible recombinants positioned *Pm57* within an interval flanked by the markers *X67593* and *X62492* on the long arm of chromosome 2S[s]. TA3809 is a CS *ph1b* deletion mutant, and 89(6)88 is a CS-*Ae. searsii* recombinant line that contains a pair of recombined chromosomes, Ti2AS-2S[s]S.2S[s]L-2AL, that carry *Pm57*. **b** A high-resolution physical map delimited *Pm57* to a 710 kb region between markers *X10* and *X13*. Markers *X67593* and *X62492* were derived from RNA-seq of the Chinese Spring (CS)-*Ae. searsii* 2S[s] disomic addition line TA3581, and markers *X1-X16* were developed using the recently released genome sequence of *Ae. searsii* (TE01)[1]. Marker analysis of the 2S[s] recombinants grouped them into six different types (Types I to VI), and the number of recombinants of each type is shown in brackets. The responses of each type to *Bgt* isolate E09 are displayed as R (resistant) or S (susceptible). **c** Twelve annotated genes (*G1-G12*) present in the mapping interval of the *Ae. searsii* reference genome (TE01). Two of these 12 genes (*G4* and *G5*) encode proteins containing putative tandem kinase domains and represent wheat tandem kinase (WTK) genes. vWA: von Willebrand factor A domain.

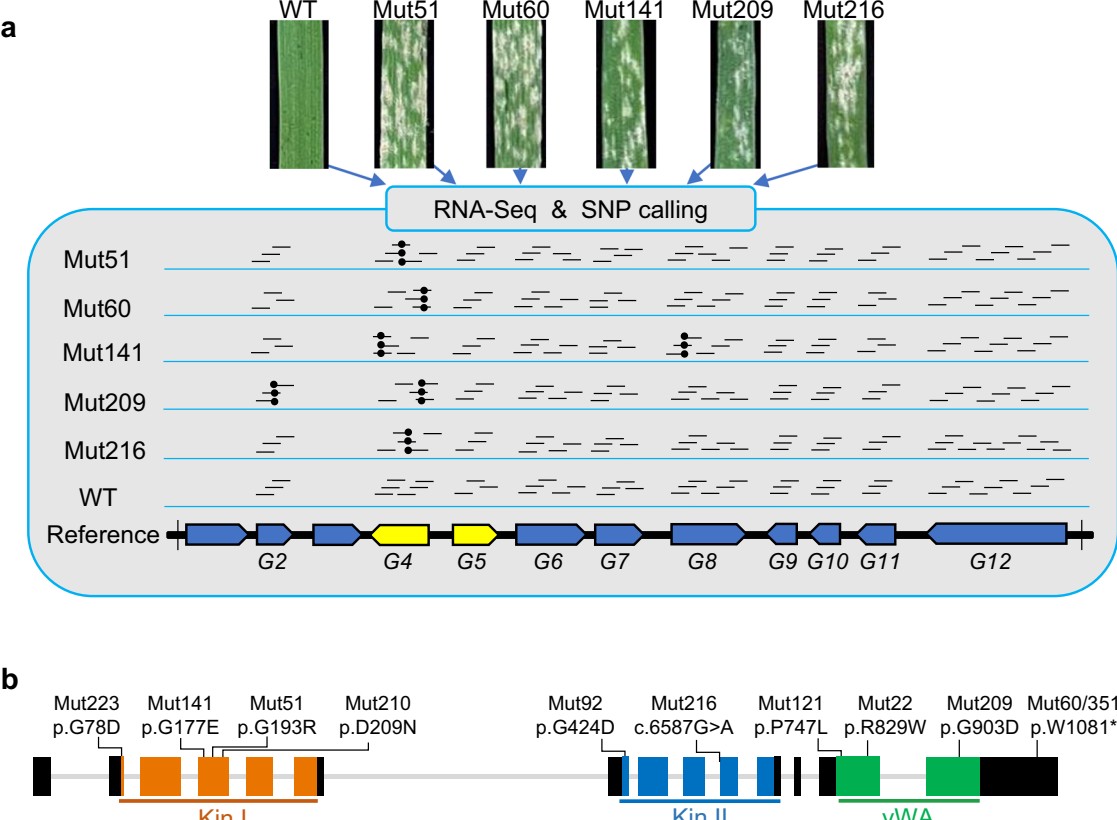

**Fig. 2 | Pm57 candidate gene identification using EMS-induced mutants.**
**a** Candidate gene identification by MutRNA-Seq. RNA-Seq reads from the parental line 89(5)69 and 5 susceptible EMS mutants (Mut51, Mut60, Mut141, Mut209 and Mut216) were mapped to the *Ae. searsii* reference genome sequence (TE01). Only *G4* of the 12 candidate genes in the mapping interval carried EMS-type (G/C to A/T) mutations (black dots) in all 5 sequenced mutants. **b** EMS-derived loss-of-function mutants with mutations in the *G4* gene sequence. Gene structure of *G4* (from the start to the stop codon) was presented. Rectangles indicate coding exons, and gray straight lines indicate introns. The positions of the three conserved domains and the predicted amino acid changes caused by the EMS mutations are indicated. The two protein kinase domains and the vWA domain are shown in orange, blue, and green, respectively. "c." for a coding DNA sequence and "p." for a protein sequence. A frameshift mutation is detected in Mut216 with a G/A point mutation in the splice acceptor site of intron 9. The other ten mutants harbored missense or nonsense mutations.

---

into wheat protoplasts. As shown in Fig. 4a, green fluorescence of the GFP control was observed in the nucleus and cytoplasm. Similarly, the Pm57-GFP fusion protein was fluorescent in the nucleus and cytoplasm in wheat (Fig. 4a).

To study the structural characteristics of the Pm57 protein sequence, we compared it with the structures of eight reported WTKs (Rpg, Un8, Yr15, Sr60, Sr62, Pm24, WTK4, and Lr9)[41,42]. Based on the sequence conservation of the key amino acid residues in the two kinase domains[43], Pm57 was classified as a tandem kinase-pseudokinase protein, similar to Yr15, Pm24, Lr9, and Sr62 (Supplementary Fig. 5). The amino acid residues in the VIb (catalytic loop) of Pm57 were LD (leucine–aspartic acid) (Supplementary Fig. 5), suggesting that Pm57 is a non-RD (non-arginine–aspartate) kinase. To explore the structure of Pm57 in detail, AlphaFold[44] was used to generate a 3D model of Pm57. The tertiary structure of Pm57 is modular, with kinase-pseudokinase domains, a vWA domain, and a putative Vwaint domain, as described by ref. 38. (Fig. 4b). We also observed that the kinase domains of Pm57 were highly symmetrical, similar to those of WTK4 and Lr9 but not similar to those of the other five reported TPKs (Fig. 4b and Supplementary Fig. 6). Notably, the amino acid sequence of Pm57 was highly similar (88.3%) to that of Lr9 (Supplementary Fig. 5).

### Histopathological characterization of Pm57
To evaluate the potential mechanism involved in *Pm57*-mediated resistance, *Bgt* development was evaluated in susceptible CS and resistant 89(5)69 plants. In 89(5)69, the *Bgt* spores germinated and developed normally from 6 to 12 hpi without histochemical differences in comparison with those of CS (Fig. 5a). In CS, *Bgt* gradually invaded epidermal host cells, with haustoria (Hau) visible at 24 hpi and the formation of secondary hyphae (Hyp) observed at 36 hpi. In contrast, no haustoria or hyphae were detected at 24 and 36 hpi, respectively, in 89(5)69 (Fig. 5a). *Bgt*-infected leaf staining by diaminobenzidine (DAB) and trypan blue (TPN) further revealed robust accumulation of $H_2O_2$ and cell death in the *Bgt*-interacting host cells of resistant 89(5)69 (Fig. 5). Beginning at 24 hpi, a much greater proportion of plant cells was detected to accumulate intracellular ROS (intraROS) in 89(5)69 (more than 40% of the *Bgt*-infected cells) than in CS (-5%) (Fig. 5a, b). Quantitative assessment of cell death at 48 hpi revealed that 37.7% of the *Bgt*-infected cells in 89(5)69 died, while only 3.3% of cells died in CS (Fig. 5c). These results suggest that *Pm57* inhibits haustorium formation in *Bgt* plants and that this process might be correlated with $H_2O_2$ accumulation and induced cell death.

qRT–PCR analyses demonstrated that *Pm57* was more highly expressed in wheat leaf tissues than in roots and stems (Supplementary Fig. 7a). Upon pathogen infection with the *Bgt* isolate E09, the expression of *Pm57* increased 4-fold and peaked at 12 hpi in the leaves of 89(5)69 seedlings, after which the expression slightly decreased at 48 hpi (Supplementary Fig. 7b). In addition, the upregulation of pathogenesis-related (PR) genes (i.e., *PR1*, *PR2*, *PR3*, *PR4*, and *PR9*) upon inoculation with the *Bgt* isolate E09 was significantly greater in

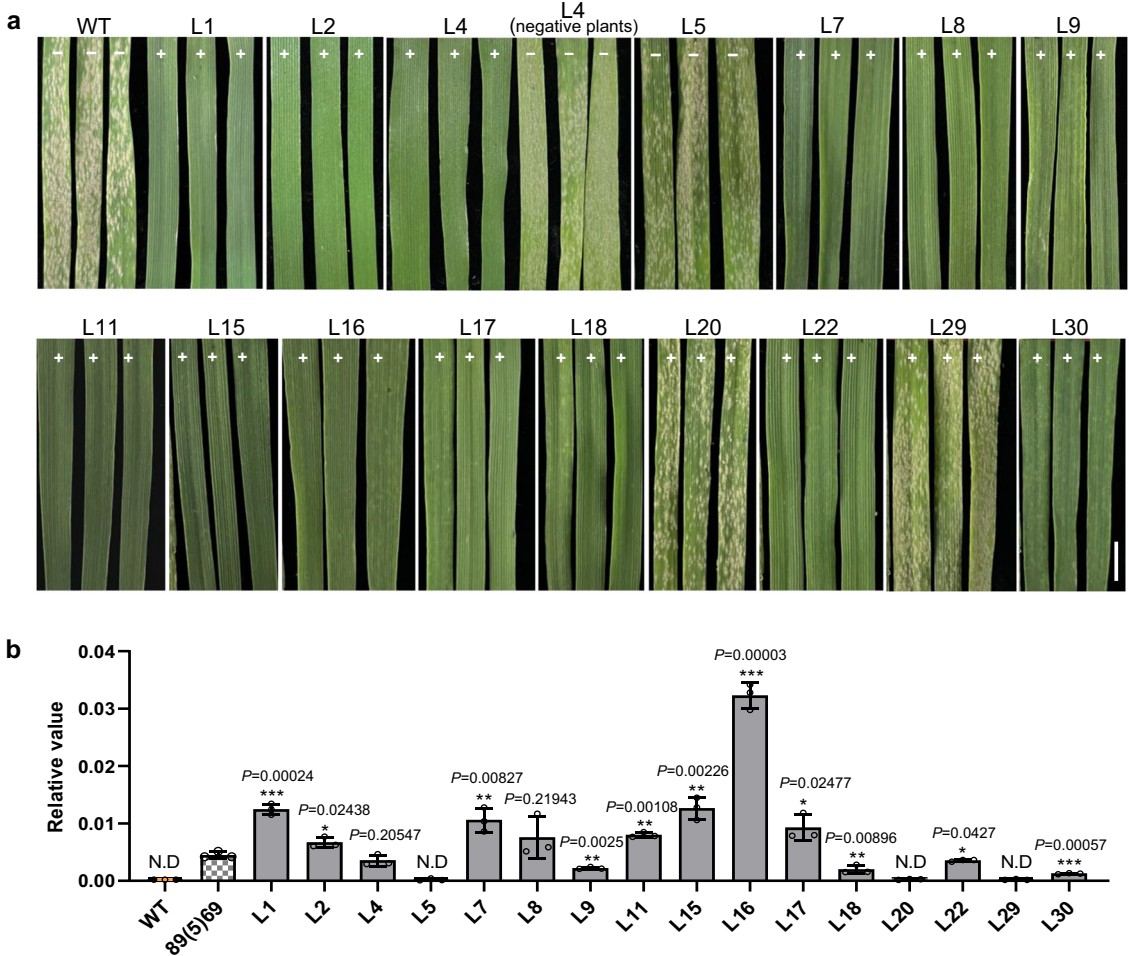

**Fig. 3 | Functional validation of *G4* by transgenic T1 lines. a** Powdery mildew resistance assay of 16 T$_1$ transgenic lines. Positive T$_1$ transgenic lines (+, except L20 and L29) of *G4* showed high resistance to the *Bgt* isolate E09. Fielder and the negative line L5 were used as susceptible controls. The L4 negative plants are the T$_1$ individuals without *G4* gene that segregated from the T$_0$ transgenic plant L4. Three individuals of each independent line are shown. The '+' and '−' signs on each leaf indicate the presence or absence of the *G4* gene, respectively. Scale bars = 0.5 cm. **b** Transcript levels of *G4* in the leaves of 16 T$_1$ transgenic lines and the *Pm57* introgression line 89(5)69. The T$_1$ line L5 was *Pm57* negative, and the remaining 15 T$_1$ lines were all positive. The first leaves of two-leaf seedlings were sampled for RNA extraction. Three biological replicates with leaves from four individual plants mixed as one replicate were used for expression analysis. Transcript levels were examined by qRT–PCR with *TaActin* as an endogenous control and calculated using the comparative CT method. The data are presented as the means ± SDs from three biological replicates. WT, nontransgenic control (Fielder); N.D., not detected. \**p* < 0.05, \*\**p* < 0.01, \*\*\**p* < 0.001 (two-tailed Student's *t* test). Asterisks indicate significant differences in each transgenic line compared with the 89(5)69 control. There was no expression of the *G4* gene in the positive transgenic lines L20 and L29, explaining why these two positive lines were susceptible to powdery mildew. Source data are provided as a Source Data file.

the leaves of the transgenic lines than in those of the susceptible control Fielder, and the expression levels of the *PR* genes peaked at timepoints later than those of *Pm57* (Supplementary Figs. 7b and 8). These expression patterns of the *Pm57* and *PR* genes imply that powdery mildew resistance of *Pm57* may involve *PR* genes up-regulated expression.

## Evolutionary analysis of Pm57

A comparative analysis of published wheat genome sequences showed synteny is conserved largely in tribe Triticeae for *Pm57* mapping genomic regions (Supplementary Fig. 9); however, *Pm57* orthologs were present only in *Aegilops bicornis* (TB01) (S$^b$S$^b$), *Ae. longissima* (TL05) (S$^l$S$^l$), *Ae. speltoides* (TS01) (SS), and the B subgenome of *T. dicoccoides* (WEWSeq v1) (AABB), and the proteins encoded by these orthologous genes had 81.8–87.8% sequence similarity with *Pm57* (Supplementary Figs. 9 and 10). Notably, *Pm57* is absent in the wheat reference genome sequence of cv. Chinese Spring, but sequences highly similar to that of *Pm57* were present in the syntenic region of several bread wheat cultivars (Supplementary Data 2). These highly

similar sequences were manually annotated and shared more than 80% amino acid sequence identity with Pm57. Synteny analysis was also performed for the flanking genomic regions around the *Pm57* locus in *Ae. searsii* (TE01) and the *Lr9* donor *Ae. umbellulata* (TA1851). The adjacent genomic region of *Pm57* was syntenic to the *Lr9* genomic region in *Ae. umbellulata* (Supplementary Fig. 11 and Supplementary Data 3), suggesting that *Pm57* may be an ortholog of *Lr9*. *Lr9*-vs.-*Pm57* reciprocal BLAST was performed between the *Ae. searsii* (TE01) and *Ae. umbellulata* (TA1851) assemblies, and the best BLAST hit with the *Lr9* genomic sequence was *Pm57*, while the best BLAST hit with the *Pm57* genomic sequence was *Lr9* (Supplementary Table 4). Therefore, genome similarity, gene collinearity and "reciprocal best hits" analyses revealed the *Pm57* gene to be an ortholog of *Lr9*; thus, this gene was designated WTK6b-vWA.

To gain insight into the evolution of *Pm57* and *Lr9*, we calculated the ratio (*Ka/Ks*) of the nonsynonymous substitution rate (*Ka*) to the synonymous substitution rate (*Ks*) for both *Pm57* and *Lr9*. The *Ka/Ks* ratio revealed that although *WTK6* (*Pm57* vs. *Lr9*) primarily underwent strong purifying selection (*Ka/Ks* = 0.42 < 1), the three domains

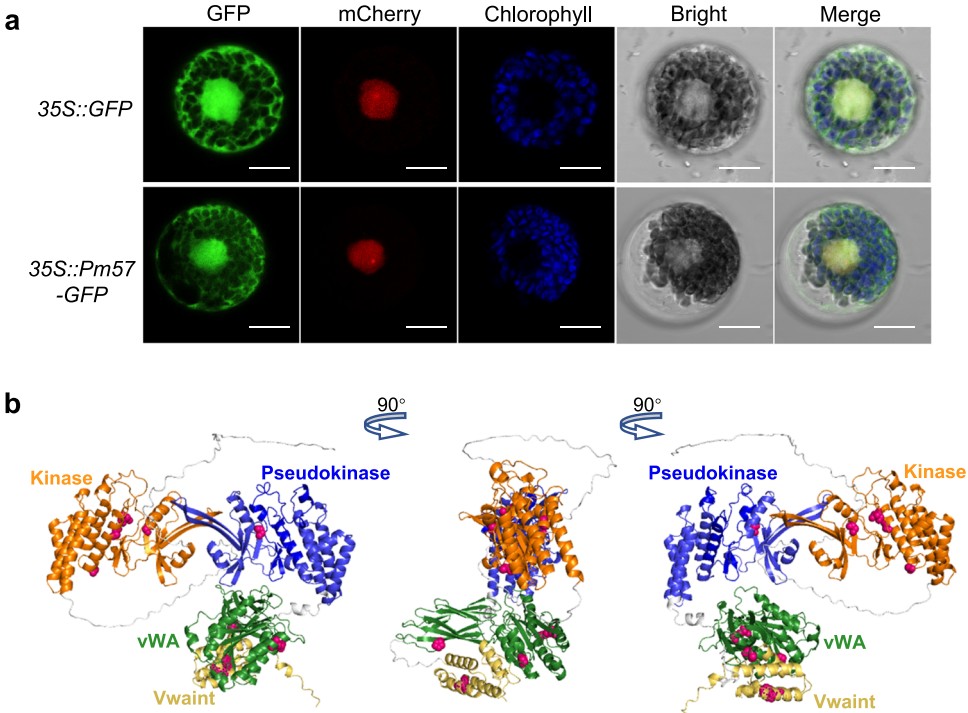

**Fig. 4 | Subcellular localization and protein structure prediction of Pm57.**
**a** Subcellular localization of Pm57 in wheat protoplasts. *35 S::GFP* or *35 S::Pm57-GFP* plasmids were cotransformed into wheat protoplasts with the nuclear marker plasmid *AtPIF4-mCherry*. The GFP, mCherry and chlorophyll fluorescence was visualized under a confocal laser-scanning microscope. *35 S::GFP* served as a control. Scale bars, 10 μm. Three independent experiments were performed. **b** Protein structure prediction of Pm57. A three-dimensional model of Pm57 was predicted by AlphaFold. Orange, kinase domain; blue, pseudokinase domain; green, vWA domain; yellow, putative Vwaint domain. The red spheres indicate amino acid substitutions resulting from the EMS mutagenesis.

presented diverse selective pressures (Supplementary Table 5). Strikingly, pseudokinase (K2) exhibited relaxed constraints or neutral evolution ($Ka/Ks = 0.77$), whereas the kinase (K1) and vWA domains exhibited strong purifying selection ($Ka/Ks = 0.14$ and $0.19$).

To evaluate the evolutionary origin of the protein kinase domains in Pm57, each of the two kinase domains in Pm57 was queried against the Hidden Markov models (HMMs) of 157 protein kinase subfamilies developed by Legti-Shiu and Shui[45]. Both protein kinase domains in Pm57 were assigned to the protein kinase subfamily DLSV (DUF26, SD-1, LRR-VIII and VWA) in the receptor-like kinase (RLK)/Pelle family. Therefore, Pm57 has a DLSV-DLSV configuration similar to that of Rpg1, Pm24, and Sr62[46]. Notably, a portion of the DLSV members were subsequently recognized as the LRR_8B subfamily (cysteine-rich receptor-like kinases)[47]. To further classify the two protein kinase domains of Pm57, we used the 182 protein kinase domains identified by Klymiuk et al.[43]. to construct a neighbor-joining (NJ) phylogenetic tree. As shown in Supplementary Fig. 12, both of the protein kinase domains in Pm57 were assigned to the LRR_8B subfamily.

To test whether the *Pm57* resistance allele is present in other wheat germplasms, an *STS-Pm57* marker with a 57 °C annealing temperature was used to amplify a 530 bp genomic sequence of the *Pm57* gene from 71 wheat accessions (including *T. urartu*, *T. boeoticum*, *Ae. tauschii*, *T. monococcum*, *T. dicoccoides*, *T. durum*, *T. aestivum* ssp. *yunnanese*, *T. aestivum* ssp. *macha*, *T. aestivum* ssp. *spelta*, *T. aestivum* ssp. *tibetanum* accessions and Chinese common wheat landraces and modern cultivars; Supplementary Data 4). The amplified fragment of *Ae. searsii* (TE01) was identical to that of CS-*Ae. searsii* chromosome 2Sˢ introgression line 89(5)69, but no identical fragment was obtained for the remaining 69 accessions (Supplementary Fig. 13). These results indicate that the *STS-Pm57* marker can be used as a diagnostic marker for the effective detection of *Pm57* in wheat breeding programs.

## Evaluation of Pm57 application in wheat breeding

Our previous studies showed that *Pm57* confers high resistance to mixed *Bgt* isolates collected in Henan Province[37]. To further study the spectrum of resistance offered by *Pm57* against genetically divergent *Bgt* isolates, we collected 29 *Bgt* isolates from major wheat-growing regions of China and used them for *Bgt* resistance assays of both the CS-*Ae. searsii* introgression line 89(5)69 and the transgenic line L1. As a result, plants of both 89(5)69 and the transgenic line L1 exhibited high resistance to all 29 divergent *Bgt* isolates, with an infection type ranging from 0–1 (Supplementary Table 6). Furthermore, we found that 89(5)69 and positive transgenic plants exhibited high resistance from the seedling stage to the adult stage (Supplementary Fig. 14). In addition, all of the $F_1$ plants derived from the CS-*Ae. searsii* recombinant (88R-3-19-1), which has a small 2Sˢ segment harboring *Pm57*, crossed with 22 wheat varieties were highly resistant to powdery mildew, indicating that *Pm57* could play a role in resistance in diverse genetic backgrounds (Supplementary Fig. 15).

To determine the value of *Pm57* in wheat breeding, we evaluated the major agronomic traits of the *Pm57* transgenic wheat lines and introgression lines. No significant differences were observed for heading date, plant height, tiller number, thousand-grain weight or spike morphology between the three transgenic lines and the control lines, which included both the negative lines and the WT (Fig. 6 and Table 1). Additionally, there was no significant difference between the introgression lines (Type IV-1 and Type IV-2; in Fig. 1b, 5.57–9.12 Mb) and the parental line TA3809 for these traits (Fig. 6 and Table 1). The results showed that *Pm57* had no obvious adverse effects on important agronomic traits.

## Discussion

The development of wheat-alien recombinants is not only a tool for cloning alien genes but also one of the best approaches for transferring

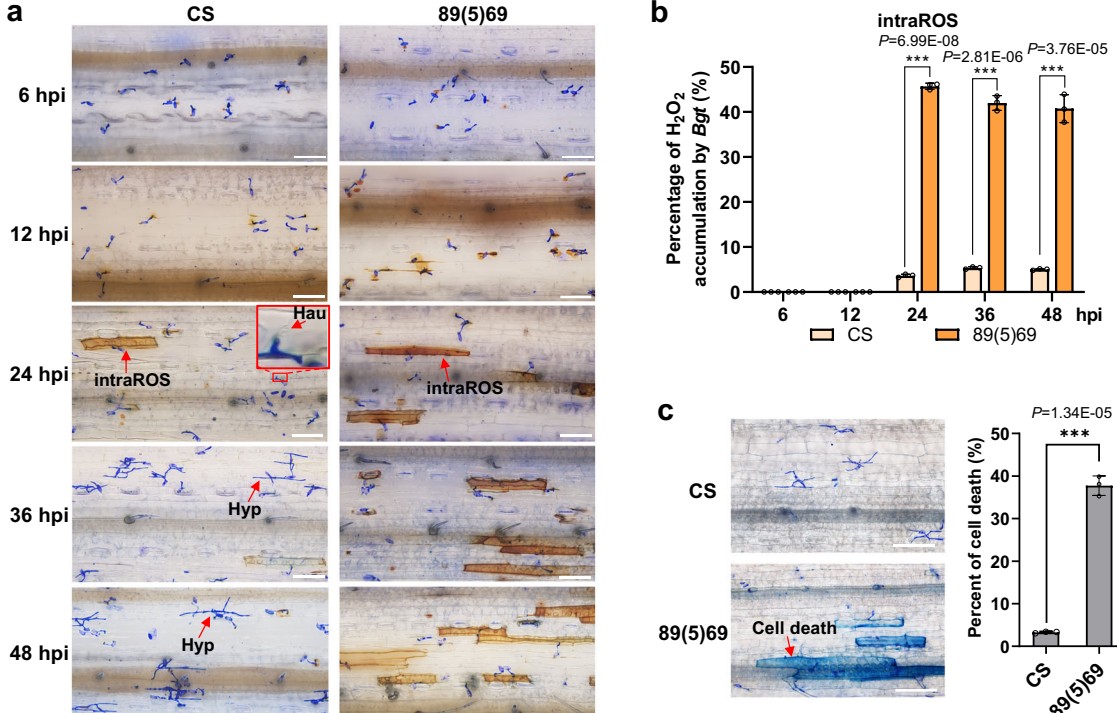

**Fig. 5 | Histopathological characterization of *Pm57*. a, b** Detection of $H_2O_2$ accumulation via DAB staining of *Bgt*-infected leaves. Seven-day-old plants were inoculated with the *Bgt* isolate E09. Staining was performed on the *Bgt*-infected leaves at 6, 12, 24, 36, and 48 h post inoculation (hpi). The *Bgt* spores germinated and developed normally from 6 to 12 hpi, and no histochemical difference was observed between 89(5)69 and CS. In CS, *Bgt* gradually invaded epidermal host cells, with haustoria (Hau) visible at 24 hpi and the formation of secondary hyphae (Hyp) observed at 36 hpi. In contrast, no haustoria or hyphae were detected at 24 and 36 hpi in 89(5)69, respectively. Brown staining shows the accumulation of $H_2O_2$. intraROS: intracellular ROS; scale bars, 100 µm. **c** Trypan blue staining of *Bgt*-infected leaves at 48 hpi to visualize plant cell death. Scale bars, 100 µm. *Bgt*-infected leaf segments were stained with Coomassie Brilliant Blue for the observation of fungal structures. The data are presented as the means ± SDs from three biological replicates. ***$p < 0.001$ (two-tailed Student's *t* test). Source data are provided as a Source Data file.

favorable genes from wild relatives to increase the genetic diversity of cultivated wheat. However, homoeologous recombination between wheat and alien chromosomes is suppressed by the pairing of homoeologous (*Ph*) genes, hampering the development of recombinants with smaller alien segments, which is essential for mapping and deploying exotic wheat genes in bread wheat cultivars[48]. In this study, we used CS-*Ae. searsii* 2S$^s$ F$_3$ recombinants developed from a cross between the *Ph1* locus deletion mutant TA3809 and the powdery mildew-resistant translocation line 89(6)88 to fine-map the broad-spectrum powdery mildew resistance gene *Pm57*. The *Pm57* candidate region harbored 12 genes in the *Ae. searsii* reference genome assembly (TE01)[1]. We further performed MutRNA-Seq to narrow the *Pm57* candidate genes in the target mapping interval and identified *G4* as the most likely candidate for further transgenic validation. Unexpectedly, the CDS of *G4* was incompletely annotated as EVM0016946 in the *Ae. searsii* reference genome; this gene encodes a protein with a single kinase domain followed by a vWA domain. We obtained the full-length sequence of *G4* from *Pm57* direct donor RNA-seq data[37] and aligned the MutRNA-Seq data from the susceptible mutants against the *G4*-corresponding unigenes. In recent years, MutRNA-Seq has been proven to be an effective method for cloning disease resistance genes; for example, *Sr62*[46] and *Lr9/Lr58*[38] were cloned by exploiting similar MutRNA-Seq approaches.

Recently, tandem kinase proteins (TKPs) have emerged as prominent new components of disease resistance in Triticeae[39]. To date, nine TKPs, namely, *Rpg1*[49], *Yr15* (*WTK1*)[43], *Sr60* (*WTK2*)[50], *Pm24* (*WTK3*)[27], *WTK4*[26], *Sr62* (*WTK5*)[46], *Lr9* (*WTK6-vWA*)[38], *Rwt4*[51], and *Pm36* (*WTK7-TM*)[28], have been identified to confer resistance against various fungal pathogens. *Pm57* (*WTK6b-vWA*) from *Ae. searsii* is an additional member of the TKP family that confers resistance to wheat powdery mildew. It contains tandem kinase domains and a Willebrand factor A (vWA) domain and is the second identified protein with this type of unusual WTK-vWA structure following Lr9, which confers resistance to wheat leaf rust. The vWA/Vwaint domains are present in bacteria, archaea and eukaryotes and are considered to participate in protein–protein interactions[52,53]. For example, the vWA domain of the human copine protein can interact with a wide variety of signaling molecules containing coiled-coil domains[54]. In plants, vWA-containing copine proteins were shown to be regulators of basal and *R*-mediated disease resistance, suggesting that vWA-containing proteins may play an important role in plant disease resistance[55]. In this study, eight susceptible mutants harbored missense mutations in the Kin I, Kin II or vWA domains (Fig. 2), indicating that each domain of Pm57 is essential for resistance to the *Bgt* pathogen. A similar phenomenon was also observed for the *Rpg1*, *Pm24*, *Sr62* and *Lr9* genes[27,38,46,49]. Neither sequence variation nor changes in the expression level of *Pm57* were found in four of the 15 susceptible mutants (Supplementary Fig. 16), providing a resource to further study other genes or elements required for *Pm57* function.

Comparative analysis of Pm57 orthologs among plant kingdoms revealed that its orthologs are present only in *Ae. bicornis* (TB01.2S01G0903000.1), *Ae. longissima* (TL05.2S01G0920200.1), *Ae. speltoides* (TS01.2B01G0896900.1), *T. dicoccoides* (TRIDC2BG085800.1) and several sequenced bread wheat accessions (Supplementary Fig. 9 and Supplementary Data 2), suggesting a likely recent origin of *Pm57* after the divergence of Triticeae species and the reticulate evolutionary nature of wheat[56,57]. However, several genes encoding proteins with a single kinase domain followed by a vWA domain were identified not only

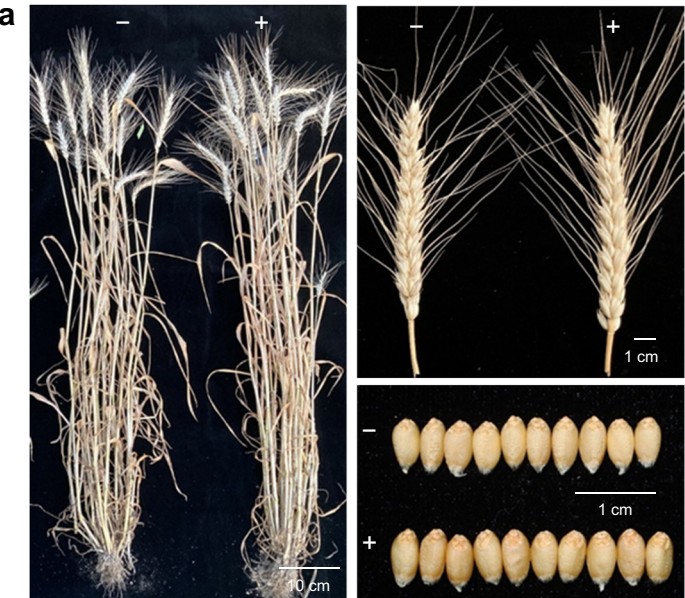

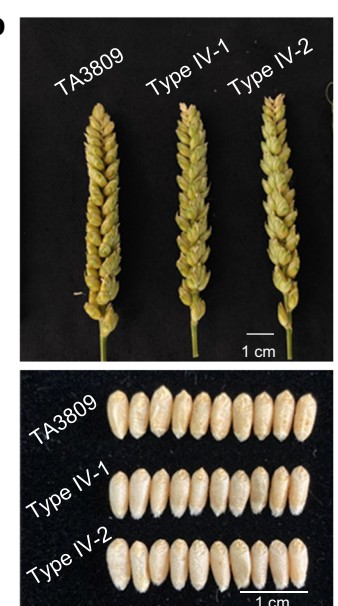

**Fig. 6 | Visual phenotypes of *Pm57* transgenic lines, introgression lines and control plants under field conditions. a** Whole-plant, spike, and seed growth habits of Fielder (−) and *Pm57* transgenic plants (+). **b** The spikes and seeds of TA3809 and the CS-*Ae. searsii Pm57* introgression lines. Types IV-1 and IV-2 are two CS-*Ae. searsii* introgression lines with small 2Sˢ segments carrying *Pm57* (Type IV in Fig. 1b; 2Sˢ fragment sizes between 5.57 Mb and 9.12 Mb, ~1% of the 2Sˢ chromosome). TA3809 served as a control for these introgression lines.

**Table 1 | Comparison of the agronomic traits of the *Pm57* transgenic lines and the CS-*Ae. searsii Pm57* introgression lines with those of the controls**

|  | Heading date (d) | Plant height (cm) | Spike length (cm) | Spikelet number per spike | Seeds per spike | Thousand grain weight (g) |
|---|---|---|---|---|---|---|
| WT (Fielder) | $72.7 \pm 0.9$ | $76.3 \pm 4.0$ | $11.0 \pm 1.0$ | $19.1 \pm 1.0$ | $60.9 \pm 3.0$ | $42.6 \pm 0.6$ |
| L1 | $73.3 \pm 1.2^{ns}$ | $75.3 \pm 2.7^{ns}$ | $11.3 \pm 0.9^{ns}$ | $18.5 \pm 1.1^{ns}$ | $60.3 \pm 2.8^{ns}$ | $41.9 \pm 0.9^{ns}$ |
| L2 | $72.3 \pm 1.2^{ns}$ | $74.7 \pm 4.4^{ns}$ | $11.2 \pm 0.9^{ns}$ | $18.5 \pm 1.3^{ns}$ | $60.9 \pm 1.9^{ns}$ | $42.2 \pm 0.7^{ns}$ |
| L4 | $72.7 \pm 1.2^{ns}$ | $75.2 \pm 2.5^{ns}$ | $10.9 \pm 0.7^{ns}$ | $18.5 \pm 0.9^{ns}$ | $60.1 \pm 1.5^{ns}$ | $42.0 \pm 0.4^{ns}$ |
| L4(negative lines) | $72.0 \pm 0.8^{ns}$ | $75.5 \pm 4.5^{ns}$ | $10.8 \pm 0.9^{ns}$ | $19.1 \pm 1.0^{ns}$ | $60.9 \pm 2.8^{ns}$ | $42.2 \pm 0.4^{ns}$ |
| L5(negative lines) | $72.3 \pm 1.2^{ns}$ | $74.7 \pm 4.3^{ns}$ | $10.8 \pm 0.5^{ns}$ | $18.9 \pm 0.7^{ns}$ | $60.2 \pm 2.1^{ns}$ | $42.3 \pm 0.5^{ns}$ |
| TA3809 | $74.2 \pm 0.5$ | $131.2 \pm 4.3$ | $8.8 \pm 1.0$ | $23.2 \pm 1.4$ | $61.9 \pm 1.5$ | $35.4 \pm 0.8$ |
| Type IV-1 | $75.0 \pm 0.8^{ns}$ | $131.6 \pm 6.0^{ns}$ | $9.2 \pm 1.1^{ns}$ | $22.8 \pm 2.0^{ns}$ | $61.9 \pm 1.3^{ns}$ | $34.9 \pm 0.3^{ns}$ |
| Type IV-2 | $75.3 \pm 0.5^{ns}$ | $131.2 \pm 6.3^{ns}$ | $9.1 \pm 1.0^{ns}$ | $22.7 \pm 2.1^{ns}$ | $61.2 \pm 0.9^{ns}$ | $35.2 \pm 0.9^{ns}$ |

L1, L2, L4 and L5 are *Pm57* transgenic lines, and WT is the receptor control. Types IV-1 and IV-2 are two CS-*Ae. searsii* introgression lines with small 2Sˢ segments carrying *Pm57* (Type IV in Fig. 1b; 2Sˢ fragment sizes between 5.57 Mb and 9.12 Mb, ~1% of the 2Sˢ chromosome). TA3809 served as a control for these introgression lines. The heading date, one thousand-grain weight, and number of seeds per spike are presented as the means ± SDs from three replicates. The data for plant height, spike length, and spikelets per spike are presented as the means ± SDs from individual plants ($n = 15$). ns: no statistically significant difference according to a two-tailed Student's *t* test. Source data are provided as a Source Data file.

in Triticeae species but also in other species (Supplementary Data 2). Moreover, the sequences of single kinase-vWA proteins from Ae.bicornis.TB01.Un01G0378900.1 and Thint.05G0470400.1.p are very similar with Pm57 (>90%, Supplementary Fig. 17). These results suggest that the tandem kinase-vWA protein Pm57 is likely derived from a single kinase-vWA protein. Previous studies have shown that tandem kinase proteins may be derived from the duplication or fusion of two kinase domains. If two kinase domains of the same gene show high sequence similarity and originate from the same kinase family or subfamily, they are derived from gene duplication; if two kinase domains originate from two different kinase families or subfamilies or share relatively low similarity, they are derived from gene fusion[43]. The two kinase domains of Pm57 shared a relatively high similarity (58.04%) in amino acid sequence, and both were classified into the LRR_8B subfamily[47] and within the same group with two kinase domains of Lr9 (Supplementary Fig. 12). These results indicated that the two kinase domains of Pm57 could have resulted from a duplication event.

Although Pm57 was inferred to be an ortholog of Lr9, with a high protein sequence similarity of 88.3%, the amino acid sequences in the pseudokinase and vWA domains and protein structures are clearly different between Pm57 and Lr9 (Supplementary Fig. 5 and 17). The overlap analysis of the predicted protein structures revealed that the overlap of Pm57 and Lr9 was very high in kinase domains but poor in the disordered regions of the pseudokinase domain and the vWA domain (Supplementary Fig. 18). Further comparison of the overall disorder of Pm57 and Lr9 revealed that the disorder of Pm57 was greater than that of Lr9, especially in the pseudokinase domain and vWA domain regions (Supplementary Fig. 19). In addition, although purifying selection was observed for both *Pm57* and *Lr9*, the selection pressures applied to the kinase and pseudokinase domains markedly differed (Supplementary Table 5). Wang et al. hypothesized that the pseudokinase and vWA domains of Lr9 might serve as integrated decoys for the detection of pathogen effectors[38]. It is reasonable to speculate that the differences in pseudokinase and vWA domains

between Pm57 and Lr9 may lead to differences in the detection of *Bgt* and *Pt* pathogen effectors and thus confer resistance to powdery mildew and leaf rust, respectively. A similar case was recently described for WTK3 and RWT4, which showed high sequence similarity but conferred resistance to powdery mildew and wheat blast, respectively[27,51]. WTK3 and RWT4 were considered alternative alleles of the same gene with different functional specificities[51]. This phenomenon of functional specificity of orthologous genes or alleles is very worthy of further study to better elucidate the resistance mechanism of resistance genes to pathogens.

In summary, we cloned *Ae. searsii*-derived *Pm57*, which confers broad-spectrum and all-stage resistance against *Bgt*. *Pm57* encodes a protein consisting of tandem kinase domains and a vWA domain (WTK6b-vWA). Since Pm57 is an ortholog of Lr9, our results provide evidence for the resistance of orthologous genes or alleles to different diseases. The isolation of *Pm57* lays a solid foundation for further understanding the molecular mechanism underlying WTK-vWA-mediated resistance to various plant diseases. The identified introgression lines harboring *Pm57* in a small alien segment, as well as diagnostic function markers, will facilitate the improvement of elite wheat varieties with durable powdery mildew resistance.

## Methods

### Plant materials

The common wheat (*T. aestivum* L.) cultivar Chinese Spring (CS, TA3808), the CS *ph1b* mutant stock (TA3809), and the CS-*Ae. searsii* chromosome 2S$^s$ recombinant lines 89(5)69 (TA5109) and 89(6)88, both of which carry *Pm57*, were used for *Pm57* mapping and cloning in this study (Supplementary Table 7). The introgression line 89(6)88 was crossed with TA3809 to produce $F_2$[37] and $F_3$ populations with homozygous *ph1b* and segregated for the recombinant chromosome 2AS-2S$^s$S • 2S$^s$L-2AL carrying *Pm57*. The CS-*Ae. searsii* 2S$^s$ recombinants identified in this study were generated from $F_2$ populations using the *Pm57* flanking markers *X67593* and *X62492*. Line 89(5)69 was used for *Pm57* gene cloning and expression analyses. The wheat cultivar Fielder was used for wheat protoplast preparation and genetic transformation. In addition, two *Pm57* introgression lines, Type IV-1 and Type IV-2, developed in this study were used to investigate the effect of *Pm57* on the main agronomic traits. The CS-*Ae. searsii Pm57* recombinant 88R-3-19-1[37] was crossed with 22 wheat varieties to determine whether *Pm57* can confer powdery mildew resistance in diverse genetic backgrounds (Supplementary Table 7). A total of 71 wheat accessions, including diploid, tetraploid and hexaploid wheat accessions, were employed to investigate the presence of the *Pm57* resistance allele in other wheat germplasms (Supplementary Data 4). TA3808 and TA3809 were obtained from the Wheat Genetics Resource Center (WGRC) at Kansas State University, USA, and the CS-*Ae. searsii* 2S$^s$ recombinants TA5109 and 89(6)88 were developed by our laboratory and provided to WGRC. The 71 wheat accessions, except 89(5)69, were kindly provided by Prof. Zhongfu Ni at China Agricultural University, China. All materials were maintained at the experimental station of Henan Agricultural University, China.

### Phenotypic response to powdery mildew

The powdery mildew (*Bgt*) isolate E09 and 28 other genetically divergent isolates (Supplementary Table 6) collected from different regions of China were used to evaluate the powdery mildew resistance spectrum of *Pm57*, while the isolate E09 was used for all other resistance assays in this study. All the seedlings to be inoculated were grown in $7 \times 7$ cm nutrient pots under aseptic conditions. When the first leaves were fully unfolded, the seedlings were inoculated with *Bgt* isolates and grown in a greenhouse maintained at 18–24 °C, 16 h light/8 h dark cycle and approximately 70% relative humidity[37]. Disease symptoms were recorded 7 days post-inoculation (dpi) using a scale from infection type 0 to 4 (IT 0 for no visible symptoms, IT 0; for hypersensitive

necrotic flecks, IT 1-4 for highly resistant, moderately resistant, moderately susceptible and highly susceptible)[58]. Based on the IT scores, the tested plants were classified into two groups: resistant (R, IT 0-2) and susceptible (S, IT 3-4). CS was used as a susceptible control and for propagating *Bgt* isolates.

The resistance of the CS-*Ae. searsii* chromosome 2S$^s$ recombinants, EMS-induced susceptible mutants and *Pm57* $T_0$ transgenic plants was determined by inoculation with E09 and further confirmed using their self-pollinated progenies. If 10-15 progenies from each line were all resistant or segregated for resistance, the parental plant was considered resistant; however, if all the progenies were susceptible, the parental plant was considered susceptible. To evaluate the *Pm57* resistance spectrum, four seedlings of CS-*Ae. searsii* chromosome 2S$^s$ introgression line 89(5)69 and *Pm57* transgenic plants were inoculated with each isolate from a total of 29 *Bgt* isolates in triplicate assays. To investigate the potential influence of diverse genetic backgrounds on *Pm57* resistance, $F_1$ hybrids derived from crosses of the *Pm57* introgression line 88R-3-19-1 with 22 elite wheat varieties were subjected to *Bgt* E09 response assays, and four seedlings from each of the hybrids were subjected to each of three independent assays.

The powdery mildew resistance of primary transgenic plants ($T_0$) was tested using detached leaves. Briefly, $T_0$ seedlings were transplanted into plastic pots containing mixed soil composed of sand, peat, and perlite (1:1:1, v/v/v) and subsequently grown in a growth chamber for 14 days under normal growth conditions. Then, three detached leaves from each $T_0$ plant, susceptible control Fielder and resistant control recombinant 89(5)69 were placed on Phytagar media (0.5% Phytagar; 30 ppm benzimidazole), inoculated with *Bgt* isolate E09 and cultured in a greenhouse. ITs were recorded 7 days after inoculation. Additionally, assessments of powdery mildew reactions in $T_1$ transgenic plants during the whole growth period were conducted through natural transmission of the *Bgt* pathogen in the greenhouse. Approximately 12 plants of each $T_1$ transgenic line were potted. Simultaneously, the highly susceptible CS was inoculated to spread the *Bgt* pathogen in the greenhouse.

### Physical mapping of Pm57

The wheat-*Ae. searsii* 2S$^s$ recombinant population was developed from the cross of the CS-*ph1b* mutant stock TA3809 and 89(6)88[37]. Nine heterozygous resistant CS-*Ae. searsii* 2S$^s$ recombinants (88R-2-32-3, 88R-4-30-2, 88R-5-28-4, 88R-2-9-2, 88R-4-5-4, 88R-4-18-3, 88R-5-1-4, 88R-2-22-3, and 88R-3-6-2) were self-pollinated to produce a secondary mapping population for further mapping of *Pm57*. CS-*Ae. searsii* 2S$^s$ recombinants in the 5.13 Mb segment harboring *Pm57* were identified with the *Pm57* flanking markers *X67593* and *X62492*. The *Ae. searsii* (TE01)[1] and Chinese Spring (v2.1)[59] genome sequences were used for the development of STS-PCR markers. All the markers (Supplementary Data 1) were designed using DNAMAN 7 software (Lynnon Biosoft, San Ramon, CA, USA). DNA was extracted using the CTAB method, and genotyping of the recombinants was performed in 15 μL volumes containing 200 ng template gDNA. PCR was performed using 2x ES Taq MasterMix (CWBIO, China) under the following conditions: 94 °C for 3 min; 34 cycles of 94 °C for 30 s, 57 °C for 30 s, and 72 °C for 30 s; and 72 °C for 3 min.

### Mutant development and MutRNA-Seq

Seeds of 89(5)69 were soaked in distilled water for 6 h and treated for 16 h with a 0.6% (v/v) EMS solution with shaking at 150 rpm at room temperature. The solution was then removed, and the treated seeds were rinsed with running water for 6 h. The mutagenized $M_0$ seeds were planted in the field, and the $M_1$ seeds from each $M_0$ plant were harvested at maturity. The $M_1$ seedlings (10-15 plants for each family) were phenotyped with *Bgt* isolate E09 in a greenhouse. Susceptible $M_1$ plants were advanced to the $M_2$ generation, and 10-15 $M_2$ seedlings were tested to confirm their susceptibility.

MutRNA-Seq was performed as described by ref. 46. Specifically, five susceptible M$_2$ mutants (Mut51, Mut60, Mut141, Mut209 and Mut216) derived from independent M$_1$ families as well as the resistant wild-type parental line 89(5)69 were selected for RNA-seq. The first leaves from five plants of each genotype were sampled for RNA extraction at 5 dpi with *Bgt*. Total RNA was extracted using TRIzol reagent (TransGen, Beijing, China). RNA-seq was performed by Anno-road Gene Technology Co., Ltd. (Beijing, China). The Illumina HiSeq X Ten platform (Illumina, USA) was used, and 39.0, 58.8, 37.8, 44.6, 52.4 and 46.1 million 150 bp paired-end clean reads were produced for 89(5)69, Mut51, Mut60, Mut141, Mut209 and Mut216, respectively. The clean reads were subsequently mapped to the *Ae. searsii* genome assembly (https://ngdc.cncb.ac.cn/gwh/Assembly/24532/show) using HISAT2 (version 2.0.5, default parameters). The alignment results were further processed into BAM format using SAMtools (version 1.9) with default parameters. Gene expression levels were quantified using the featureCounts tool (version 1.4.4; parameters: -T 10 -t exon -g gene_id). Based on the alignment results, we conducted single nucleotide polymorphism (SNP) calling using BCFtools (version 1.7). Briefly, SNPs were detected using the commands bcftools mpileup (with the parameters "-A -C50 -Q20 -q30 -Ou -r -f") and bcftools call (with the parameters "-vmO z -V indels -o"). The SNPs were filtered according to the following set of read characteristics: SNP quality (QUAL < 30), genotype (GT = 0/1), low read depth (DP < 10), mapping quality (MQ < 40), non-EMS-induced point mutations (A/T to G/C transitions), and nucleotide positions with more than 2 samples supporting alternative alleles. These SNPs were excluded from further analysis.

### Gene cloning and sequence analysis

TRIzol reagent (TransGen, Beijing, China) was used for RNA extraction, and 2 µg of total RNA was used for cDNA synthesis using a HiScript II 1st Strand cDNA Synthesis Kit (+gDNA wiper) (Vazyme, Nanjing, China). The full-length genomic DNA (gDNA) and cDNA sequences of *G4* from 89(5)69 and each of the 15 susceptible mutants were amplified using the primers listed in Supplementary Data 1. PCRs were performed in 30 µL volumes using high-fidelity Primestar polymerase (TaKaRa, Dalian, China). The PCR conditions were previously described[60]. Specifically, 98 °C for 3 min; 34 cycles of 98 °C for 15 s, 60 °C for 20 s, and 72 °C for 3 min; and 72 °C for 3 min. The PCR products were sequenced by the Sanger dideoxy DNA sequencing method. The sequences of *G4* in each mutant and in the wild-type 89(5)69 were compared using DNAMAN 7 software.

### Alternative splicing variant analysis

RNA was isolated from 89(5)69 leaf tissue inoculated with *Bgt* isolate E09 at 0 and 24 hpi. The primers 1 F/3R1 for the full-length *G4* were used to clone alternative splicing variants from cDNA. The PCR products were purified and cloned and inserted into the pEASY-Blunt cloning vector (TransGen, Beijing, China). A total of 77 and 89 colonies at 0 and 24 h post inoculation (hpi), respectively, were selected for Sanger sequencing.

### Virus-induced gene silencing

Virus-induced gene silencing (VIGS) was performed as previously described[57]. In brief, the plants were grown in a growth chamber under 16-h light (24 °C)/8-h dark (18 °C) and approximately 70% relative humidity. To develop specific VIGS targets, we blasted the sequences of the *G4* and *G5* genes against the CS and *Ae. searsii* genome sequences. The 3' ends of the *G4* and *G5* fragments, 200-250 bp in length and with low similarity (identity <50% and stretches of 100% nucleotide identity<21 nt) with other genes, were selected as targets; these fragments were separately cloned and inserted into the BSMV-γ (BSMV, barley stripe mosaic virus) vector, resulting in the construction of γ-G4 and γ-G5. Equimolar amounts of in vitro transcripts of BSMV-α, BSMV-β and γ-G4 or γ-G5 were mixed to inoculate the fully expanded

second leaves of 89(5)69 seedlings, and leaves infected with BSMV-TaPDS and BSMV-γ (empty vector) were used as controls[61]. Approximately 14 days after viral infection, the 3rd and 4th leaves with symptoms of viral infection were sampled for *Bgt* inoculation and gene silencing expression analyses, respectively. For *Bgt* inoculation, the samples were placed on 1% agar plates supplemented with 20 mg/mL 6-phenyladenine (6-BA) and subsequently inoculated with *Bgt*. After seven days, the powdery mildew resistance phenotype was evaluated. The experiments were repeated three times.

### Gene expression analysis

For the gene expression analyses of *G4* and *G5* in the VIGS experiments, the 3rd and 4th leaves were mixed together as one sample, and three technical replicates were performed for each sample. RNA extraction and cDNA synthesis were performed as described above. Quantitative RT–PCR (qRT–PCR) analysis was carried out in a reaction volume of 20 µL using SYBR Mix (TaKaRa, Dalian, China) on a CFX96 real-time PCR detection system (Bio-Rad, Hercules, CA, USA). The conditions for qRT–PCR were as follows: denaturation at 95 °C for 4 min, followed by 40 cycles of 94 °C for 20 s, 60 °C for 20 s and 72 °C for 20 s. Wheat *TaActin* was used as an internal reference gene[27]. The transcript levels were calculated using the comparative CT method[62]. The primer pairs 3 F/2 R and M-5F/5 R were used to determine the expression of *G4* and *G5*, respectively (Supplementary Data 1).

To study the tissue-specific expression of *Pm57*, seedling roots and leaves from the three-leaf period and adult plant roots, stems, and flag leaves from the heading stage were obtained from 89(5)69. Three biological replicates with tissues from four individual plants mixed as one replicate were used for expression analysis. For expression analysis of *Pm57* and the pathogenesis-related (PR) genes *PR1*, *PR2*, *PR3*, *PR4* and *PR9* in 89(5)69, the transgenic lines (L1 and L16), and Fielder, the first leaves at the two-leaf stage were used to extract total RNA before inoculation (0 h) and at 12, 24, 36, and 48 hpi with the *Bgt* isolate E09; for each time point, qRT–PCR was performed on three biological replicates, with the first leaves from four individual plants mixed as one replicate. The RNA extraction, cDNA synthesis and qRT–PCR conditions were the same as those above. The primers used to evaluate the transcript levels of the *Pm57* and *PR* genes are listed in Supplementary Data 1. The wheat *TaActin* gene was used as the endogenous control. The comparative CT method was used to quantify relative gene expression.

### Wheat transformation

The full-length coding sequence of *G4* from 89(5)69 was inserted into pWMB110 vectors using the restriction enzymes *Bam*H I under the control of the maize *ubiquitin* (*Ubi*) promoter. Wheat transformation was performed using the *Agrobacterium*-mediated method with strain EHA105 and calluses induced from cv. Fielder immature embryos[63]. To determine positive transgenic events, DNA was extracted from leaves of independent T$_0$ plants, and specific PCR primers (3 F/2 R) were designed to amplify a 344 bp fragment of the *G4* gene. qRT–PCR analysis was performed to evaluate the expression levels of *G4* in the first leaves of transgenic wheat plants in the T$_1$ generation. The first leaves of four seedlings at the two-leaf stage were sampled for each line to extract RNA. RNA extraction, cDNA synthesis, and qRT–PCR analysis were performed as above. All the reactions were performed in triplicate. The disease responses of the transgenic plants to powdery mildew were tested as described above.

### Subcellular localization analysis

To determine the subcellular location of *Pm57*, the coding sequence of *Pm57* was cloned into pJIT163-GFP vector, in which the expression of *Pm57-GFP* was driven by the *CaMV35S* promoter. An empty vector, pJIT163-GFP, was used as the negative control. The constructs were delivered into wheat protoplasts via PEG-mediated transformation

according to a prior study[64]. Under an induction of 40% PEG-4000, control or recombinant plasmids (10 μg per construct) were co-transformed into wheat protoplasts with nucleus marker plasmid *AtPIF4-mCherry*[65]. The transformed protoplasts were cultured at 25 °C for 16 h under dark conditions, and observed using a laser confocal microscope (A1F, Nikon, Tokyo, Japan). Image acquisition was conducted with the NIS-Elements Viewer Imaging Software (version 5.21.00). The subcellular localization of Pm57 protein was determined at three times.

### Pm57 protein 3D modeling prediction
To predict the 3D structure of Pm57 and other WTKs, we used the open-source code of AlphaFold v2.1[44]. We input the amino acid sequence of each WTKs into AlphaFold v2.1, and obtained five unrelaxed, five relaxed and five ranked models in.pdb format. Among the output models, the ranked_1.pdb model had the highest confidence with the best Local Distance Difference Test (lDDT) score were utilized. The structural graphics and the positions of amino acid substitutions were visualized using PyMOL (v. 2.6.0a0).

### Detection of $H_2O_2$ accumulation and plant cell death
To detect the accumulation of $H_2O_2$, the first leaves cut from 89(5)69 and CS at 6, 12, 24, 36, and 48 h post inoculation (hpi) were immediately incubated in a 3, 3'-diaminobenzidine (DAB) solution (1 mg/mL, pH 5.8) for 12 h at 25 °C, and then bleached in absolute ethanol. Before assessing the accumulation of $H_2O_2$, the bleached leaves were incubated in a 0.6% (w/v) Coomassie blue solution for 10 s and then washed with water. To detect plant cell death, the primary leaves from the 89(5)69 and CS at 48 hpi were incubated in a 0.4% Trypan blue solution for 1 min in boiling water, washed with sterile water, bleached for 16 h in chloral hydrate solution (2.5 g/mL), fixed for 20 min in ethanol-acetic acid 3:1 (v/v), and stained in a 0.6% (w/v) Coomassie blue solution for 30 s. The treated leaves were viewed under a microscope (Olympus BX53). The ratios of ROS and cell death were calculated as described by Li et al.[66]. Specifically, three independent experiments were performed, and three leaves were collected as biological replicates for each time, and at least 100 infected cells per leaf were observed to identify the ROS and cell death responses.

### Collinearity analysis, homology searching and phylogenic analysis
Collinearity analysis among different species or subgenomes was performed using the online tool Triticeae-GeneTribe with default parameters[67]. The WheatOmics 1.0 (http://wheatomics.sdau.edu.cn/) and Phytozome v13 database (https://phytozome-next.jgi.doe.gov/) were used to find proteins similar to Pm57 in plant genomes[68,69]. Pm57 were used as queries for BLAST analysis with default parameters. All the retrieved proteins were scanned by website pfam 35.0 (http://pfam.xfam.org/) in batch mode with an E value of 0.01, and the retrieved proteins with kinase domain and vWA domain were selected. For kinase domain analysis, the 182 putative kinase or pseudokinase domains used for phylogenetic analysis of WTK1 (Yr15)[43] and WTK3 (Pm24)[27] were also used in this study. In addition, the proteins with a tandem kinase -vWA structure were included in the phylogenetic analysis of kinase domains. Multiple sequence alignments were carried out with ClustalW software with default settings[70]. The conserved motifs in the Kin I and Kin II domains were previously annotated by Klymiuk et al.[43]. Classification of Pm57 to protein kinase subfamily was performed by HMMER v3.3.2 (http://hmmer.org/) with HMM models developed by Legti-Shiu and Shui[45]. The classification was further confirmed by a neighbor-joining (NJ) phylogenetic analysis, which was conducted using MEGA7.0[71] with Poisson model and 1000 repetitions of bootstraps. Lr9-vs-Pm57 reciprocal BLAST is performed between *Ae. searsii* (TE01) and *Ae. umbellulata* (TA1851) assemblies by BLASTN

2.13.0+ software with default settings. The *Ka* (non-synonymous substitution) and *Ks* (synonymous substitution) substitution rates was calculated to estimate the selection pressure between *Pm57* and *Lr9* genes using KaKs_Calculator 3.0[72].

### Agronomic trait evaluation of *Pm57* transgenic lines and introgression lines
Three $T_2$ transgenic lines and two introgression lines of *Pm57*, and the control lines were planted in the experimental fields of Henan Agricultural University (Zhengzhou, China) using a randomized block design with three replications. Each plot consisted of two 1.5-m rows spaced 25 cm apart, and 20 seeds were sown in each row. Field managements including irrigation, fertilization, herbicide and pesticide applications strictly followed local practice.

Agronomic traits were phenotyped as previously described[73]. In brief, for each replicate, five plants for each plant material are selected to measure agronomic traits that includes heading date (the date from sowing to the date when half of the spikes had emerged from the flag leaf of the main tiller), plant height (the length from the ground to the tip of the main spike excluding awns), spike length (the length from the rachi base to the uppermost spikelet top excluding awns), spikelet number per spike, seeds per spike and one-thousand-grain weight. The significance of differences among means of agronomic traits was determined using two-tailed Student's *t*-test.

### Prediction of the disorder of Pm57 and Lr9
The predisposition of Pm57 and Lr9 to a disordered structure was assessed by the Predictor of Naturally Disordered Regions (PONDR; http://www.pondr.com/) server with default settings.

### Reporting summary
Further information on research design is available in the Nature Portfolio Reporting Summary linked to this article.

## Data availability
Data supporting the findings of this work are available within the paper and its Supplementary Information files. Detailed sequence data of *Pm57* gene was deposited in NCBI database under accession OQ675542. The MutRNA-Seq data derived from the WT and five mutant plants have been deposited in NCBI's Sequence Read Archive (SRA) under accession number PRJNA947672. Public databases WheatOmics 1.0 (http://wheatomics.sdau.edu.cn/) and Phytozome v13 (https://phytozome-next.jgi.doe.gov/) were used in this study. Source data are provided with this paper.

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

## Acknowledgements

We are grateful to Prof. Zhongfu Ni and Huiru Peng from China Agricultural University, Prof. Zhiyong Liu from Chinese Academy of Sciences, for their advice and supports during this research. We are also grateful to Prof. Pengtao Ma of Yantai University, Yantai, Shandong, China, for providing *Bgt* isolates and powdery mildew resistance assays. This research was financially supported by the National Natural Science Foundation of China (31971887 and 32372089 to W.X.L., and 32272070 to H.H.L.), the Scientific and Technological Research Project of Henan Province of China (222103810004 to Y.Z.) and the Key Scientific Research Projects of Higher Education Institutions in Henan Province (23A210020 to Y.Z.) and South Dakota Wheat Commission (3×2030) and USDA AFRI 2022-68013-36439 (WheatCAP) to S.K.S.

## Author contributions

W.X.L., H.H.L. and S.K.S. designed the study. Y.Z., Z.J.D., J.N.M., C.M., X.B.T. and J.Q.H. performed the research. Q.W.L., H.H.B., W.Y., T.L., H.S.G., Z.B.Z., A.Z.C., B.L., H.H.L. and S.K.S. analyzed the data. Y.Z., W.X.L. and S.K.S. wrote the manuscript and all authors contributed to revision and editing.

## Competing interests

The authors declare no competing interests.
