## [Peer Review File · Nature Communications]

Pm57 from *Aegilops searsii* encodes a tandem kinase protein and confers wheat powdery mildew resistanceReviewers' Comments:

Reviewer #1:

Remarks to the Author:

Zhao and colleagues describe their work at cloning the wheat powdery mildew resistance gene Pm57, a gene derived from the wheat relative *Aegilops searsii* (Sitopsis section). Using a wheat introgression line, the authors used a combination of map-based cloning, mutagenesis, virus-induced gene silencing, and transgenic complementation to show that a protein kinase-protein kinase-vWA encoding gene underlies Pm57-mediated resistance. Pm57 is localized to the cytoplasm and used several histological and molecular assays to establish the characteristics of the immune response. Analysis of the locus found Pm57 was present in closely related species, although no specific analysis was performed on the emergence of this gene in the evolution of the Triticeae. Lastly, the authors performed field evaluations to assess the use of Pm57 in a transgenic context and found it had no obvious impact on diverse agronomic traits.

Major concerns

Major concern 1. The authors identified 15 independent mutants that were loss-of-function for Pm57. Five mutants were initially sequenced using RNAseq and three were found to carry mutations in G4 (WTK7-vWA). Sequencing cDNA and gDNA of the 15 mutants found that 10 mutants carried polymorphisms that impacted the protein sequence of WTK7-vWA. The authors correctly comment on the potential of the additional five mutants to uncover components of Pm57 immune signaling. I was surprised to not see some form of complementation tests performed between mutants (including Pm57 and non-Pm57 mutants). Furthermore, was Pm57 expressed in the five mutants? Was RNAseq performed for all?

Major concern 2. The authors provide a limited analysis of 182 protein kinase domains. Similar to Li et al., I suggest the authors look at "Diversity, classification and function of the plant protein kinase superfamily" by Melissa D. Lehti-Shiu and Shin-Han Shiu for a protocol on classifying the protein kinase domain in Pm57. This analysis should guide the classification of individual protein kinase domains and the relevant subfamilies to evaluate the evolutionary origin of the protein kinase domains in Pm57. Phylogenetic trees must be bootstrapped and based on appropriate sampling to have an ability to infer evolutionary relationships.

Major concern 3. Is Pm57 the ortholog of Lr9? This relationship was unclear and the phylogenetic analysis suggests they are highly related.

Minor concerns

1. Results, line 96. "on genome sequence" to "on the genome sequence".
2. Figure 1. Panel C should be relative to the genomic distribution, not the cartoon version.
3. Line 114. Was RNAseq also performed on the parent of the mutagenesis population?
4. Figure 4. Individual domains should be labelled in panel B.
5. References, Wang et al. is now published. <https://doi.org/10.1038/s41588-023-01401-2>
6. Results, it should be mentioned in the results that the ubiquitin promoter was used to drive Pm57 expression to generate transgenics.
7. Figure 6. While I appreciate that the experimental plan in Figure 6 was to determine if Pm57 had a negative agronomic characteristics, it was unclear why the various introgression lines were not used in field experiments in parallel. It would be useful to see the contrast between introgression lines and transgenic lines.
8. There was no evaluation of natural variation in Pm57 in *Aegilops searsii* (and also related species). There appear to be four data sets of *Aegilops searsii* RNAseq in NCBI that could be investigated.

Reviewer #2:

Remarks to the Author:

In this manuscript, Zhao and colleagues report the cloning of the powdery mildew resistance gene Pm57 from bread wheat. Pm57 has been introgressed into bread wheat from the wild wheat relative

Aegilops searsii. Using map-based cloning, EMS mutagenesis, virus-induced gene silencing, and transgenic complementation, the authors identify a wheat tandem kinase (WTK) protein fused to a vWA domain as Pm57. This is only the second example of a WTK-vWA encoding gene involved in disease resistance. In general, the study is well executed, well written, and of general interest to the broad readership of Nature Communications. I do have some comments that the authors should consider before publication.

Major comments

- For the transformation, the authors used an overexpressing promoter (Ubi) and coding sequence. There are several examples of disease resistance genes that have possibly been wrongly identified because of candidate gene overexpression. A critical experiment is to demonstrate that the overexpressing transgenic lines maintain race-specificity. Are there any powdery mildew isolates that are virulent on Pm57? Fig. S2 does not include a wheat line with the endogenous Pm57 gene, so it is not possible to compare the Pm57 transcript levels in the transgenic lines to the endogenous Pm57 expression levels?
- The authors state in the abstract that Pm57 transgenic lines had 'no adverse effects on agronomic traits'. This is quite a strong statement. The data shown in figure 6 are based on relatively few plants grown in single rows. The authors did not perform proper yield trials (macro-plots) in different environments to test for agronomic performance. This is not required here, but I suggest that the authors are a bit more conservative in their wording. It is still possible that Pm57 has a minor yield penalty that could not be assessed with the current experimental set-up.
- The discussion mainly repeats the main findings of the manuscript. For me, the most exciting conclusion of this manuscript is the fact that orthologous WTK-vWA genes (Pm57 and Lr9) from different *Aegilops* species confer resistance against different pathogens. This could be highlighted and discussed more prominently (maybe even in the abstract) and in more detail. Did the authors overlap the predicted protein structures of Pm57 (Fig. 4b) and Lr9? Are there any structural differences that could explain the difference in recognition specificity between Pm57 and Lr9? The authors provide an amino acid alignment of Pm57 and Lr9 (Fig. S11), showing that most differences are in the pseudokinase and vWA domains. Are these amino acid polymorphisms mainly affecting surface-localized residues that might bind a putative effector? In addition, a Ka/Ks analysis could be done to test if the two kinase domains are under opposing selective pressure?

Minor comments

- Title: Not sure if the word 'novel' is appropriate in the title.
- Abstract: It is unclear to me how a 'poor understanding of the resistance mechanisms' has constrained the use of wild relatives for wheat improvement. I would argue that the main constraints are difficulty to make crosses and linkage drag.
- Line 34: The global wheat loss estimates due to powdery mildew are in the range of 1% (Supplementary Table 3 of Savary et al. 2019 Nature Ecology & Evolution 3: 430-439). The 10-15% indicated here seem to be overestimated.
- Line 63: 'unavailability of annotated reference genomes.'
- Line 71-73: Please indicate how Pm57 was introgressed into bread wheat. Was this done through homologous recombination? Was irradiation required? This is mentioned later and in previous publications, but I think it is important to repeat this here. Was it confirmed that the introgression is indeed homologous?
- Lines 83-84: Was Pm57 mapped in *Ae. searsii* or in a bread wheat introgression line? In other words, was the gene mapped to the long arm of chromosome 2SS or to an *Ae. searsii* chromosome segment located on bread wheat chromosome arm 2B? This is a fine but important detail. The wording here suggests mapping in *Ae. searsii*, but the experiments were done in a bread wheat introgression line. Are the authors sure that the introgression occurred at a homologous position? For Lr9 / Lr58, for example, the translocation from the wild relative was not homologous, which caused confusion during the mapping.
- Line 85: 'were used to develop a large'
- Line 112: The statement 'were identified by Pm57-specific markers and a 55K SNP chip' is not clear.

I assume the mutants were identified because they were susceptible. Does this sentence indicate that the genetic integrity of susceptible putative mutants were validated using markers and a 55K SNP chip? If so, the data should be provided as supplementary material. How many susceptible putative mutants were discarded because they showed a different genotype with the Pm57-specific markers and the SNP array? This needs more explanation here (it is mentioned in the methods section, lines 384-386, but I suggest moving this part to the results section).

- Please provide representative images of all 15 mutants in the supplementary material. In particular, was there any phenotypic difference between the 10 Pm57 mutants and the 5 putative second-site mutants?
- Line 437: 'In each plot, 5 plants were chosen to measure various agronomic traits.' This does not match with the numbers provided in Fig. 6 ($n = 3$, or $n = 15$).
- Please include a scale bar in Fig. S1
- The resolution of Fig. S5 needs to be improved.

Reviewer #3:

Remarks to the Author:

Zhao et al. reported cloning Pm57, the first gene isolated from *Aegilops searsii*. It encodes a wheat tandem kinase (WTK) protein with putative kinase-pseudokinase domains followed by a von Willebrand factor A (vWA) domain, designated WTK7-vWA. The experimental approach and the described results are solid. Independent mutants, gene silencing, and transgenic assays validated Pm57. These findings highlight the emerging role of kinase fusion proteins in plant disease resistance and provide a powerful gene for wheat breeding. However, the writing should be improved, the English is not good, and the interpretations of results should be improved. Some parts of the introduction and discussion should be improved, focusing on the main topic and going more in-depth.

Another wheat tandem kinase with a vWA domain, designated WTK6-vWA, was recently cloned. However, the relationships between these two genes are poorly described. The authors mostly ignored WTK6-vWA and only mentioned it in the discussion part towards the end of the manuscript. WTK6-vWA and WTK7-vWA show 88.3% homology but confer resistance to two diseases: leaf rust and powdery mildew. A similar case was recently described for WTK3 and RWT4, which show high sequence similarity yet confer resistance to different diseases, such as powdery mildew and wheat blast. WTK3 and RWT4 share the same chromosome location. Therefore, they can be considered alternative alleles of the same gene with different functional specificities. We are wondering whether similar relationships exist in the case of WTK6-vWA and WTK7-vWA.

According to Wang et al. (2023), the WTK6-vWA -containing *Ae. umbellulata* segment is homoeologous to the bread wheat chromosomes 2 (See Peer review file: https://static-content.springer.com/esm/art%3A10.1038/s41588-023-01401-2/MediaObjects/41588_2023_1401_MOESM3_ESM.pdf). Since chromosome 2S of *Ae. searsii* is also homoeologous to wheat chromosomes 2 (Dong et al. Int J Mol Sci. 2020), then WTK6-vWA and WTK7-vWA probably share a common origin and should be considered orthologs or homoeologs. Thus, we propose to describe WTK6-vWA already in the introduction of the current manuscript, compare the evolutionary relationships between WTK6-vWA and WTK7-vWA in the results, provide evidence that they are orthologs, properly discuss these relationships, and compare to similar cases mentioned in the literature, such as WTK3 and RWT4.

If WTK6-vWA and WTK7-vWA are orthologs, maybe WTK7-vWA (Pm57) should be named WTK6-vWAPm. Please note that WTK7 was already claimed by Zhiyong Liu's group for the wheat tandem kinase gene WTK7-TM in a paper that is now under revision in Nature Communications.

Questions/suggestions

1. Lines 51-52,

Please consider combining alleles/homologs of the same genes, such as Pm21/Pm12, Pm60/MIIW172/MIIWE8, etc., or provide a more detailed explanation for the possible

homologous/orthologous/allelic relationships between similar genes.

2. Lines 53-54,

Please carefully check all references since we noticed a problem with the numbering of some of the references. For example, Pm4 is described in reference 19, not in reference 3. Furthermore, the reference for WTK4 is missing.

3. Lines 66-68,

This sentence is not clear. Please improve. Maybe you want to change "thus" to "as well as".

4. Lines 74-75.

WTK7- vWA is the second gene discovered with this structure. Therefore, the terms "novel" and "unusual" should not be used in association with its structure.

5. Line 78

I do not see some important molecular mechanism results here. Please change this statement to fit the results.

6. Line 84

Please provide more clear information on wheat-Ae. searsii 2S.

7. Line 87

Please provide a full name for Chinese Spring (CS) when mentioned for the first time.

8. Line 88-89

Improve English.

9. Fig. 1b.

Please explain what "Type I – Type VI" is.

Add spaces between marker names (X1, X2, etc.) in the map, or use a smaller font.

10. Could you give a clear explanation why there were only three of the five mutants that had the missense mutation of gene G4?

11. Figure 2b

Mut216 G6587A? Something is wrong here.

In lines 128-129, the title of Fig. 2b is confusing and does not reflect the content of the figure. We propose to mention the isoforms in the title.

12. Lines 185-188

The sentence is too long and not well written. Please rephrase.

13. Please add information on the promoter, CDS, or genomic DNA in the main text.

14. Line 197 and title of Fig 5.

These titles do not reflect the content of the paragraph and the figure. These experiments describe microscopic observations associated with plant-pathogen interactions and not studies of molecular mechanisms. We propose changing the titles to "histopathological characterizations of..." to reflect the content.

15. Figure 5.

CS also showed ROS accumulation. We propose to show quantitative differences between resistant and susceptible accessions. You can see an example in the following publication: "Intracellular reactive oxygen species (intraROS)-aided localized cell death contributing to immune responses against wheat powdery mildew pathogen." <https://apsjournals.apsnet.org/doi/10.1094/PHYTO-07-22-0271-FI>.

16. Lines 206-207.

Fig 5C shows the down-regulation of Pm57 by Bgt at 48 hpi. Please refer to this result in the text.

17. 210-211.

The conclusion is wrong since the PR genes are also upregulated in CS after Bgt infection.

18. Line 221

The evolutionary study is ignoring completely WTK6-vWA. Therefore, missing the most important issue in TKP convergent evolution. We propose adding WTK6-vWA to the evolutionary analysis. It gives the impression that the authors are trying to hide WTK6-vWA, while this is one of the most exciting findings of this work dealing with homoeologous genes that provide solutions against different pathogens.

19. Lines 275-280

The paragraph is confusing. First, you say that the A. searsii genome is absent and then present. We suggest better describing the story less confusingly.

20. Fig.6.

The seeds seem bigger in transgenic lines. Did you measure seed size? Were these plants infected with Bgt in the field?

21. Please provide the Kinase I and II subfamily (e.g., LRR-8B, WAK, etc.) according to rice or Arabidopsis kinome, as previously published for other WTK genes.

22. Lines 470-471

The 182 putative kinase or pseudokinase domains used for phylogenetic analysis of WTK3 were discovered by Klymiuk et al. (2018)³³ following the cloning of WTK1 and not by Lu et al. (2020)³. Therefore, we propose to give proper acknowledgment by adding "WTK1 (Yr15) 33 and WTK3 (Pm24) 3....."

23. Lines 474-475

The conserved motifs in the Kin I and Kin II domains of WTKs were first described by Klymiuk et al. (2018)³³ following the cloning of WTK1 and not by Lu et al. (2020)³. Therefore, we propose giving proper acknowledgment to Klymiuk et al. (2018).

24. Line 554

Update reference 27; The paper by Wang et al. 2023 is now published in Nature Genetics.

Reviewed by Tzion Fahima and Yinghui Li

Reviewer #4:

Remarks to the Author:

Zhao et al. revealed the cloning and characterization of Pm57, a gene that confers wheat resistance to multiple isolates of powdery mildew. The researchers have dedicated substantial efforts towards fine-mapping a gene derived from a wild wheat source in a unique manner and identifying a novel kinase fusion protein. Furthermore, they have employed several complementary strategies to functionally validate and characterize the gene. However, the material and methods section lacks crucial information, making it difficult to assess the execution of the experiment. A more rigorous description of the experiment and data is necessary. For instance, it remains unclear whether the T2 individuals used for field evaluation were fixed for Pm57, and there is limited information about the protocol employed to assess various agronomic traits.

Based on the provided data, it appears that there are certain speculative aspects in the findings. For instance, the claim of a correlation between H₂O₂ accumulation and the inhibition of haustorium formation lacks quantitative data and is not clearly supported by the presented images. Furthermore, saying that Pm57 acts as an upstream regulator is quite speculative as you analysed expression of PR in different genetic backgrounds (CS and 89(5)69). To provide more robust conclusions, it would have been beneficial to include the analysis of expression in transgenic models.

The absence of expression data for the VIGS experiment makes it difficult to ascertain whether Pm57 expression was genuinely downregulated.

The quality of images is not good which does not facilitate the analysis of symptoms for instance.

There are typos. Some references are not the right ones.

Could you please rearrange your materials and methods section to follow the results section?

It seems that the initial two paragraphs of the discussion section serve almost as a summary of the study, which could be redundant considering the preceding sections. Furthermore, the inclusion of figures from the supplementary section exclusively described in the next paragraphs may be better suited for the results section. In the results section, a dedicated paragraph describing the protein in comparison to other known kinases would provide a more focused and informative presentation of the findings.

Further comments:

L27 « powerful » seems inappropriate.

L35 this is not the right reference. there are few important references estimating the global impact of wheat diseases.

L39 I'm not sure that this reference is adapted here.

L40 Could you please provide references?

L45 I do not agree with this sentence. With the availability of new genomic resources and tools, I'm not sure that the classical map-based cloning strategy is the most effective as demonstrated by the cloning of numerous R genes using other strategies. Furthermore, in the absence of a sequenced and annotated genome, it is almost impossible to identify candidate genes. In the following sentences, you said that map-based cloning is time-consuming and challenging and that genome sequence facilitates map-based cloning, which is a bit contradictory to your previous sentence. I recommend rewriting this section. Usually, the strategy for cloning dependant on many features such as the origin of your genes, and its position on the chromosome....

L85 Which recombinants?

L87 Which kind of individuals, provide the generation

L98 it would be great here to know how those 12 genes are conserved on the susceptible parent CS.

L110 you treated M0 seeds and end up with M2 families, Are those M2 obtained directly from the M0 plants? If yes they should be M1?

L111 In the material section it is mentioned 10-15 plants

L112 what do you mean by independent mutants? Do they originate from different M2 families?

L113 I think the sentence should be modified as the susceptible were not identified by Pm57 markers, they were identified after phenotyping with isolate Bg09. What was further validated in M3 (and how many plants in M3).

L116 I supposed that the RNA-Seq data were aligned against the Pm57 candidate genes and not the opposite. Please correct.

L136 sequenced instead of cloned?

L139 Could you please mention how the sequence of this gene was different between TE01 and 89(6)69.

L140 There is no material and method section for this result !

L157 The plants were not as fully susceptible as the susceptible control CS. Furthermore, the absence of expression data hinders our ability to draw conclusions regarding the role of G4 expression in resistance.

L161 Indicate what part of G4 was used for transformation.

L162 Quality of images from figure 3a are not really good but it seems that pustules are also present in other transgenic families like L3, L9, L18? If this is right how do you explain this observation?

L167 Are the T1 named L1, L2... progenies of the T0 named L1, L2 respectively? I'm not sure to understand why T1 L3 is not shown in Figure 3 and Figure S2? as it has a strange phenotype in T0 it would have been great to understand why.

L168 it is not explained why the expression of G4 was studied only in some T0 and not all. It would have been great to see if there is a correlation between the level of expression and the phenotypes as some T0 show few pustules despite the presence of G4.

L173 which generation

L200 it is not mentioned in the legend of figure 5 that Coomassie blue was used for staining. Could please show on your figure the inhibition of haustorium formation.

L204 this is speculation as you do not have any quantitative evaluation of cell death and H2O2 and haustorium formation.

L206 What mean highly expressed compared to what ?

L210 this is also speculation as the genetic background you used for the analyses are quite different. We would be able to suggest this if you would have used the transgenics for the expression data.

L231 here you evaluate if the resistance allele if present not if the gene is present.

L238 I guess there are few wheat cultivars as we do not have this information and most of them are from China so you cannot say that Pm57 has not been used in modern wheat breeding, possibly in China but what about breeding programs in other part of the world?

L239 Please provide the exact PCR condition used for the diagnostic marker.

L255 which recombinants?

L353 Provide the type of population.

L359 Provide the number of individuals evaluated for each type of experiment, how many F3..

L361 Which leave was inoculated.
L374 I guess the genome of CS was used as well?
L388 Which part of the plant was collected? How many plants? Which generation? Which time point? Please provide details about the sample.
L392 please provide the numbers of reads per sample and the parameters used for alignment.
L400 low similarities, indicate a threshold. Which regions were targeted for VIGS? In which conditions were the plants grown for VIGS
L409 provide more information in Sup table 1 about the usage of primers and for instance which one was used to clone the cDNA.
L424 coding sequence from which accession?
L429 Which primers and conditions? At which stages leaves were sampled for RNA extraction? How many plants replicates? how cDNA were obtained... too many missing informations.
L436 What is a regular field management? Please provide detail.
L438 How many spikes were evaluated for each of the five plants. Please provide more detail about how the different traits were phenotyped.
L441 how the protoplasts were obtained ? how was done image acquisition? How many replicates? How the transformation was done, which plasmids concentration, provide reference for the AtPif4 constructs.....
L472 which filters/thresholds were applied?

Figure 1

Explain what is 89(6)88. Provide the number of recombinants for each type. b) Is it a genetic or physical map as distances are indicated in Mb but you said genetic map in the legend? Provide a scale. If this is a genetic map you should expect to have recombination between each marker as none of them co-segregate, which is not the case according to your type of recombinant. Please explain. c) please spell out WTK and vWA.

Figure 2 could you please explain the L4 positive and the L4 negative

Figure 5 L218 in CS as well. The analyses of expression seems to be done differentially between the Pr genes and Pm57. Could you please describe the methods for both analyses. There are significant overexpression for Pm57 but compared to what?

Sup Fig2 What are the replicates? One leave each, two leaves? During one experiment?

Sup Fig4 reference 29 is not the right one. There is no explanation of RBH, SBH, in last genome? How the small phylogenetic tree was built?

Sup Fig6 Indicate the size of bands of the ladder or give the exact reference

Sup Fig7 no materials and methods section at all for this study !

Sup Fig8 Not material and methods section for this experiment, how many F1 were evaluated? Pm57 was not transformed

Sup Table 2 What is the type? Mutant name are different from the text or Figure 2

Sup Table 4 How was performed the manual annotation? Which reference for CS? For the annotated gene, do they seem functional or pseudogene? What is the percentage of identity with Pm57

Sup Table 5 Please provide more information about the isolates

Sup Table 6 please provide information in the hexaploid wheat are landraces or cultivars and if these accessions are available in a genetic resource center.

REVIEWER COMMENTS

Reviewer #1 (Remarks to the Author):

Zhao and colleagues describe their work at cloning the wheat powdery mildew resistance gene *Pm57*, a gene derived from the wheat relative *Aegilops searsii* (Sitopsis section). Using a wheat introgression line, the authors used a combination of map-based cloning, mutagenesis, virus-induced gene silencing, and transgenic complementation to show that a protein kinase-protein kinase-*vWA* encoding gene underlies *Pm57*-mediated resistance. *Pm57* is localized to the cytoplasm and used several histological and molecular assays to establish the characteristics of the immune response. Analysis of the locus found *Pm57* was present in closely related species, although no specific analysis was performed on the emergence of this gene in the evolution of the Triticeae. Lastly, the authors performed field evaluations to assess the use of *Pm57* in a transgenic context and found it had no obvious impact on diverse agronomic traits.

We thank you for your comments and positive assessment of the manuscript.

Major concerns

Major concern 1. The authors identified 15 independent mutants that were loss-of-function for *Pm57*. Five mutants were initially sequenced using RNAseq and three were found to carry mutations in *G4* (*WTK7-vWA*). Sequencing cDNA and gDNA of the 15 mutants found that 10 mutants carried polymorphisms that impacted the protein sequence of *WTK7-vWA*. The authors correctly comment on the potential of the additional five mutants to uncover components of *Pm57* immune signaling. I was surprised to not see some form of complementation tests performed between mutants (including *Pm57* and non-*Pm57* mutants). Furthermore, was *Pm57* expressed in the five mutants? Was RNAseq performed for all?

>Our response: Partial mutants that uncover the causal TKPs have been reported previously, such as *WTK3* (15/26) and *WTK5* (7/14), where about half of the susceptible mutants represented second-site mutations. In addition, one out of 121 loss-of-function mutants did not carry a mutation in *Lr9*. We obtained 15 susceptible mutants in *Pm57* lines, of which 4 are second-site mutations (the previous statement of 5 is incorrect), which support a partner protein may be involved in TKP signaling. We also calculated the probability that the 11 independent mutations were the result of chance alone based on Sanchez-Martin et al. (2016). The probability that the 11 independent mutations in *Pm57* are the result of chance alone is 1×10^{-11} .

We have analyzed the expression levels of *Pm57* in these four non-*Pm57* mutants (see Lines 375-376). They showed a similar expression level with the resistant wild type parental line 89(5)69 (see Sup Fig. 15), indicating that these

four mutants will be important materials for studying the components of the *Pm57* immune signaling pathway.

We did not perform MutRNA-seq on all mutants, but selected 5 mutants instead. Our selection is based on the preprint report of Yu et al. (2022); their analysis indicated that in the scenario of scrutinizing all genes in a discrete mapping interval, the minimum number of independent mutants required for identifying a candidate gene with a 2000 bp CDS at $p = 0.01$ is three. In addition, our selection is also supported by Wang et al. (2023) who recently showed that with 4-5 independent mutants, the probability of identifying a false positive transcript was very low (Supplementary Note 1, Wang et al., 2023).

Sánchez-Martín, J. et al. Rapid gene isolation in barley and wheat by mutant chromosome sequencing. *Genome Biol* 17, 221 (2016).

Yu, G. et al. *Aegilops sharonensis* genome-assisted identification of stem rust resistance gene *Sr62*. *Nature Communications* 13, 1607 (2022).

Wang, Y. et al. An unusual tandem kinase fusion protein confers leaf rust resistance in wheat. *Nat Genet* 55, 914-920 (2023).

Major concern 2. The authors provide a limited analysis of 182 protein kinase domains. Similar to Li et al., I suggest the authors look at “Diversity, classification and function of the plant protein kinase superfamily” by Melissa D. Lehti-Shiu and Shin-Han Shiu (2012) for a protocol on classifying the protein kinase domain in *Pm57*. This analysis should guide the classification of individual protein kinase domains and the relevant subfamilies to evaluate the evolutionary origin of the protein kinase domains in *Pm57*. Phylogenetic trees must be bootstrapped and based on appropriate sampling to have an ability to infer evolutionary relationships.

>Our response: As suggested, we have classified the protein kinase domains in *Pm57* based on the protocol described by Melissa D. Lehti-Shiu and Shin-Han Shiu (2012) and found that these two kinase domains in *Pm57* belong to RLK/Pelle_DLSV subfamily. Therefore, *Pm57* has a DLSV-DLSV configuration, similar to *Rpg1*, *Pm24*, and *Sr62* (Yu et al., 2022). Notably, the Arabidopsis DLSV members was later recognized as LRR_8B, G-LPK, etc. subfamilies (Zulawski et al., 2014). This suggests that LRR_8B falls within the DLSV subfamily. To further classify the two protein kinase domains of *Pm57*, we used the 182 protein kinase domains discovered by Klymiuk et al. (2018) to construct a neighbor-joining (NJ) phylogenetic tree. As shown in Supplementary Fig. 11, the two protein kinase domains in *Pm57* were assigned to LRR_8B subfamily (see Lines 293-302).

Bootstrap values have also been added in the phylogenetic tree.

Lehti-Shiu, M.D. & Shiu, S.H. Diversity, classification and function of the plant

protein kinase superfamily. *Philos Trans R Soc Lond B Biol Sci* 367, 2619-39 (2012).

Yu, G. et al. *Aegilops sharonensis* genome-assisted identification of stem rust resistance gene *Sr62*. *Nature Communications* 13, 1607 (2022).

Zulawski, M., Schulze, G., Braginets, R., Hartmann, S. & Schulze, W.X. The *Arabidopsis* Kinome: phylogeny and evolutionary insights into functional diversification. *BMC Genomics* 15, 548 (2014).

Klymiuk, V. et al. Cloning of the wheat *Yr15* resistance gene sheds light on the plant tandem kinase-pseudokinase family. *Nature Communications* 9, 3735 (2018).

Major concern 3. Is *Pm57* the ortholog of *Lr9*? This relationship was unclear and the phylogenetic analysis suggests they are highly related.

>Our response: Current orthology inference algorithms are generally based on sequence similarity, domain architecture, collinearity, and reciprocal best hit (Chen et al., 2022 and Zieleszinski et al., 2017). *Pm57* had a high similarity of 88.3% in amino acid sequences with *Lr9* and they had the same kind of domains (Sup Fig. 4 and 5). The adjacent genomic region of *Pm57* was syntenic to the *Lr9* genomic regions in the *Ae. umbellulata* (Sup Fig. 10 and Sup Table 6). Reciprocal BLAST showed that *Lr9* and *Pm57* are each other's best BLAST hits. (Sup Table 7). Therefore, analyses of sequence similarity, gene collinearity and "reciprocal best hits" inferred *Pm57* is the ortholog of *Lr9*. We have added these results to the results section of the text (see Lines 278-286).

Chen, Y. et al. A collinearity-incorporating homology inference strategy for connecting emerging assemblies in the Triticeae tribe as a pilot practice in the plant pangenomic era. *Mol Plant* 13, 1694-1708 (2020).

Zieleszinski, A. et al. ORCAN-a web-based meta-server for real-time detection and functional annotation of orthologs. *Bioinformatics* 33, 1224-1226 (2017).

Minor concerns

1. Results, line 96. "on genome sequence" to "on the genome sequence".

>Our response: We have corrected it (see Line 109).

2. Figure 1. Panel C should be relative to the genomic distribution, not the cartoon version.

>Our response: We have corrected it.

3. Line 114. Was RNAseq also performed on the parent of the mutagenesis population?

>Our response: Yes, RNA-seq was also performed on the parental line 89(5)69. We have explained this in detail in the text (see Line 141).

4. Figure 4. Individual domains should be labelled in panel B.

>Our response: We have labelled individual domains as suggested.

5. References, Wang et al. is now published. <https://doi.org/10.1038/s41588-023-01401-2>

>Our response: We have corrected it (see Line 735).

6. Results, it should be mentioned in the results that the ubiquitin promoter was used to drive Pm57 expression to generate transgenics.

>Our response: We have added it as suggested (see line 189).

7. Figure 6. While I appreciate that the experimental plan in Figure 6 was to determine if Pm57 had an negative agronomic characteristics, it was unclear why the various introgression lines were not used in field experiments in parallel. It would be useful to see the contrast between introgression lines and transgenic lines.

>Our response: We added the agronomic characteristic data of two small-fragment introgression lines in Figure 6. Specifically, they are translocation lines Type IV-1 and Type IV-2 (in Fig. 1b), which contains 5.57-9.12 Mb chromosome fragments from *Ae. Searsii*. Compared to the control line TA3809, the small-fragment translocation lines showed no significant differences in the investigated traits.

8. There was no evaluation of natural variation in Pm57 in *Aegilops searsii* (and also related species). There appear to be four data sets of *Aegilops searsii* RNAseq in NCBI that could be investigated.

>Our response: We downloaded RNA-seq raw reads of the four data sets of *Aegilops searsii* (KU-14651, KU-6143, KU-6142, and KU-5755) from the NCBI sequence read archive (SRA) database. *Pm57* cannot be mapped to any reads in these four *Ae. searsii* RNA-seq data sets, suggesting the functional *Pm57* allele may be not present or not expressed in these accessions. Since we do not have these *Ae. searsii* accessions, further confirmation was not performed and the above analysis results were not added in the manuscript.

Reviewer #2 (Remarks to the Author):

In this manuscript, Zhao and colleagues report the cloning of the powdery mildew resistance gene Pm57 from bread wheat. Pm57 has been introgressed into bread wheat from the wild wheat relative *Aegilops searsii*. Using map-based cloning, EMS mutagenesis, virus-induced gene silencing, and transgenic complementation, the authors identify a wheat tandem kinase (WTK) protein fused to a vWA domain as Pm57. This is only the second example of a WTK-vWA encoding gene involved in disease resistance. In general, the study is well executed, well written, and of general interest to the broad readership of Nature

Communications. I do have some comments that the authors should consider before publication.

We thank you for your comments and positive assessment of the manuscript.

Major comments

- For the transformation, the authors used an overexpressing promoter (*Ubi*) and coding sequence. There are several examples of disease resistance genes that have possibly been wrongly identified because of candidate gene overexpression. A critical experiment is to demonstrate that the overexpressing transgenic lines maintain race-specificity. Are there any powdery mildew isolates that are virulent on *Pm57*? Fig. S2 does not include a wheat line with the endogenous *Pm57* gene, so it is not possible to compare the *Pm57* transcript levels in the transgenic lines to the endogenous *Pm57* expression levels?

>Our response: We are aware that several R genes were wrongly identified because of candidate gene overexpression and lack of relevant analyses using ≥ 5 mutants. Therefore, it will be critical to check whether the overexpressing transgenic lines maintain race-specificity or not. We tested the resistance of *Pm57* introgression lines and transgenic lines to 29 divergent *Bgt* isolates and found that *Pm57* was resistant to all of these *Bgt* isolates (see Sup Table 11). Therefore, until now, we do not find any *Bgt* isolates that are virulent on *Pm57*.

We added the expression level of endogenous *Pm57* gene in Sup Fig. 3b, and found that the *Pm57* transcript levels in some transgenic lines are higher than or similar with or even lower than endogenous *Pm57* expression level. With the exception of L20 and L29 which lack the expression of *Pm57*, all the positive transgenic lines showed a disease-resistant phenotype compared with the control. These results suggest that the *Pm57* candidate gene should have not been wrongly identified in this study because of using an overexpressing promoter (*Ubi*).

- The authors state in the abstract that *Pm57* transgenic lines had 'no adverse effects on agronomic traits'. This is quite a strong statement. The data shown in figure 6 are based on relatively few plants grown in single rows. The authors did not perform proper yield trials (macro-plots) in different environments to test for agronomic performance. This is not required here, but I suggest that the authors are a bit more conservative in their wording. It is still possible that *Pm57* has a minor yield penalty that could not be assessed with the current experimental set-up.

>Our response: We have toned down our conclusion. Now we state that no significant adverse agronomic traits were observed in our current experimental set-up. Meanwhile we have added more information about the measurement of related traits in the Materials and methods section (see Lines 622-626), and

added agronomic traits data of small-fragment translocation lines in the Results section (see Lines 327-329). We think that additional data will further support our conclusions.

- The discussion mainly repeats the main findings of the manuscript. For me, the most exciting conclusion of this manuscript is the fact that orthologous WTK-vWA genes (*Pm57* and *Lr9*) from different *Aegilops* species confer resistance against different pathogens. This could be highlighted and discussed more prominently (maybe even in the abstract) and in more detail. Did the authors overlap the predicted protein structures of *Pm57* (Fig. 4b) and *Lr9*? Are there any structural differences that could explain the difference in recognition specificity between *Pm57* and *Lr9*? The authors provide an amino acid alignment of *Pm57* and *Lr9* (Fig. S11), showing that most differences are in the pseudokinase and vWA domains. Are these amino acid polymorphisms mainly affecting surface-localized residues that might bind a putative effector? In addition, a Ka/Ks analysis could be done to test if the two kinase domains are under opposing selective pressure?

>Our response: In the Results, we proved that *Pm57* and *Lr9* were homologous genes based on sequence similarity, synteny and "reciprocal best hits" analyses (see Lines 278-286). We have added the statement that *Pm57* is an ortholog of *Lr9* to the Abstract (see Lines 26-27). In addition, according to the suggestions of reviewer, we analyzed the overlap of different structural domains of *Pm57* and *Lr9*, and found that the difference was mainly located in the disordered regions of pseudokinase domain and vWA domain (see Sup Fig. 17). Further, we analyzed protein disorder rates, and found that the disorder rates of *Pm57* in the pseudokinase domain and vWA domain regions were higher than those of *Lr9* (see Sup Fig. 18). 5, 84 and 25 amino acids had polymorphisms between *Pm57* and *Lr9* in kinase, pseudokinase, and vWA domains, respectively. Among them, about half amino acids with polymorphisms in each domains were predicted to be surface-localized, which are likely to interfere with protein function. Meanwhile, the other half amino acids with polymorphisms in each domains were internal residues that might affect protein structure and stability. In addition, Ka/Ks analysis showed that the two kinase domains of *Pm57* and *Lr9* were subjected to different degrees of selective pressure (see Sup Table 8). These analyses are presented in the Results section in lines 278-292 and Discussion section in lines 396-409.

Minor comments

- Title: Not sure if the word 'novel' is appropriate in the title.

>Our response: We have deleted 'novel' from the title.

- Abstract: It is unclear to me how a 'poor understanding of the resistance mechanisms' has constrained the use of wild relatives for wheat improvement.

I would argue that the main constraints are difficulty to make crosses and linkage drag.

>Our response: We agree that this point was not clear. We have modified it in the Abstract (see Lines 18-20).

• Line 34: The global wheat loss estimates due to powdery mildew are in the range of 1% (Supplementary Table 3 of Savary et al. 2019 Nature Ecology & Evolution 3: 430-439). The 10-15% indicated here seem to be overestimated.

>Our response: We have corrected it and cited this reference (see Lines 35-36).

• Line 63: 'unavailability of annotated reference genomes.'

>Our response: We have corrected it (see Line 63).

• Line 71-73: Please indicate how *Pm57* was introgressed into bread wheat. Was this done through homologous recombination? Was irradiation required? This is mentioned later and in previous publications, but I think it is important to repeat this here. Was it confirmed that the introgression is indeed homologous?

>Our response: *Pm57* was introgressed into bread wheat through homologous recombination using the *ph1b* mutation. Irradiation was not used. We have further elaborated this in the text (see Lines 72-79). Cytogenetic analysis of the translocation lines (Liu et al., 2017) and comparative synteny in *Pm57* candidate region (Dong et al., 2020) revealed that the introgression is indeed homologous.

Liu, W. et al. Homoeologous recombination-based transfer and molecular cytogenetic mapping of powdery mildew-resistant gene *Pm57* from *Aegilops searsii* into wheat. Theoretical and Applied Genetics 130, 841-848 (2017).

Dong, Z. et al. Physical mapping of *Pm57*, a powdery mildew resistance gene derived from *Aegilops searsii*. Int J Mol Sci 21, 322 (2020).

• Lines 83-84: Was *Pm57* mapped in *Ae. searsii* or in a bread wheat introgression line? In other words, was the gene mapped to the long arm of chromosome 2SS or to an *Ae. searsii* chromosome segment located on bread wheat chromosome arm 2B? This is a fine but important detail. The wording here suggests mapping in *Ae. searsii*, but the experiments were done in a bread wheat introgression line. Are the authors sure that the introgression occurred at a homologous position? For Lr9 / Lr58, for example, the translocation from the wild relative was not homologous, which caused confusion during the mapping.

>Our response: We have improved this part to make it clearly described. The *Pm57* mapping populations (F_2) were developed from a cross between a bread wheat introgression line 89(6)88 and CS *ph1b* mutation TA3809. 89(6)88 is a powdery mildew resistant translocation line (Ti2AS-2S^sS.2S^sL-2AL), in which a large section of wheat chromosome 2A was replaced by homoeologous

segments derived from 2S^s. In previous studies, we found chromosome 2S^s of *Ae. searsii* is homoeologous to wheat group 2 chromosomes (Dong et al., 2020) and our translocation lines 89(6)88 and 89(5)69 were obtained by homoeologous recombination (Liu et al., 2017). Therefore, we are sure the introgression occurred at a homoeologous position. We have elaborated this in the text (see Lines 89-97).

Different from *ph1b*-induced homoeologous recombination, the translocations between wheat and alien chromosomes induced by irradiation can occur at different chromosomes or positions of a chromosome. *Lr9* was introgressed into bread wheat using irradiation. The resulting bread wheat line 'Transfer' carried an *Ae. umbellulata* segment translocating to the end of chromosome 6BL. However, the *Lr9*-containing *Ae. umbellulata* segment is homoeologous to the bread wheat group 2 genomes (Wang et al., 2023), which caused confusion during the mapping.

Dong, Z. et al. Physical mapping of *Pm57*, a powdery mildew resistance gene derived from *Aegilops searsii*. *Int J Mol Sci* 21, 322 (2020).

Liu, W. et al. Homoeologous recombination-based transfer and molecular cytogenetic mapping of powdery mildew-resistant gene *Pm57* from *Aegilops searsii* into wheat. *Theoretical and Applied Genetics* 130, 841-848 (2017).

Wang, Y. et al. An unusual tandem kinase fusion protein confers leaf rust resistance in wheat. *Nat Genet* 55, 914-920 (2023).

• Line 85: 'were used to develop a large'
>Our response: We have revised this sentence.

• Line 112: The statement 'were identified by Pm57-specific markers and a 55K SNP chip' is not clear. I assume the mutants were identified because they were susceptible. Does this sentence indicate that the genetic integrity of susceptible putative mutants were validated using markers and a 55K SNP chip? If so, the data should be provided as supplementary material. How many susceptible putative mutants were discarded because they showed a different genotype with the Pm57-specific markers and the SNP array? This needs more explanation here (it is mentioned in the methods section, lines 384-386, but I suggest moving this part to the results section).

>Our response: Indeed, the mutants were identified because they were susceptible. In order to eliminate the susceptibility caused by seed contamination or missing 2S^s, the 15 susceptible mutants were also verified using 2S^s-specific molecular markers X67593 and X62492 flanking *Pm57*. We confirmed that these 15 susceptible mutants are all real mutants. So no susceptible putative mutants were discarded. SNP chip verification of mutants is unnecessary. Therefore, we removed the statement about 55 K SNP chip. In addition, we moved the description of verifying mutants with *Pm57*-specific

markers from the Materials and methods section to the Results section (see Lines 135-139).

- Please provide representative images of all 15 mutants in the supplementary material. In particular, was there any phenotypic difference between the 10 Pm57 mutants and the 5 putative second-site mutants?

>Our response: We have provided representative images of all 15 mutants in Sup Fig. 1. No obvious phenotypic difference was found between the *Pm57* mutants and the putative second-site mutants.

- Line 437: 'In each plot, 5 plants were chosen to measure various agronomic traits.' This does not match with the numbers provided in Fig. 6 ($n = 3$, or $n = 15$).

>Our response: We have improved this part to make it clearly described. The transgenic lines and control lines were planted with three replicates. In each replicate, five representative plants were chosen to measure agronomic traits. Therefore, $n = 3 \times 5 = 15$ plants were investigated to measure plant height, spike length, and spikelets per spike in each line. However, heading date, thousand-grain weight, and seeds per spike are recorded on average measurement value of the five plants for each replicate, therefore $n = 3$. To make it clear, we made the following modification in Fig. 6 legend (see Lines 338-340). 'Heading date, one thousand-grain weight, and seeds per spike are represented as mean \pm SD from three replicates. Data in plant height, spike length, and spikelets per spike are represented as mean \pm SD from individual plants ($n = 15$)'.

- Please include a scale bar in Fig. S1

>Our response: We have added it.

- The resolution of Fig. S5 needs to be improved.

>Our response: We have improved it.

Reviewer #3 (Remarks to the Author):

Zhao et al. reported cloning Pm57, the first gene isolated from *Aegilops searsii*. It encodes a wheat tandem kinase (WTK) protein with putative kinase-pseudokinase domains followed by a von Willebrand factor A (vWA) domain, designated WTK7-vWA. The experimental approach and the described results are solid. Independent mutants, gene silencing, and transgenic assays validated Pm57. These findings highlight the emerging role of kinase fusion proteins in plant disease resistance and provide a powerful gene for wheat breeding. However, the writing should be improved, the English is not good, and the interpretations of results should be improved. Some parts of the

introduction and discussion should be improved, focusing on the main topic and going more in-depth.

We thank you for your comments and positive assessment of the manuscript.

Another wheat tandem kinase with a vWA domain, designated WTK6-vWA, was recently cloned. However, the relationships between these two genes are poorly described. The authors mostly ignored WTK6-vWA and only mentioned it in the discussion part towards the end of the manuscript. WTK6-vWA and WTK7-vWA show 88.3% homology but confer resistance to two diseases: leaf rust and powdery mildew. A similar case was recently described for WTK3 and RWT4, which show high sequence similarity yet confer resistance to different diseases, such as powdery mildew and wheat blast. WTK3 and RWT4 share the same chromosome location. Therefore, they can be considered alternative alleles of the same gene with different functional specificities. We are wondering whether similar relationships exist in the case of WTK6-vWA and WTK7-vWA. According to Wang et al. (2023), the WTK6-vWA -containing *Ae. umbellulata* segment is homoeologous to the bread wheat chromosomes 2 (See Peer review file: https://static-content.springer.com/esm/art%3A10.1038/s41588-023-01401-2/MediaObjects/41588_2023_1401_MOESM3_ESM.pdf). Since chromosome 2S of *Ae. searsii* is also homoeologous to wheat chromosomes 2 (Dong et al. *Int J Mol Sci.* 2020), then WTK6-vWA and WTK7-vWA probably share a common origin and should be considered orthologs or homoeologs. Thus, we propose to describe WTK6-vWA already in the introduction of the current manuscript, compare the evolutionary relationships between WTK6-vWA and WTK7-vWA in the results, provide evidence that they are orthologs, properly discuss these relationships, and compare to similar cases mentioned in the literature, such as WTK3 and RWT4.

If WTK6-vWA and WTK7-vWA are orthologs, maybe WTK7-vWA (Pm57) should be named WTK6-vWAPm. Please note that WTK7 was already claimed by Zhiyong Liu's group for the wheat tandem kinase gene WTK7-TM in a paper that is now under revision in *Nature Communications*.

>Our response: Dear Tzion and Yinghui, thanks a lot for your valuable and constructive comments. We have revised the article according to your suggestions. We have stated in the abstract that Pm57 is an ortholog of Lr9 and named it WTK6-vWAPm (see Lines 26-27). In the introduction Pm57 is described as the second protein with a WTK-vWA structure following Lr9 (see Lines 81-82). In the Results, we showed that Pm57 is an ortholog of Lr9 and compared their evolutionary relationships (see Lines 278-302). We discussed the findings that orthologous Pm57 and Lr9 conferred resistance to two different pathogens, included WTK3 and RWT4 as an example, and highlighted the importance of these TKPs in the wheat resistance breeding (see Lines 396-414).

Questions/suggestions

1. Lines 51-52,

Please consider combining alleles/homologs of the same genes, such as Pm21/Pm12, Pm60/MIIW172/MIIWE8, etc., or provide a more detailed explanation for the possible homologous/orthologous/allelic relationships between similar genes.

>Our response: We have combined alleles/homologs of same genes (see Lines 50-52).

2. Lines 53-54,

Please carefully check all references since we noticed a problem with the numbering of some of the references. For example, Pm4 is described in reference 19, not in reference 3. Furthermore, the reference for WTK4 is missing.

>Our response: We have checked all references and made related corrections.

3. Lines 66-68,

This sentence is not clear. Please improve. Maybe you want to change "thus" to "as well as".

>Our response: We have corrected it (see Line 67).

4. Lines 74-75.

WTK7- vWA is the second gene discovered with this structure. Therefore, the terms "novel" and "unusual" should not be used in association with its structure.

>Our response: We have removed the terms "novel" and "unusual" (see Line 80).

5. Line 78

I do not see some important molecular mechanism results here. Please change this statement to fit the results.

>Our response: We have revised this sentence (see Lines 84-86). 'Our results provide a valuable resistance gene for further understanding the molecular mechanisms underlying wheat powdery mildew resistance and will enable the development of wheat varieties with broad-spectrum powdery mildew resistance.'

6. Line 84

Please provide more clear information on wheat-*Ae. searsii* 2S.

>Our response: We have provided more clear information on wheat-*Ae. searsii* 2S (see Lines 89-97).

7. Line 87

Please provide a full name for Chinese Spring (CS) when mentioned for the first time.

>Our response: We have corrected it (see Line 89).

8. Line 88-89

Improve English.

>Our response: We have revised this sentence (see Lines 100-102). 'Following this, *Bgt* response assays were conducted on the progenies of these 104 recombinants by inoculating *Bgt* isolate E09 at seedling stage in growth chambers.'

9. Fig. 1b.

Please explain what "Type I – Type VI" is. Add spaces between marker names (X1, X2, etc.) in the map, or use a smaller font.

>Our response: We have explained "Type I – Type VI" in the legend of Fig. 1b and added spaces between marker names (X1, X2, etc.) in the map.

10. Could you give a clear explanation why there were only three of the five mutants that had the missense mutation of gene G4?

>Our response: Because our previous RNA-seq analyses did not utilize the *Ae. searsii* TE01 reference genome sequence, the SNP analysis results were inaccurate. Recently, we re-analyzed the RNA-seq data and obtained new SNP information. The method of data analysis is detailed in the Materials and Methods section (see Lines 505-516). The relevant results have been modified in the text (see Lines 141-146).

11. Figure 2b

Mut216 G6587A? Something is wrong here.

In lines 128-129, the title of Fig. 2b is confusing and does not reflect the content of the figure. We propose to mention the isoforms in the title.

>Our response: G6587A refers to that the nucleotide base G at 6587 to mutated to base A in Mut216. Specifically, Mut216 was a frameshift mutation with a G/A point mutation in the splice acceptor site of intron 9. We added annotations to the Fig. 2b and explained them in the legend (see Lines 157-158, "c." for a coding DNA sequence, "p." for a protein sequence). In addition, the title of Fig. 2b has also been modified, and the isoforms were mentioned in the title (see Lines 154-155), '(b) Schematic representation of G4 alternative transcript variants and identification of G4 using EMS-induced mutants.'

12. Lines 185-188

The sentence is too long and not well written. Please rephrase.

>Our response: We have rephrased this sentence (see Lines 220-224). 'The tertiary structure revealed that Pm57 possesses a modular structure, with kinase-pseudokinase domains, a vWA domain, and a putative Vwaint domain as described by Wang et al.³⁹ (Fig. 4b). We also observed that kinase domains in Pm57 were highly symmetrical like WTK4 and Lr9 but not similar with the other five reported TPKs (Fig. 4b and Supplementary Fig. 5).'

13. Please add information on the promoter, CDS, or genomic DNA in the main text.

>Our response: We have added information on the CDS and genomic DNA in the main text (see Lines 165-167). 'In the resistant line 89(5)69, the G4 gene was 9,473 bp and contained 14 exons with a coding sequence of 3,489 bp (Fig. 2b), and the sequence of G4 in 89(6)69 was identical to that in reference genome TE01.'

14. Line 197 and title of Fig 5.

These titles do not reflect the content of the paragraph and the figure. These experiments describe microscopic observations associated with plant-pathogen interactions and not studies of molecular mechanisms. We propose changing the titles to "histopathological characterizations of..." to reflect the content.

>Our response: We have corrected it (see Line 234 and 259).

15. Figure 5.

CS also showed ROS accumulation. We propose to show quantitative differences between resistant and susceptible accessions. You can see an example in the following publication: "Intracellular reactive oxygen species (intraROS)-aided localized cell death contributing to immune responses against wheat powdery mildew pathogen." <https://apsjournals.apsnet.org/doi/10.1094/PHYTO-07-22-0271-FI>.

>Our response: We agree with the reviewer that CS also showed the accumulation of ROS. Therefore, we re-conducted ROS and cell death experiments, and carried out quantitative evaluation according to the methods described in the above literature. We found that about 5% of the *Bgt*-infected cells were detected with intracellular ROS (intraROS) accumulation in CS from 24 hpi, in contrast, Pm57 caused intraROS accumulation in more than 40% of the *Bgt*-infected cells (Fig. 5a and 5b). Quantitative assessment of cell death at 48 hpi revealed that 37.7% of the *Bgt*-infected cells showed cell death in 89(5)69, while only 3.3% cell death was observed in CS (Fig. 5c). This suggests that Pm57 does enhance intraROS and cell death responses.

16. Lines 206-207.

Fig 5C shows the down-regulation of Pm57 by Bgt at 48 hpi. Please refer to this result in the text.

>Our response: We have added relevant descriptions in the text (see Lines 250-252). 'Upon pathogen infection, the expression of *Pm57* showed a 4-fold up-regulation and peaked at 12 hpi with *Bgt* isolate E09 in the seedling leaves of 89(5)69, and slightly declined at 48 hpi (Supplementary Fig. 6b)'. The graph has been moved to Sup Fig. 6b.

17. 210-211.

The conclusion is wrong since the PR genes are also upregulated in CS after Bgt infection.

>Our response: We have revised the conclusion (see Lines 256-257, 'These expression patterns of *Pm57* and *PR* genes imply that powdery mildew resistance of *Pm57* may involve *PR* genes up-regulated expression.'). In addition, we used transgenic lines to analyze the expression of *PR* genes, and found that the expression of *PR* genes in transgenic lines after *Bgt* infection was higher than that of recipient control. This is consistent with previous results obtained using translocation lines, so we replaced previous results with new ones. The graph has been moved to Sup Fig. 7.

18. Line 221

The evolutionary study is ignoring completely WTK6-vWA. Therefore, missing the most important issue in TKP convergent evolution. We propose adding WTK6-vWA to the evolutionary analysis. It gives the impression that the authors are trying to hide WTK6-vWA, while this is one of the most exciting findings of this work dealing with homoeologous genes that provide solutions against different pathogens.

>Our response: We have added WTK6-vWA to the evolutionary analysis. Since the WTK6-vWA genome is at contig level and there is no gene annotation file, we performed synteny analysis by sequence alignment. It was found that the adjacent genomic region of *Pm57* was syntenic to the *WTK6-vWA* genomic regions in the *Ae. umbellulata* (Sup Fig. 10 and Sup Table 6). This is an important evidence that *Pm57* is an ortholog of *Lr9* (see Lines 278-286). In addition, we also carried out sequence structure and *Ka/Ks* analysis between *Pm57* and *Lr9* according to the suggestions of other reviewers. The results will be helpful to study the functional differentiation of orthologous genes.

19. Lines 275-280

The paragraph is confusing. First, you say that the *A. searsii* genome is absent and then present. We suggest better describing the story less confusingly.

>Our response: We have deleted this part according to the suggestions of other reviewers.

20. Fig.6.

The seeds seem bigger in transgenic lines. Did you measure seed size? Were these plants infected with Bgt in the field?

>Our response: We measured the seed size of transgenic lines and the control and found no significant difference between them. We also investigated agronomic traits using small fragment translocation lines, and found that *Pm57* gene did not affect these agronomic traits, and seed size did not change significantly. In addition, the experimental fields for agronomic trait investigation were routinely sprayed with pesticides, and the plants were not infected with

Bgt. We have added this information in the Material and methods section (see Lines 625-626).

21. Please provide the Kinase I and II subfamily (e.g., LRR-8B, WAK, etc.) according to rice or Arabidopsis kinome, as previously published for other WTK genes.

>Our response: We added the LRR-8B, WAK, etc. on the evolutionary tree (see Sup Fig. 11).

22. Lines 470-471

The 182 putative kinase or pseudokinase domains used for phylogenetic analysis of WTK3 were discovered by Klymiuk et al. (2018)³³ following the cloning of WTK1 and not by Lu et al. (2020)³. Therefore, we propose to give proper acknowledgment by adding “WTK1 (Yr15)³³ and WTK3 (Pm24)³.....”

>Our response: We have corrected it (see Line 609).

23. Lines 474-475

The conserved motifs in the Kin I and Kin II domains of WTKs were first described by Klymiuk et al. (2018)³³ following the cloning of WTK1 and not by Lu et al. (2020)³. Therefore, we propose giving proper acknowledgment to Klymiuk et al. (2018).

>Our response: We have corrected it (see Line 612).

24. Line 554

Update reference 27; The paper by Wang et al. 2023 is now published in Nature Genetics.

>Our response: We have updated it (see Line 735).

Reviewed by Tzion Fahima and Yinghui Li

Reviewer #4 (Remarks to the Author):

Zhao et al. revealed the cloning and characterization of Pm57, a gene that confers wheat resistance to multiple isolates of powdery mildew. The researchers have dedicated substantial efforts towards fine-mapping a gene derived from a wild wheat source in a unique manner and identifying a novel kinase fusion protein. Furthermore, they have employed several complementary strategies to functionally validate and characterize the gene.

We thank you for your comments and positive assessment of the manuscript.

However, the material and methods section lacks crucial information, making it difficult to assess the execution of the experiment. A more rigorous description of the experiment and data is necessary. For instance, it remains unclear

whether the T2 individuals used for field evaluation were fixed for Pm57, and there is limited information about the protocol employed to assess various agronomic traits. Based on the provided data, it appears that there are certain speculative aspects in the findings. For instance, the claim of a correlation between H₂O₂ accumulation and the inhibition of haustorium formation lacks quantitative data and is not clearly supported by the presented images. Furthermore, saying that Pm57 acts as an upstream regulator is quite speculative as you analysed expression of PR in different genetic backgrounds (CS and 89(5)69). To provide more robust conclusions, it would have been beneficial to include the analysis of expression in transgenic models. The absence of expression data for the VIGS experiment makes it difficult to ascertain whether Pm57 expression was genuinely downregulated. The quality of images is not good which does not facilitate the analysis of symptoms for instance. There are typos. Some references are not the right ones. Could you please rearrange your materials and methods section to follow the results section?

>Our response: We have added more detailed information in the materials and methods section and rearranged this section according to the results section. In the results section, we added a quantitative analysis of ROS accumulation and cell death, analyzed the expression of PR genes in transgenic plants, and supplemented the expression data of target genes in the VIGS experiment. In addition, we have improved the quality of images and carefully corrected text errors and reference errors. As all of the above comments were further mentioned in the reviewer's 'Further comments', we responded to them one-on-one in the 'Further comments'.

It seems that the initial two paragraphs of the discussion section serve almost as a summary of the study, which could be redundant considering the preceding sections. Furthermore, the inclusion of figures from the supplementary section exclusively described in the next paragraphs may be better suited for the results section. In the results section, a dedicated paragraph describing the protein in comparison to other known kinases would provide a more focused and informative presentation of the findings.

>Our response: We modified the discussion part according to the reviewer's suggestion. We reduced the first two paragraphs of the discussion to one paragraph, and have provided a more focused and informative presentation of the protein in comparison to other known kinases in Results section (see lines 214-219).

Further comments:

L27 « powerful » seems inappropriate.

>Our response: We replaced 'powerful' with 'valuable' (see Line 28).

L35 this is not the right reference. there are few important references estimating

the global impact of wheat diseases.

>Our response: We cited a new reference (see Line 36). 'This disease typically leads to yield losses higher than 1% globally which may reach up to 3.27% in China²'.

Savary, S. et al. The global burden of pathogens and pests on major food crops. *Nat Ecol Evol* 3, 430-439 (2019).

L39 I'm not sure that this reference is adapted here.

>Our response: We changed it to two references (see Line 40).

Hafeez, A.N. et al. Creation and judicious application of a wheat resistance gene atlas. *Mol Plant* 14, 1053-1070 (2021).

McIntosh, R.A., Dubcovsky, J., Rogers, W.J., Xia, X.C. & Raupp, W.J. Catalogue of gene symbols for wheat - 2020 Supplement. In: GrainGenes Database (<https://wheat.pw.usda.gov/GG3/wgc>). (2020).

L40 Could you please provide references?

>Our response: We have added four related references to this sentence (see Line 41). '*Pm2*, *Pm4*, *Pm5*, *Pm6*, *Pm8*, *Pm21*, and *Pm52* have been widely deployed disease-resistant wheat varieties⁵⁻⁸'.

Gao, H. et al. Identification of the powdery mildew resistance in Chinese wheat cultivar Heng 4568 and its evaluation in marker-assisted selection. *Front Genet* 13, 819844 (2022).

Xu, H. et al. Molecular tagging of a new broad-spectrum powdery mildew resistance allele *Pm2c* in Chinese wheat landrace Niaomai. *Theor Appl Genet* 128, 2077-84 (2015).

Zhao, Z. et al. Genetic analysis and detection of the gene *MILX99* on chromosome 2BL conferring resistance to powdery mildew in the wheat cultivar Liangxing 99. *Theor Appl Genet* 126, 3081-9 (2013).

Nematollahi, G., Mohler, V., Wenzel, G., Zeller, F.J. & Hsam, S.L.K. Microsatellite mapping of powdery mildew resistance allele *Pm5d* from common wheat line IGV1-455. *Euphytica* 159, 307-313 (2008).

L45 I do not agree with this sentence. With the availability of new genomic resources and tools, I'm not sure that the classical map-based cloning strategy is the most effective as demonstrated by the cloning of numerous R genes using other strategies. Furthermore, in the absence of a sequenced and annotated genome, it is almost impossible to identify candidate genes. In the following sentences, you said that map-based cloning is time-consuming and challenging and that genome sequence facilitates map-based cloning, which is a bit contradictory to your previous sentence. I recommend rewriting this section. Usually, the strategy for cloning dependant on many features such as the origin

of your genes, and its position on the chromosome....

>Our response: We have rewritten this paragraph to highlight new genomic resources and tools to facilitate wheat gene cloning (see Lines 46-49).

L85 Which recombinants?

>Our response: The 'recombinants' refer to 'nine heterozygous recombinants'. We have rewritten this sentence (see Line 97) and listed the nine individuals in the Materials and Methods section (see Lines 480-482).

L87 Which kind of individuals, provide the generation

>Our response: We have added F₃ individuals in Line 99.

L98 it would be great here to know how those 12 genes are conserved on the susceptible parent CS.

>Our response: We have added the description of these 12 genes in CS (see Lines 112-114). 'Ten of the 12 genes were conserved and shared more than 80% amino acid sequence identity with corresponding genes on group 2 chromosomes of common wheat CS, whereas G4 and G5 genes were absent in CS (Supplementary Table 2)'.

L110 you treated M0 seeds and end up with M2 families, Are those M2 obtained directly from the M0 plants? If yes they should be M1?

>Our response: We added this information in Line 134. 'M₁ seeds were harvested from about 1,598 surviving M₀ plants.'

L111 In the material section it is mentioned 10-15 plants

>Our response: We germinated 15 seeds for each M₁ family, but we removed the seeds that did not germinate or germinated slowly, so for some families, the number of susceptible plants screened did not reach 15. Therefore, we changed '15 seeds' to '10-15 plants' (see Line 134).

L112 what do you mean by independent mutants? Do they originate from different M2 families?

>Our response: Independent mutants refer to mutants originating from different M₀ plants, also as different M₁ family. We added this information in Line 138.

L113 I think the sentence should be modified as the susceptible were not identified by Pm57 markers, they were identified after phenotyping with isolate Bg09. What was further validated in M3 (and how many plants in M3).

>Our response: In order to eliminate the susceptibility caused by seed contamination or missing 2S^s, the susceptible were identified by Pm57 markers after phenotyping with isolate E09. We have revised this sentence (see Lines 135-139). 10-15 plants were further validated.

L116 I supposed that the RNA-Seq data were aligned against the Pm57 candidate genes and not the opposite. Please correct.

>Our response: We have corrected it (see Line 142).

L136 sequenced instead of cloned?

>Our response: We have corrected it (see Line 163).

L139 Could you please mention how the sequence of this gene was different between TE01 and 89(6)69.

>Our response: We have added relevant descriptions in the text (see Lines 166-167). 'the sequence of G4 in 89(6)69 was identical to that in reference genome TE01.'

L140 There is no material and method section for this result !

>Our response: We added alternative splicing variants analysis in the Materials and methods section (see Lines 527-530).

L157 The plants were not as fully susceptible as the susceptible control CS. Furthermore, the absence of expression data hinders our ability to draw conclusions regarding the role of G4 expression in resistance.

>Our response: We added expression data for the VIGS experiment (Sup Fig. 2) and found that the transcript levels of G4 in G4-VIGS plants were significantly lower than that in BSMV- γ control plants. In addition, we noted that the transcript level of G4 in the less susceptible individual was higher than that in the other two highly susceptible individuals. Therefore, we suspect that the expression of G4 gene may be related to resistance.

L161 Indicate what part of G4 was used for transformation.

>Our response: We have indicated in the text that the full-length CDS of G4 (isoform IF1) was used for transformation (see Line 188).

L162 Quality of images from figure 3a are not really good but it seems that pustules are also present in other transgenic families like L3, L9, L18? If this is right how do you explain this observation?

>Our response: We have improved the quality of Fig. 3a to make it as clear as possible, while providing the original image in the Source data file. As the reviewer observed, transgenic lines L3, L9 and L18 showed a small number of pustules, but they were all disease-resistant types compared to susceptible controls. We believe that this is mainly due to G4 expression levels; the expression levels of G4 in L9 and L18 were lower than endogenous G4 expression levels in introgression line 89(5)69 (see Sup Fig. 3b). In addition, due to the inconsistent growth of the primary transgenic seedlings (T_0), the growth status of the detached leaves may slightly affect the results of powdery mildew resistance evaluation.

L167 Are the T1 named L1, L2... progenies of the T0 named L1, L2 respectively? I'm not sure to understand why T1 L3 is not shown in Figure 3 and Figure S2? as it has a strange phenotype in T0 it would have been great to understand why.
>Our response: As the reviewer commented, the name of lines in T₁ corresponds to that of T₀. At that time, the L3 T₀ plant was very weak, and it was harvested late with a small number of T₁ seeds compared with other transgenic plants. Therefore, L3 was excluded for disease resistance evaluation and for the expression analysis, while most of other T₁ transgenic lines were assessed for disease resistance. We are going to conduct the disease resistance evaluation and expression analysis of L3 when L3 propagates enough seeds for the conduction.

L168 it is not explained why the expression of G4 was studied only in some T0 and not all. It would have been great to see if there is a correlation between the level of expression and the phenotypes as some T0 show few pustules despite the presence of G4.

>Our response: At that time, we mainly focused on disease resistance assessments of transgenic plants and did not consider the influence of G4 expression levels on disease resistance. Therefore, we just determined the expression levels of G4 in some T₀. Recently, we detected the expression levels of G4 gene in 16 T₁ transgenic lines that were evaluated as the first batch for disease resistance, and found that transgenic lines L9 and L18 with a small number of pustules had low expression levels of G4 (see Sup Fig. 3). These results indicated that the expression level of *Pm57* might relate to disease resistance.

L173 which generation

>Our response: We added generation to the title of Fig.3 (see Line 201).

L200 it is not mentioned in the legend of figure 5 that Coomassie blue was used for staining. Could please show on your figure the inhibition of haustorium formation.

>Our response: We have added the statement that Coomassie blue was used for staining in the legend of Fig. 5 (see Lines 263-264). Haustorium (Hau) has been labeled on the Fig. 5.

L204 this is speculation as you do not have any quantitative evaluation of cell death and H₂O₂ and haustorium formation.

>Our response: We re-conducted ROS accumulation and cell death experiments, and quantified ROS accumulation and cell death according to the method of Li et al. (2023) (see Fig. 5).

Li, Y. et al. Intracellular reactive oxygen species-aided localized cell death

contributing to immune responses against wheat powdery mildew pathogen. *Phytopathology* 113, 884-892 (2023).

L206 What mean highly expressed compared to what ?

>Our response: We have revised this sentence (see Line 249). *Pm57* was highly expressed in wheat leaf tissues compared with roots and stems.

L210 this is also speculation as the genetic background you used for the analyses are quite different. We would be able to suggest this if you would have used the transgenics for the expression data.

>Our response: We re-analyzed the expression of *PR* genes using two transgenic lines (L1 and L16) and control (Fielder). (see Sup Fig.7).

L231 here you evaluate if the resistance allele if present not if the gene is present.

>Our response: We have corrected it (see Line 303).

L238 I guess there are few wheat cultivars as we do not have this information and most of them are from China so you cannot say that *Pm57* has not been used in modern wheat breeding, possibly in China but what about breeding programs in other part of the world?

>Our response: We have removed the description that '*Pm57* has not been used in modern wheat breeding'.

L239 Please provide the exact PCR condition used for the diagnostic marker.

>Our response: We have added the description of annealing temperature (see Line 303, 'an STS-*Pm57* marker with 57°C annealing temperature were designed to amplify a 530 bp genomic sequence of the *Pm57* gene') and provided PCR conditions in the Materials and methods section (see Lines 488-490).

L255 which recombinants?

>Our response: We added the name of the recombinant (see Line 320). This recombinant (88R-3-19-1) is derived from our previous research, which we describe in detail in the Materials and methods section (see Line 436).

L353 Provide the type of population.

>Our response: They were F₂ and F₃ segregating populations (see Line 429). We have rewritten the Plant materials section and provided a detailed description for all the plant materials (see Lines 426-444).

L359 Provide the number of individuals evaluated for each type of experiment, how many F₃..

>Our response: We have rewritten this paragraph and explained in detail the

number of plants used in each experiment (see Lines 457-477). 10-15 seedlings of EMS-mutagenized M₂ plants and *Pm57* T₁ transgenic plants were used for resistance evaluation. For evaluation of *Pm57* resistance spectrum, four seedlings for each of three replicates were used for the *Bgt* responsive assays of 29 *Bgt* isolates. To investigate the potential influence of adverse genetic backgrounds on *Pm57* resistance, F₁ hybrids derived from the crosses of *Pm57* introgression line 88R-3-19-1 with 22 wheat elite varieties were used for the *Bgt* E09 responsive assay, with each hybrid had three biological replicates and four seedlings per replicate.

L361 Which leave was inoculated.

>Our response: The just fully unfolded first leaves were inoculated (see Line 449).

L374 I guess the genome of CS was used as well?

>Our response: Yes, we have added it (see Line 485).

L388 Which part of the plant was collected? How many plants? Which generation? Which time point? Please provide details about the sample.

>Our response: We have described in detail the preparation of RNA-seq samples (see Lines 498-501). Five susceptible M₂ mutants (Mut51, Mut60, Mut141, Mut209 and Mut216) derived from independent M₁ families as well as resistant wild type parental line 89(5)69 were selected for RNA-seq. The first leaves from five plants of each genotype were sampled for RNA extraction at 5 dpi with *Bgt*.

L392 please provide the numbers of reads per sample and the parameters used for alignment.

>Our response: We have provided the numbers of reads per sample and the parameters used for alignment (see Lines 503-508). '39.0, 58.8, 37.8, 44.6, 52.4 and 46.1 million 150 bp pair-ended clean reads were produced for 89(5)69, Mut51, Mut60, Mut141, Mut209 and Mut216, respectively. The clean reads were mapped to the *Ae. searsii* genome assembly (<https://ngdc.cncb.ac.cn/gwh/Assembly/24532/show>) using HISAT2 (version 2.0.5, default parameters). The alignment results were further processed into BAM format using SAMtools (version 1.9) with default parameters.'

L400 low similarities, indicate a threshold. Which regions were targeted for VIGS? In which conditions were the plants grown for VIGS

>Our response: We have provided detailed information about the VIGS experiment (see Lines 532-537). The 3' end of G4 and G5 fragments of 200-250 bp with low similarities (identity < 50% and the stretch of 100% nucleotide identity < 21-nt) with other genes were selected as targets. Seedlings were grown in a growth chamber under 16-h light (24 °C)/8-h dark (18 °C) and

approximately 70% relative humidity.

L409 provide more information in Sup table 1 about the usage of primers and for instance which one was used to clone the cDNA.

>Our response: We have provided more information about used primers in Sup table 1, including the purpose of each pair of primers.

L424 coding sequence from which accession?

>Our response: We have added it (see Line 548). The full-length coding sequence of G4 from 89(5)69 was inserted into pWMB110 vectors.

L429 Which primers and conditions? At which stages leaves were sampled for RNA extraction? How many plants replicates? how cDNA were obtained... too many missing informations.

>Our response: We have described in detail the analysis of G4 gene expression in transgenic wheat lines (see Lines 553-557). 'qRT-PCR analysis was performed to evaluate the expression levels of G4 in the first leaves of transgenic wheat plants in the T₁ generation. The first leaves of four seedlings at the two-leaf stage were sampled for each line to extract RNA. RNA extraction, cDNA synthesis, and qRT-PCR analysis were performed as above. All the reactions were performed in triplicate.'

L436 What is a regular field management? Please provide detail.

>Our response: We have changed the description of 'regular field management' to 'Field managements including irrigation, fertilization, herbicide and pesticide applications strictly followed local practice.' (see Lines 625-626).

L438 How many spikes were evaluated for each of the five plants. Please provide more detail about how the different traits were phenotyped.

>Our response: Only the main spike from each of five plants was evaluated for spike length, spikelets per spike and seeds per spike. We have provided more detail about how the different traits were phenotyped (see Lines 627-632).

L441 how the protoplasts were obtained ? how was done image acquisition? How many replicates? How the transformation was done, which plasmids concentration, provide reference for the AtPif4 constructs.....

>Our response: We have provided detailed information in Subcellular localization analysis section (see Line 559). The constructs were delivered into wheat protoplasts via PEG-mediated transformation according to a prior study (Luo et al., 2022). Under an induction of 40% PEG-4000, control or recombinant plasmids (10 µg per construct) were co-transformed into wheat protoplasts with nucleus marker plasmid AtPIF4-mCherry (Huq et al., 2002). The transformed protoplasts were cultured at 25°C for 16 h under dark conditions, and observed using a laser confocal microscope (A1F, Nikon, Tokyo, Japan). Image

acquisition was conducted with the NIS-Elements Viewer Imaging Software (version 5.21.00). The subcellular localization of Pm57 protein was determined three times.

Luo, G., Li, B. & Gao, C. Protoplast isolation and transfection in wheat. *Methods Mol Biol* 2464, 131-141 (2022).

Huq, E. & Quail, P.H. PIF4, a phytochrome-interacting bHLH factor, functions as a negative regulator of phytochrome B signaling in Arabidopsis. *EMBO J* 21, 2441-50 (2002).

L472 which filters/thresholds were applied?

>Our response: We have added filters parameter information (see Lines 606-607). All the retrieved proteins were scanned by website pfam 35.0 (<http://pfam.xfam.org/>) in batch mode with an E value of 0.01, and the proteins with kinase domain and vWA domain were selected.

Figure 1

Explain what is 89(6)88. Provide the number of recombinants for each type. b) Is it a genetic or physical map as distances are indicated in Mb but you said genetic map in the legend? Provide a scale. If this is a genetic map you should expect to have recombination between each marker as none of them co-segregate, which is not the case according to your type of recombinant. Please explain. c) please spell out WTK and vWA.

>Our response: 89(6)88 is a CS-*Ae. searsii* recombinant line that contains a pair of recombined chromosomes Ti2AS-2S^sS.2S^sL-2AL carrying *Pm57*. We have described it in the legend (see Lines 119-120). It is a physical map, and we have provided a scale and showed the number of the recombinants for each type. The full names of WTK and VWA were also presented in the figure legend.

Figure 3 could you please explain the L4 positive and the L4 negative

>Our response: The L4 positive are the transgenic plants with G4 gene. The L4 negative plants are the individuals without G4 gene segregating in T₁ generation of T₀ transgenic plant L4. We have illustrated this in the figure legend (see Lines 204-205).

Figure 5 L218 in CS as well. The analyses of expression seems to be done differentially between the Pr genes and Pm57. Could you please describe the methods for both analyses. There are significant overexpression for Pm57 but compared to what?

>Our response: We have moved the expression analyses in Figure 5 to Sup figures (Sup Fig. 6 and 7). According to the suggestion of the reviewer, we re-analyzed the expression of *PR* genes in transgenic plants. Detailed analyses methods have been presented in the figure legend of Sup Fig. 6 and 7.

Sup Fig2 What are the replicates? One leave each, two leaves? During one experiment?

>Our response: We have replaced the expression of G4 in T₀ transgenic plants with the expression of G4 in T₁ transgenic lines, and the results are shown in Sup Fig. 3b.

Sup Fig4 reference 29 is not the right one. There is no explanation of RBH, SBH, in last genome? How the small phylogenetic tree was built?

>Our response: We have corrected the reference (Chen et al., 2020). RBH, SBH and so on are described in detail in the figure legend of Sup Fig. 8. RBH: gene pairs belonging to the Reciprocal Best Hits; SBH: gene pairs belonging to the Single-side Best Hits, where RBH is not found but the best matching gene is found; Singleton: the genes with no homologous genes. In last genome: The last genome in the picture. Because the Triticeae-GeneTribe database uses RBH and SBH data structures, where SBH is unidirectional, the last genome cannot be compared with other genomes when drawing from the bottom up. The left tree is obtained from a pre-computed relationship derived from Triticeae-GeneTribe.

Chen, Y. et al. A collinearity-incorporating homology inference strategy for connecting emerging assemblies in the Triticeae tribe as a pilot practice in the plant pangenomic era. *Mol Plant* 13, 1694-1708 (2020).

Sup Fig6 Indicate the size of bands of the ladder or give the exact reference

>Our response: We have indicated the size of bands of the ladder (see Sup Fig. 12).

Sup Fig7 no materials and methods section at all for this study !

>Our response: We have added the methods of this study (Sup Fig. 13) in the Materials and methods section (see Lines 474-477). 'Additionally, the assessments of powdery mildew reactions of T₁ transgenic plants during a whole growth period were conducted through natural transmission of *Bgt* pathogen in the greenhouse. About 12 plants of each T₁ transgenic line were potted. Simultaneously, the highly susceptible CS was inoculated to spread the *Bgt* pathogen in the greenhouse.'

Sup Fig8 Not material and methods section for this experiment, how many F1 were evaluated? Pm57 was not transformed

>Our response: We have added the descriptions of this experiment (Sup Fig. 14) in the Materials and methods section (see Lines 464-467). We also made the modification in Sup Fig. 14 legend. 12 F₁ seedlings, with four seedlings for each of three replicates, were determined for powdery mildew resistance by inoculation with *Bgt* E09.

Sup Table 2 What is the type? Mutant name are different from the text or Figure 2

>Our response: We have removed the 'Type' and corrected the mutant names to make them consistent with the text (see Sup Table 3).

Sup Table 4 How was performed the manual annotation? Which reference for CS? For the annotated gene, do they seem functional or pseudogene? What is the percentage of identity with Pm57

>Our response: Manual gene annotation was performed based on the full-length coding sequence of *Pm57*. The coding sequence of *Pm57* was used as a query for BLASTN analysis in different wheat assemblies. BLAST analysis was performed using WheatOmics 1.0 (<http://202.194.139.32/blast/blast.html>) with default settings. We found that in some wheat genomes, there are highly similar sequences that are matched to *Pm57*, and they are sequentially arranged on chromosomes 2A or 2B, as if separated by introns. Our manual annotation is to concatenate the sequence fragments from these alignments (usually with 100% coverage and more than 75% identity) to get a complete CDS sequence. However, the genes formed by these sequences were not annotated in their corresponding genomes, and no expressed sequences were found in the NCBI Expressed sequence tags (EST) database, so we inferred that these genes were pseudogenes. These manually annotated genes shared more than 80% amino acid sequence identity to Pm57 (see Lines 277-278).

Sup Table 5 Please provide more information about the isolates

>Our response: We have added the collection sites for the first 9 *Bgt* isolates in the current Sup Table 11. The remaining 20 *Bgt* isolates were provided by Prof. Pengtao Ma from Yantai University, and most of their collection locations were not clear, but they were single-spore derived *Bgt* isolates with different virulent spectrums collected from different cities in China. We have added a note below the Table.

Sup Table 6 please provide information in the hexaploid wheat are landraces or cultivars and if these accessions are available in a genetic resource center.

>Our response: We have indicated landraces or cultivars for the hexaploid wheat. As far as we know, common wheat varieties in China are generally kept by breeders and are not collected by genetic resource centers. These materials can be obtained by contacting us or material suppliers.

Reviewers' Comments:

Reviewer #2:

Remarks to the Author:

I thank the authors for performing a thorough revision of their manuscript, which has greatly improved. It is a pity that they could not find a virulent powdery mildew isolate to be tested on the transgenic lines. Nevertheless, I think that the various experiments (11 independent EMS mutants, virus-induced gene silencing, transgenic lines) provide sufficient evidence that the cloned gene is correct.

Minor comments

- I suggest to tweak the sentence in the abstract as follows: 'Stable Pm57 transgenic wheat lines and introgression lines showed high levels of all-stage resistance against diverse isolates of the Bgt fungus, and no negative impact on agronomic parameters were observed in our experimental set-up.'
- *Ae. umbellulata* is sometimes misspelled as *A. umbellulate*.
- I am not sure if 'WTK6-vWAPm' is a good protein designation. This designation should reflect the protein domains and 'Pm' does not refer to a domain here. This protein designation clearly has a limitation in naming orthologs. I suggest that the authors either stick to their original designation (WTK7-vWA) or use WTK6b-vWA.

Reviewer #3:

Remarks to the Author:

The authors have made several enhancements to the Pm57 manuscript, incorporating additional analyses and results, such as PR gene expression, intraROS and cell death quantification, and evolutionary analysis. The newly included information contributes to a more comprehensive understanding of the role of Pm57 in conferring resistance against Bgt. However, we still believe that the manuscript requires further scientific and English editing before it can be accepted for publication.

We have some further comments and suggestions for editing the revised manuscript, as follows:

1. Line 55 - "Pm4 encodes a putative serine/threonine kinase" is wrong since it contains additional domains; we propose to define it as a "kinase fusion protein" (KFP).
2. Line 82 - Please describe the orthologous relationship between Lr9 and Pm57.
3. Lines 89 to 93 - This paragraph shows some overlap with the introduction. Thus, we propose to reorganize it and avoid duplication s.
4. Line 101 - The use of "Bgt response assays" is incorrect. We propose to change it to "the plant response to Bgt".
5. Line 134-135 - Please explain why you have chosen 300 M1 lines for screening mutants while you had 1,598 surviving M0 plants. The number "10-15" should not be at the beginning of the sentence. Please rephrase these sentences or provide the correct numbers for the mutant screening.
6. Line 155 - we propose to change "identification of G4 using EMS-induced mutants" to "distribution of independent mutations."
7. Line 165 - We propose to change "The G4 gene was 9,473 bp" to "The G4 gene length was 9,473 bp".
8. Line 202 - Please add some information to explain why they are positive but susceptible here.
9. Fig.3 - There is no need to show both T0 and T1 phenotypes in a Figure presented in the main text. Thus, we propose to change Fig.3 to Supplementary Fig. 3 and include more information.
10. Line 333 - (a) We propose changing to "Whole-plant, spike, and seed growth habits of...".

Reviewed by Tzion Fahima and Yinghui Li

Reviewer #4:

Remarks to the Author:

The authors have responded relevantly to most of the comments and requested improvements. I noticed just few minor comments:

L50 this is 17 or not but not « about 17 genes ».

For VIGS, there is still no detail about the protocol of the RT-qPCR, the method used to analyse the data, gene for normalization, the reference used? Are the third and fourth leaves mixed together in the same sample, this is not clear. In Supplementary Figure 2, 2 technical replicates. "technical" can be interpreted differently according to readers, please explain. How many leaves per replicate? Please indicate these details for all qPCR experiment, including the PR genes.

L264-265 I don't understand why you mentioned these different types of tubes as it is not shown on the figure.

Reviewer #5:

I have reviewed the rebuttal letter of Dr. Li et al. for their responses to questions and comments made by Reviewer 1 on the manuscript entitled "*Pm57* from *Aegilops searsii* encodes a tandem kinase protein conferring powdery mildew resistance in bread wheat" for Nature Communications. To better understand how Reviewer 1 raised the questions, I have also reviewed the script. Here is my report.

Reviewer 1 had three major concerns about the manuscript.

Reviewer's major concern #1. The authors identified **15** independent mutants that were loss-of-function for Pm57. **Five** mutants were initially sequenced using RNAseq and **three** were found to carry mutations in G4 (WTK7-vWA). Sequencing cDNA and gDNA of the **15** mutants found that **10** mutants carried polymorphisms that impacted the protein sequence of WTK7-vWA. The authors **correctly comment on** the potential of the additional **five** mutants to **uncover components** of Pm57 immune signaling. I was surprised to not see some form of **complementation tests** performed between mutants (including Pm57 and non-Pm57 mutants). Furthermore, **was Pm57 expressed in the five mutants? Was RNAseq performed for all?**

The authors' response: Partial mutants that uncover the causal TKPs have been reported previously, such as WTK3 (15/26) and WTK5 (7/14), where about half of the susceptible mutants represented second-site mutations. In addition, one out of 121 loss-of-function mutants did not carry a mutation in Lr9. We obtained 15 susceptible mutants in Pm57 lines, of which 4 are second-site mutations (**the previous statement of 5 is incorrect**), which support a partner protein may be involved in TKP signaling. We also calculated the probability that the 11 independent mutations were the result of chance alone based on Sanchez-Martin et al. (2016). The probability that the 11 independent mutations in Pm57 are the result of chance alone is 1×10^{-11} .

My opinion: I do not understand why the authors did not answer the questions directly but said that partial mutants that uncover the causal TKPs have been reported previously. What did it mean 'partial mutants'? Were some of the 15 mutants reported in previous papers already?

We have analyzed the expression levels of Pm57 in these four non-Pm57 mutants (see Lines 375-376). They showed a similar expression level with the resistant wild type parental line 89(5)69 (see Sup Fig. 15), indicating that these four mutants will be important materials for studying the components of the Pm57 immune signaling pathway.

My opinion: I do not agree with the claim 'these four mutants will be important materials for studying the components of the Pm57 immune signaling pathway'. How can this result 'indicate'

We did not perform MutRNA-seq on all mutants, but selected **5** mutants instead. Our selection is based on the preprint report of Yu et al. (2022); their analysis indicated that in the scenario of scrutinizing all genes in a discrete mapping interval, the minimum number of independent mutants required for identifying a candidate gene with a 2000 bp CDS at $p = 0.01$ is three. In addition, our selection is also supported by Wang et al. (2023) who recently showed that with 4-5

independent mutants, the probability of identifying a false positive transcript was very low (Supplementary Note 1, Wang et al., 2023). Sánchez-Martín, J. et al. Rapid gene isolation in barley and wheat by mutant chromosome sequencing. *Genome Biol* 17, 221 (2016)

My opinion: The authors explained why ‘five’ but not three mutant were selected for RNAseq, but this was not a question Reviewer 1 raised. Finally, I do not find any response to the question about ‘complementation tests’.

Reviewer’s major concern #2. The authors provide a limited analysis of 182 protein kinase domains. Similar to Li et al., I suggest the authors look at “Diversity, classification and function of the plant protein kinase superfamily” by Melissa D. Lehti-Shiu and Shin-Han Shiu (2012) for a protocol on classifying the protein kinase domain in Pm57. This analysis should guide the classification of individual protein kinase domains and **the relevant subfamilies to evaluate the evolutionary origin of the protein kinase domains in Pm57**. Phylogenetic trees must be bootstrapped and based on appropriate sampling to have an ability to infer evolutionary relationships.

The authors’ response: As suggested, we have classified the protein kinase domains in Pm57 based on the protocol described by Melissa D. Lehti-Shiu and Shin-Han Shiu (2012) and found that these two kinase domains in Pm57 belong to RLK/Pelle_DLSV subfamily. Therefore, Pm57 has a DLSV-DLSV configuration, similar to Rpg1, Pm24, and Sr62 (Yu et al., 2022). Notably, the Arabidopsis DLSV members was later recognized as LRR_8B, G-LPK, etc. subfamilies (Zulawski et al., 2014). This suggests that LRR_8B falls within the DLSV subfamily. To further classify the two protein kinase domains of Pm57, we used the 182 protein kinase domains discovered by Klymiuk et al. (2018) to construct a neighbor-joining (NJ) phylogenetic tree. As shown in Supplementary Fig. 11, the two protein kinase domains in Pm57 were assigned to LRR_8B subfamily (see Lines 293-302).

Bootstrap values have also been added in the phylogenetic tree. Lehti-Shiu, M.D. & Shiu, S.H. Diversity, classification and function of the plant protein kinase superfamily. *Philos Trans R Soc Lond B Biol Sci* 367, 2619-39 (2012).

My opinion: The authors reconstructed a neighbor-joining (NJ) phylogenetic tree and added Bootstrap values, as suggested by Reviewer 1, but they did not directly answer the question how Pm57 was originated during the evolution.

Reviewer’s major concern #3. Is Pm57 the ortholog of Lr9? This relationship was unclear and the phylogenetic analysis suggests they are highly related.

The authors’ response: Current orthology inference algorithms are generally based on sequence similarity, domain architecture, collinearity, and reciprocal best hit (Chen et al., 2022 and Zielezinski et al., 2017). Pm57 had a high similarity of 88.3% in amino acid sequences with Lr9 and they had the same kind of domains (Sup Fig. 4 and 5). The adjacent genomic region of Pm57 was syntenic to the Lr9 genomic regions in the *Ae. umbellulata* (Sup Fig. 10 and Sup Table 6). Reciprocal BLAST showed that Lr9 and Pm57 are each other’s best BLAST hits. (Sup Table 7). Therefore, analyses of sequence similarity, gene collinearity and “reciprocal best hits” inferred

Pm57 is the ortholog of Lr9. We have added these results to the results section of the text (see Lines 278-286).

My opinion: I have no comment on this one.

When reviewing the manuscript, I found substantial flaws that need to be fixed for publication anywhere. Here are examples.

Lines 26-27

There was some problem in the sentence ‘Further analyses revealed that Pm57 is an ortholog of Lr9 (WTK6-vWA) which confers wheat leaf rust resistance, thus designated WTK6-vWAPm’.

Lines 109-111

Of these 12 genes only genes G4 and G5 contain putative tandem kinase domains and were considered potential candidate genes for Pm57, while none of the remaining 10 genes was annotated to resemble any previously identified disease-resistant genes (Fig. 1c).

It was an incorrect statement that ‘only genes G4 and G5 contain putative tandem kinase domains’. The ‘genes’ do not have ‘kinase domains’.

Why should only those genes that encode kinase domains be considered the candidates for Pm57?

Lines 144-148

Alignment of the MutRNA-seq data to the 12 genes in Pm57 mapping interval revealed that gene G4 had EMS-type (G/C to A/T) mutations in all the five mutants, whereas only one EMS-type mutation was found in genes G2 (in Mut209) and G8 (in Mut141), and none in genes G5, G6, G7, G9, G10, G11 and G12. The expression levels of the remaining two genes (G1 and G3) were too low to reliably call mutations in transcriptome sequences (Fig. 2a and Supplementary Table 3). Therefore, gene G4 encoding a tandem kinase -vWA domains protein emerged as the most likely candidate of Pm57 among the 12 genes.

If the expression levels of the remaining two genes (G1 and G3) were too low to reliably call mutations in transcriptome sequences, how can the two genes be (therefore) excluded as the candidates?

Fig. 2 The figure has mixed with DNA mutants and mutated proteins in different lines. a). Five lines for 10/12 genes. b). 11 lines for G4. "c." for a coding DNA sequence, "p." for a protein sequence. How did six (IF2-7) produce premature stop codon (marked in red star)? Why exon 7 was missing before the stop codon? I have never seen such a confused figure.

Lines 167-175

Intriguingly, G4 had at least seven alternative splicing variants, designated IF1 - IF7, of which, IF1 was the main isoform, with a proportion of 50.6% (39 out of 77 tested G4 cDNA clones) at 0 h post-inoculation (hpi) with Bgt isolate E09 increasing to 80.9% at 24 hpi, and isoforms IF2 - IF7 were much less abundant (1.3% - 27.3%). IF1 encodes a full-length intact G4 protein with Kin I, Kin II and vWA domains, while IF2 and IF3 encode proteins with truncated Kin I and Kin II domains, and IF4-IF7 encode proteins with only truncated Kin I domain (Fig. 2b). Gene sequence comparison revealed that 11 of the 15 susceptible mutants had SNPs in G4 that resulted

in **amino acid substitutions**, premature stop codons, or relocation of the intron/exon splice sites (Fig. 2b). Specifically, a frameshift mutation was detected in Mut216 with a G/A point mutation in the splice acceptor site of intron 9. Mut351 was the same as Mut60 in G4, both of which had a nonsense mutation that gave rise to a premature stop codon at the amino acid position of 1,081. The other eight mutants (G78D in Mut223, G177E in Mut141, G193R in Mut51, D209N in Mut210, G424D in Mut92, P747L in Mut121, R829W in Mut22, and G903D in Mut209) harbored missense mutations that occurred in the kinase I (Kin I), kinase II (Kin II) or vWA domains (Fig. 2b). In addition, no sequence variations in gene G5 were found among all susceptible mutants.

What did it mean ‘the main isoform’? How can they claim ‘IF1 encodes a full-length intact G4 protein’? If they were amino acid substitutions only, how can they change the phenotypes? How can it be said that the ‘relocation’ of the intron/exon splice sites? How can they know the ‘missense mutations’ change protein functions?

REVIEWER COMMENTS

Reviewer #2 (Remarks to the Author):

I thank the authors for performing a thorough revision of their manuscript, which has greatly improved. It is a pity that they could not find a virulent powdery mildew isolate to be tested on the transgenic lines. Nevertheless, I think that the various experiments (11 independent EMS mutants, virus-induced gene silencing, transgenic lines) provide sufficient evidence that the cloned gene is correct.

We are grateful for the positive assessment of the manuscript and highlighting the strength of the presented results.

Minor comments

- I suggest to tweak the sentence in the abstract as follows: ‘Stable Pm57 transgenic wheat lines and introgression lines showed high levels of all-stage resistance against diverse isolates of the Bgt fungus, and no negative impact on agronomic parameters were observed in our experimental set-up.’

>Response: The sentence has been modified as suggested (see Lines 24-26).

- Ae. umbellulata is sometimes misspelled as A. umbellulate.

>Response: Thanks for pointing out this error. We have corrected this mistake throughout the manuscript.

I am not sure if ‘WTK6-vWAPm’ is a good protein designation. This designation should reflect the protein domains and ‘Pm’ does not refer to a domain here. This protein designation clearly has a limitation in naming orthologs. I suggest that the authors either stick to their original designation (WTK7-vWA) or use WTK6b-vWA.

>Response: Thanks for your suggestion. Considering that Lr9 (WTK6-vWA) and Pm57 are orthologs, we follow your suggestion and use WTK6b-vWA for Pm57 designation in the revised version.

Reviewer #3 (Remarks to the Author):

The authors have made several enhancements to the Pm57 manuscript, incorporating additional analyses and results, such as PR gene expression, intraROS and cell death quantification, and evolutionary analysis. The newly included information contributes to a more comprehensive understanding of the role of Pm57 in conferring resistance against Bgt. However, we still believe that the manuscript requires further scientific and English editing before it can be

accepted for publication.

Dear Tzion and Yinghui,

Thank you for a thorough review of the revised manuscript and recognizing that the revised version provides a comprehensive understanding of the role of *Pm57* in conferencing resistance to *Bgt*. We appreciate your constructive comments and suggestions. We have improved our manuscript through English Language Editing by SPRINGER NATURE. The edited version of our manuscript has been revised for scientific content and English writing with the goal of enhancing clarity for the readers.

We have some further comments and suggestions for editing the revised manuscript, as follows:

1. Line 55 - “*Pm4* encodes a putative serine/threonine kinase” is wrong since it contains additional domains; we propose to define it as a “kinase fusion protein” (KFP).

>Response: We agree with you and have changed “*Pm4* encodes a putative serine/threonine kinase” to “*Pm4* encodes a kinase fusion protein”

2. Line 82 - Please describe the orthologous relationship between *Lr9* and *Pm57*.

>Response: We have described the orthologous relationship between *Lr9* and *Pm57* in Lines 78-81. “Interestingly, we found that *Pm57* encodes a tandem kinase protein with putative kinase-pseudokinase domains followed by a von Willebrand factor A (vWA) domain, and further analyses revealed that *Pm57* is an ortholog of *Lr9* (WTK6-vWA), which mediates resistance to wheat leaf rust³⁵.”

3. Lines 89 to 93 – This paragraph shows some overlap with the introduction. Thus, we propose to reorganize it and avoid duplications.

>Response: As per your suggestion, we have reorganized the paragraph as “Previously, we mapped *Pm57* to the long arm of 2S⁵ in a 5.13 Mb genomic region using an F₂ population generated by crossing recombinant line 89(6)88 containing *Pm57* with TA3809, a CS *ph1b* deletion mutant that can promote homoeologous chromosome pairing and recombination³⁴.” (see Lines 88-90 in current version).

4. Line 101 - The use of “*Bgt* response assays” is incorrect. We propose to change it to “the plant response to *Bgt*”.

>Response: The suggestion has been incorporated (see Line 94).

5. Line 134-135 - Please explain why you have chosen 300 M1 lines for screening mutants while you had 1,598 surviving M0 plants. The number “10-15” should not be at the beginning of the sentence. Please rephrase these

sentences or provide the correct numbers for the mutant screening.

>Response: We randomly selected 300 M₁ lines for screening mutants from 1,598 M₁ lines and identified 15 independent susceptible mutants derived from 15 different M₁ lines. At that time, we believed 15 mutants should be enough for functional validation of *Pm57* in this study. Thus, we didn't use the remaining M₁ lines in this study. In terms of the number "10-15", we have rephrased this sentence to "At least ten plants from each of the randomly selected 300 M₁ families were screened for susceptible mutants using *Bgt* isolate E09." (see Lines 129-130).

6. Line 155 – we propose to change "identification of G4 using EMS-induced mutants" to "distribution of independent mutations."

>Response: We have updated the sentence (see Line 149). We have modified the content of Fig. 2b, and changed the title to "(b) EMS-derived loss-of-function mutants with mutations in the G4 gene sequence."

7. Line 165 – We propose to change "The G4 gene was 9,473 bp" to "The G4 gene length was 9,473 bp"

>Response: The sentence has been updated as suggested (see Line 160).

8. Line 202 – Please add some information to explain why they are positive but susceptible here.

>Response: We have made modifications to Fig.3. The names of the L20 and L29 are no longer highlighted in red, as we believe it is unnecessary to emphasize these two lines. Additionally, following your suggestion, Fig.3 has been replaced with Supplementary Fig.3. Therefore, in the latest revised manuscript, we have deleted the previous sentence "The susceptible positive T₀ transgenic plants are indicated in red." and added "All the positive T₀ transgenic plants (+) excluding L20 and L29 were highly resistant to the *Bgt* isolate E09," (see the legend of Supplementary Fig. 4). We also explained why lines L20 and L29 are susceptible though being positive in Line 207-208.

9. Fig.3 – There is no need to show both T0 and T1 phenotypes in a Figure presented in the main text. Thus, we propose to change Fig.3 to Supplementary Fig. 3 and include more information.

>Response: We agree with you and have moved Supplementary Fig. 3 to Fig.3. Also, Fig. 3a was moved to the Supplementary figure 4 and Fig. 3b was deleted.

10. Line 333 - (a) We propose changing to "Whole-plant, spike, and seed growth habits of..."

>Response: We have changed it (see Line 334).

Reviewer #4 (Remarks to the Author):

The authors have responded relevantly to most of the comments and requested improvements. I noticed just few minor comments:

Thank you for your suggestions and acknowledgment of the enhancements made in the revised manuscript in response to the comments.

L50 this is 17 or not but not « about 17 genes ».

>Response: Thanks for pointing out the error, and we have deleted 'about'.

For VIGS, there is still no detail about the protocol of the RT-qPCR, the method used to analyse the data, gene for normalization, the reference used? Are the third and fourth leaves mixed together in the same sample, this is not clear.

>Response: According to your suggestion, we have significantly improved the "Gene expression analysis" sections (see Lines 550-558) and provided all the relevant details about qRT-PCR experiment. "For the gene expression analyses of *G4* and *G5* in the VIGS experiments, the 3rd and 4th leaves were mixed together as one sample, and three technical replicates were performed for each sample. RNA extraction and cDNA synthesis were performed as described

above. Quantitative RT-PCR (qRT-PCR) analysis was carried out in a reaction volume of 20 µL using SYBR Mix (TaKaRa, Dalian, China) on a CFX96 real-time PCR detection system (Bio-Rad, Hercules, CA, USA). The conditions for

qRT-PCR were as follows: denaturation at 95 °C for 4 min, followed by 40 cycles of 94 °C for 20 s, 60 °C for 20 s and 72 °C for 20 s. Wheat *TaActin* was used as an internal reference gene²⁶. The transcript levels were calculated using the comparative CT method⁵⁸. The primer pairs 3F/2R and M-5F/5R were used to determine the expression of *G4* and *G5*, respectively (Supplementary

Table 1)."

In Supplementary Figure 2, 2 technical replicates. "technical" can be interpreted differently according to readers, please explain. How many leaves per replicate? Please indicate these details for all qPCR experiment, including the PR genes.

>Response: Thanks for your suggestion. We have added all the relevant details about all qRT-PCR experiments, including the interpretation of technical replicates (see Lines 559-570). At the same time, we also described the qRT-PCR experimental method in detail in the legend of Supplementary Figures. For example, in Supplementary Figure 2 (Supplementary Fig. 3 in the latest version), we wrote: "The 3rd and 4th leaves of each plant were mixed as one sample. qRT-PCR was performed three times for each sample, and each time as a technical replicate. Wheat *TaActin* was used as endogenous control, and

the relative expression levels were calculated using the comparative CT

n

L264-265 I don't understand why you mentioned these different types of tubes as it is not shown on the figure.

>Response: We thank you for the suggestion and have revised the previous legend of Fig. 5 (see Lines 261-264).

Reviewer #5 (Remarks to the Author):

I have reviewed the rebuttal letter of Dr. Li et al. for their responses to questions and comments made by Reviewer 1 on the manuscript entitled "*Pm57* from *Aegilops searsii* encodes a tandem kinase protein conferring powdery mildew resistance in bread wheat" for Nature Communications. To better understand how Reviewer 1 raised the questions, I have also reviewed the script. Here is my report.

>Response: We thank you for your comments and positive assessment of the rebuttal letter and the manuscript.

Reviewer 1 had three major concerns about the manuscript.

Reviewer's major concern #1. The authors identified [redacted] independent mutants that were loss-of-function for *Pm57*. [redacted] mutants were initially sequenced using RNAseq and [redacted] three were found to carry mutations in G4 (WTK7-vWA). Sequencing cDNA and gDNA of the [redacted] 15 mutants found that [redacted] mutants carried polymorphisms that impacted the protein sequence of WTK7-vWA. The authors correctly comment on the potential of the additional [redacted] mutants to uncover components of *Pm57* immune signaling. I was surprised to not see some form of **complementation tests** performed between mutants (including *Pm57* and non-*Pm57* mutants). Furthermore, **was *Pm57* expressed in the five mutants? Was RNAseq performed for all?**

The authors' response: Partial mutants that uncover the causal TKPs have been reported previously, such as WTK3 (15/26) and WTK5 (7/14), where about half of the susceptible mutants represented second-site mutations. In addition, one out of 121 loss-of-function mutant did not carry a mutation in *Lr9*. We obtained 15 susceptible mutants in *Pm57* lines, of which 4 are second-site mutations (the previous statement of 5 is incorrect), which support a partner protein may be involved in TKP signaling. We also calculated the probability that the 11 independent mutations were the result of chance alone based on Sanchez-Martin et al. (2016).

The probability that the 11 independent mutations in *Pm57* are the result of chance alone is 1×10^{-11}

My opinion: I do not understand why the authors did not answer the questions directly but said that partial mutants that uncover the causal TKPs have been reported previously. What did it mean "partial mutants"? Were some of the 15 mutants reported in previous papers already?

>Response: We apologize for not providing a satisfactory reply. At present, we have not performed complementation tests between mutants. We understand that 4 of 15 mutant having no mutations in *Pm57* gene sequence, however,

these four mutants do not invert our functional validation result and similar observations have also reported by other successfully cloned TKPs such as WTK3 and WTK5. In terms of WTK3, among 26 mutants identified, 15 mutants had mutations in *WTK3* gene. As for WTK5, among 14 mutants identified, 7 mutants had mutations in *WTK5* gene. Therefore, we proceeded with 11 mutants as the probability of having 11 independent mutations by chance was 1×10^{-11} and our results confirm *WTK6b-vWA* is *Pm57* and confers resistance to *Bgt*.

We are sorry about the confusion due to partial mutants. Term “partial mutants” in the REVIEWER COMMENTS meant that only a portion of mutants had mutations in the candidate gene, while some mutants do not have mutations in the candidate gene but exhibited susceptibility.

We have analyzed the expression levels of *Pm57* in these four non-*Pm57* mutants (see Lines 375-376). They showed a similar expression level with the resistant wild type parental line 89(5)69 (see Sup Fig. 15), indicating that these four mutants will be important materials for studying the components of the *Pm57* immune signaling pathway.

My opinion: I do not agree with the claim ‘these four mutants will be important materials for studying the components of the *Pm57* immune signaling pathway’. How can this result ‘indicate’ _____

>Response: In our revised manuscript, we analyzed the expression levels of *Pm57* in the four non-*Pm57* mutants (the mutants that are susceptible to *Bgt* but have intact *Pm57*) and found that their expression levels were indifferent with that of wild-type plants. These results further indicate that the susceptibility of these four non-*Pm57* mutants was not caused by the sequence mutations or expression level variation of *Pm57* gene. We believe the reasonable explanation is that mutations in other genes involved in *Pm57* immune signaling pathway may lead to the susceptibility of these four non-*Pm57* mutants. Therefore, we claimed that these four mutants will be important materials for studying the components of the *Pm57* immune signaling pathway. Anyway, we have followed the reviewer’s suggestion and modified the sentence to “Neither sequence variation nor changes in the expression level of *Pm57* were found in four of the 15 susceptible mutants (Supplementary Fig. 16), indicating that mutations may have occurred in other genes or elements involved in the *Pm57* regulatory pathway” in the current version.

We did not perform MutRNA-seq on all mutants, but selected 5 mutants instead. Our selection is based on the preprint report of Yu et al. (2022); their analysis indicated that in the scenario of scrutinizing all genes in a discrete mapping interval, the minimum number of independent mutants required for identifying a candidate gene with a 2000 bp CDS at $p = 0.01$ is three. In addition, our selection is also supported by Wang et al. (2023) who recently showed that with 4-5

independent mutants, the probability of identifying a false positive transcript was very low (Supplementary Note 1, Wang et al., 2023). Sanchez-Martin, J. et al. Rapid gene isolation in barley and wheat by mutant chromosome sequencing. *Genome Biol* 17, 221 (2016)

My opinion: The authors explained why 'five' but not three mutant were selected for RNAseq, but this was not a question Reviewer 1 raised. Finally, I do not find any response to the question about 'complementation tests'.

>Response: The RNAseq was not performed for all 15 susceptible mutants we identified in this study. Instead, five of 15 mutants were used to perform RNAseq. After discovering all five mutants (In the first version of the manuscript we claimed that three out of the five mutants used for RNAseq had mutations in candidate gene *G4*, but later when we reanalyzed the RNAseq data, we found that actually all the five mutants had mutations in *G4*) had mutations in candidate gene *G4*, the other remaining 10 mutants were used to amplify gDNA and cDNA sequences of *G4*, analyze mutations in *G4* sequence, and validate the function of candidate gene *G4*.

About the complementation tests, we are very sorry for not providing a direct reply. We didn't perform complementation tests between mutants in this study. We agree with the reviewers that the complementation tests between mutants are very helpful to uncover components of *Pm57* immune signaling. However, complementation was not essential in discovering that *WTK6b-vWA* is *Pm57* and it confers resistance to *Bgt* which was the primary goal of this study. Over functional validation using independent mutants, VIGS and finally transgenics are sufficient and all reviewers have agreed and supported our findings.

We appreciate your suggestion and we plan to perform complementation tests as a follow-up study to understand the *Pm57* regulatory pathway.

Reviewer's major concern #2. The authors provide a limited analysis of ████ protein kinase domains. Similar to Li et al., I suggest the authors look at "Diversity, classification and function of the plant protein kinase superfamily" by Melissa D. Lehti-Shiu and Shin-Han Shiu (2012) for a protocol on classifying the protein kinase domain in Pm57. This analysis should guide the classification of individual protein kinase domains and **the relevant subfamilies to evaluate the evolutionary origin of the protein kinase domains in Pm57**. Phylogenetic trees must be bootstrapped and based on appropriate sampling to have an ability to infer evolutionary relationships.

The authors' response: As suggested, we have classified the protein kinase domains in Pm57 based on the protocol described by Melissa D. Lehti-Shiu and Shin-Han Shiu (2012) and found that these two kinase domains in Pm57 belong to RLK/Pelle_DLSV subfamily. Therefore, Pm57 has a DLSV-DLSV configuration, similar to Rpg1, Pm24, and Sr62 (Yu et al., 2022). Notably, the Arabidopsis DLSV members was later recognized as LRR_8B, G-LPK, etc. subfamilies (Zulawski et al., 2014). This suggests that LRR_8B falls within the DLSV subfamily. To further classify the two protein kinase domains of Pm57, we used the 182 protein kinase domains discovered by Klymiuk et al. (2018) to construct a neighbor-joining (NJ) phylogenetic tree. As shown in Supplementary Fig. 11, the two protein kinase domains in Pm57 were assigned to LRR_8B subfamily (see Lines 293-302)

Bootstrap values have also been added in the phylogenetic tree. Lehti-Shiu, M.D. & Shiu, S.H. Diversity, classification and function of the plant protein kinase superfamily. *Philos Trans R Soc Lond B Biol Sci* 367, 2619-39 (2012).

My opinion: The authors reconstructed a neighbor-joining (NJ) phylogenetic tree and added Bootstrap values, as suggested by Reviewer 1, but they did not directly answer the question how Pm57 was originated during the evolution.

>Response: Thank you for your comments. We would like to make a further explanation for this concern. Through phylogenetic tree analysis, we found that the two kinase domains of Pm57 along with the two kinase domains of Rpg1, Pm24, and Sr62 belong to the LRR 8B subfamily, indicating that they probably have the same origin. In addition, based on phylogenetic tree analysis, we also observed that the two kinase domains of Pm57 are classified in the same group. Therefore, we speculate that Pm57 might have originated from the duplication of a kinase domain (see Lines 394-397). However, it was impossible to conclude how Pm57 originated during the evolution only by phylogenetic tree analysis. Studying the evolution of the Pm57 would be very interesting and needs further investigation, however, it was beyond the scope of the current study.

Reviewer's major concern #3. Is Pm57 the ortholog of Lr9? This relationship was unclear and the phylogenetic analysis suggests they are highly related.

The authors' response: Current orthology inference algorithms are generally based on sequence similarity, domain architecture, collinearity, and reciprocal best hit (Chen et al., 2022 and Zielezinski et al., 2017). Pm57 had a high similarity of 88.3% in amino acid sequences with Lr9 and they had the same kind of domains (Sup Fig. 4 and 5). The adjacent genomic region of Pm57 was syntenic to the Lr9 genomic regions in the *Ae. umbellulata* (Sup Fig. 10 and Sup Table 6). Reciprocal BLAST showed that Lr9 and Pm57 are each other's best BLAST hits. (Sup Table 7). Therefore, analyses of sequence similarity, gene collinearity and "reciprocal best hits" inferred

Pm57 is the ortholog of Lr9. We have added these results to the results section of the text (see Lines 278-286).

My opinion: I have no comment on this one.

>Response: Thank you, we have addressed this part in response to another reviewer in the previous revision.

When reviewing the manuscript, I found substantial flaws that need to be fixed for publication anywhere. Here are examples.

>Response: Thank you for your comments. We have carefully read through the article, made revisions as much as possible, and improved the language in our manuscript using a professional English Language Editing service by Springer Nature.

Lines 26-27

There was some problem in the sentence `Further analyses revealed that Pm57 is an ortholog of Lr9 (WTK6-vWA) which confers wheat leaf rust resistance, thus designated WTK6-vWAPm

>Response: We have modified the sentence. "Further analyses revealed that Pm57 is an ortholog of Lr9 (WTK6-vWA), and was thus designated WTK6b-vWA." (see Lines 26-27).

Lines 109-111

Of these 12 genes only genes G4 and G5 contain putative tandem kinase domains and were considered potential candidate genes for Pm57, while none of the remaining 10 genes was annotated to resemble any previously identified disease-resistant genes (Fig. 1c).

It was an incorrect statement that `only genes G4 and G5 contain putative tandem kinase domains'. The `genes' do not have `kinase domains'.

Why should only those genes that encode kinase domains be considered the candidates for Pm57?

>Response: Thank you for your comments. We have modified the sentence as follows: "Of these 12 genes, only G4 and G5 were annotated as disease resistance genes, both of which encode putative tandem kinase domain proteins, and were considered potential candidates for *Pm57* (Fig. 1c)."

Lines 144-148

Alignment of the MutRNA-seq data to the 12 genes in Pm57 mapping interval revealed that gene G4 had EMS-type (G/C to A/T) mutations in all the five mutants, whereas only one EMS-type mutation was found in genes G2 (in Mut209) and G8 (in Mut141), and none in genes G5, G6, G7, G9, G10, G11 and G12. The expression levels of the remaining two genes (G1 and G3) were too low to reliably call mutations in transcriptome sequences (Fig. 2a and Supplementary Table 3). Therefore, gene G4 encoding a tandem kinase -vWA domains protein emerged as the most likely candidate of Pm57 among the 12 genes.

If the expression levels of the remaining two genes (G1 and G3) were too low to reliably call mutations in transcriptome sequences, how can the two genes be (therefore) excluded as the candidates?

Fig. 2 The figure has mixed with DNA mutants and mutated proteins in different lines. a). Five lines for 10/12 genes. b). 11 lines for G4. "c." for a coding DNA sequence, "p." for a protein sequence. How did six (IF2-7) produce premature stop codon (marked in red star)? Why exon 7 was missing before the stop codon? I have never seen such a confused figure.

>Response: In this study, we conducted transcriptome analysis using 5 identified powdery mildew susceptible mutants and compared them with the resistant parent line. We found that G1 and G3 showed no expression in the examined resistant and susceptible materials, indicating that either these two annotated genes are pseudogenes, or not expressed in seedlings whether infected by *Bgt* isolate or not, neither supported G1 or G3 as *Pm57* candidates conferring resistance to powdery mildew. Therefore, they were excluded as candidate genes for *Pm57*. This practice is in accordance with the studies by Yu et al., 2022 and Wang et al., 2023; the authors also excluded non-expressed genes when cloning the *Sr62* and *Lr9/Lr58* genes using MutRNA-Seq approaches.

We apologize for the mistakes in Figure 2. To avoid confusion, we have retained only a full-length Pm57 protein (IF1) in Fig. 2, and displayed mutation positions and varied AA. The different alternative splicing forms of Pm57 were moved to Supplementary Fig. 2. The premature stop codons in IF2-7 (marked by a red star) are caused by alternative splicing. For example, in IF2, unlike in IF1, the 12th intron is not spliced out; this results in a different coding sequence of IF2 with IF1. This different coding sequence of IF2 caused a premature stop codon during translation at the position marked with a red star. In IF4-6, alternative splicing of the gene leads to the loss of exon 7 compared to IF1, resulting in different coding sequences of IF4-6 with IF1. These different coding sequences of IF4-6 caused premature stop codons during translation at the position marked with a red star.

Yu, G. et al. *Aegilops sharonensis* genome-assisted identification of stem rust resistance gene *Sr62*. *Nat. Commun.* 13, 1607 (2022).

Wang, Y. et al. An unusual tandem kinase fusion protein confers leaf rust resistance in wheat. *Nat. Genet.* 55, 914-920 (2023).

Lines 167-175

Intriguingly, G4 had at least seven alternative splicing variants, designated IF1 - IF7, of which, IF1 was the main isoform, with a proportion of 50.6% (39 out of 77 tested G4 cDNA clones) at 0 h post-inoculation (hpi) with Bgt isolate E09 increasing to 80.9% at 24 hpi, and isoforms IF2 - IF7 were much less abundant (1.3% - 27.3%). IF1 encodes a full-length intact G4 protein with Kin I, Kin II and vWA domains, while IF2 and IF3 encode proteins with truncated Kin I and Kin II domains, and IF4-IF7 encode proteins with only truncated Kin I domain (Fig. 2b). Gene sequence comparison revealed that 11 of the 15 susceptible mutants had SNPs in G4 that resulted

in **amino acid substitutions**, premature stop codons, or relocation of the intron/exon splice sites (Fig. 2b). Specifically, a frameshift mutation was detected in Mut216 with a G/A point mutation in the splice acceptor site of intron 9. Mut351 was the same as Mut60 in G4, both of which had a nonsense mutation that gave rise to a premature stop codon at the amino acid position of 1,081. The other eight mutants (G78D in Mut223, G177E in Mut141, G193R in Mut51, D209N in Mut210, G424D in Mut92, P747L in Mut121, R829W in Mut22, and G903D in Mut209) harbored missense mutations that occurred in the kinase I (Kin I), kinase II (Kin II) or vWA domains (Fig. 2b). In addition, no sequence variations in gene G5 were found among all susceptible mutants.

What did it mean 'the main isoform'? How can they claim 'IF1 encodes a full-length intact G4 protein'? If they were amino acid substitutions only, how can they change the phenotypes? How can it be said that the 'relocation' of the intron/exon splice sites? How can they know the 'missense mutations' change protein functions?

>Response: The main isoform refers to the most abundant form among the various alternative splicing forms produced after the transcription of a gene. This term has been used in previous publications, such as Lu et al., 2020.

In this study, we found that the G4 gene encoded at least 7 alternative splicing forms. IF1 is the most abundant and longest form that translated a protein containing two protein kinase domains (Kin I and Kin II) and a vWA domain (Fig. 2 and Supplementary Fig. 2). The remaining forms either translated a protein composed of only the Kin I and Kin II protein kinase domains (IF2 and IF3), or encoded a protein containing a Kin I domain (IF4 to IF7) (Supplementary Fig. 2). Therefore, we conclude that IF1 encodes a full-length intact G4 protein.

At present, it is not clear how only amino acid substitutions can change the phenotypes. In this study, we induced mutations in the disease-resistant materials using EMS mutagenesis, screened for disease-susceptible mutants, and found that eight of 11 susceptible mutants had missense mutations caused by amino acid substitutions in *Pm57* genes. Besides, amino acid substitutions in a gene can change its phenotype is a common phenomenon, as demonstrated by the cloning of other disease resistance genes such as *Pm24*, *Sr62*, *Lr9*, etc. In short, amino acid substitutions in *Pm57* and many other disease-resistant proteins can change their phenotype of resistance to susceptibility, indicating that the missense mutations in a protein could change its function and lead to a phenotype change.

The term of 'relocation' of the intron/exon splice site was based on the article by Lu et al., 2020. In our study, intron 9 of the *Pm57* gene has the most common

GT-AG type intron/exon splice site, but in the disease-susceptible mutant Mut216, the splice acceptor site of intron 9 mutated from AG to AA. This change leads to alterations in intron/exon splice sites, ultimately resulting in a frameshift mutation.

Lu, P. et al. A rare gain of function mutation in a wheat tandem kinase confers resistance to powdery mildew. *Nat. Commun.* 11, 680 (2020).

Reviewers' Comments:

Reviewer #3:

Remarks to the Author:

The revised manuscript addressed properly our comments and suggestions and we recommend it for publication.

Tzion Fahima and Yinghui Li

Reviewer #5:

Remarks to the Author:

I found that the manuscript has been significantly improved in reading, as it has undergone comprehensive editing by a language expert in English, as the authors claimed.

The authors answered most of the questions I asked but still did not answer some questions directly. For instance, the authors did not answer the question: 'Why should only those genes that encode kinase domains be considered the candidates for Pm57?'

I am still concerned about the authors' conclusive claim, '..... four of the 15 susceptible mutants (Supplementary Fig. 16), indicating that mutations may have occurred in other genes or elements involved in the Pm57 regulatory pathway". These four plants showed susceptibility, which was an unexpected phenotype. The 4 out of 15 ratio was a high proportion in cloning a major resistance gene. Without any experimental evidence for the presence of 'other genes' under the studied genetic background, the possibility that the cloned gene was not the causal gene could not be excluded. The authors should experimentally address the four exceptional susceptible mutants.

REVIEWER COMMENTS

Reviewer #3 (Remarks to the Author):

The revised manuscript addressed properly our comments and suggestions and we recommend it for publication.

Tzion Fahima and Yinghui Li

Dear Tzion and Yinghui,

We are thankful to you for providing a thorough review of our manuscript and valuable comments that helped to improve the quality and rigor of our manuscript. We are glad to see three reviewers have acknowledged the improvements in the latest version of the manuscript and supported our findings.

Reviewer #5 (Remarks to the Author):

I found that the manuscript has been significantly improved in reading, as it has undergone comprehensive editing by a language expert in English, as the authors claimed.

We thank you for your comments and positive assessment of the manuscript.

The authors answered most of the questions I asked but still did not answer some questions directly. For instance, the authors did not answer the question: 'Why should only those genes that encode kinase domains be considered the candidates for *Pm57*?'.

>Response: We understand the reviewer's question and would like to clarify our response. We didn't consider only those genes encoding kinase domains as the candidates of *Pm57*, instead, we chose all resistance (R) genes in *Pm57* harboring interval as putative *Pm57* candidates.

As described in our manuscript, we mapped *Pm57* to a 710 kb interval, which harbored 12 annotated genes based on the genome sequence of *Ae. searsii* (TE01). Of those 12 genes, only two (*G4* and *G5*) were annotated as R genes, and both *G4* and *G5* coincidentally encode kinase domains. Further, no amino acid sequence variation and expression level differences were found for the other 10 genes among resistant wild type and five susceptible mutants. Thus, we selected the two disease resistance (R) genes to pursue as potential candidates.

To avoid confusion, we have modified the sentence as follows: "Of these 12 genes, only *G4* and *G5* were annotated as disease resistance genes, which

coincidentally encode putative tandem kinase domain proteins." (see Lines 104-105)

I am still concerned about the authors' conclusive claim, '..... four of the 15 susceptible mutants (Supplementary Fig. 16), indicating that mutations may have occurred in other genes or elements involved in the Pm57 regulatory pathway". These four plants showed susceptibility, which was an unexpected phenotype. The 4 out of 15 ratio was a high proportion in cloning a major resistance gene. Without any experimental evidence for the presence of 'other genes' under the studied genetic background, the possibility that the cloned gene was not the causal gene could not be excluded. The authors should experimentally address the four exceptional susceptible mutants.

>Response: We agree with the reviewer's statement that the 4 out of 15 ratio was a high proportion in cloning a major resistance gene, in comparison to cloning of genes such as *Pm1a*¹, *Pm3b*², *Pm55a*³, *Pm55b*³, or *Yr15(WTK1)*⁴. In these studies, a total of 6, 13, 3, 10, and 10 susceptible mutant plants were used to validate the gene function, and all susceptible mutants had mutation sites in encoding sequences of these genes. However, it is also important to note that not all the R gene cloning studies were able to find mutations in all of the susceptible mutants such as the recent cloning of *Pm5e*⁵, *Pm24(WTK3)*⁶, *Sr62*⁷, *Lr13*⁸, etc.

As the reviewer mentioned, "4 out of 15 ratio was a high proportion in cloning a major resistance gene". However, the ratio translates to 26.7% (4 out of 15) for *Pm57* cloning in this study, which is very reasonable when compared with the successful cloning of other major resistance genes, such as *Pm5e*⁵, *Pm24(WTK3)*⁶, *Sr62*⁷, or *Lr13*⁸. For instance, the proportions of mutant plants having no mutation sites in encoding sequences of the cloned genes were 68.9% (20/29), 42.3% (11/26), 50.0% (7/14), and 25.0% (2/8) in case of *Pm5e*⁵, *Pm24(WTK3)*⁶, *Sr62*⁷, and *Lr13*⁸, respectively while performing functional validation using mutagenesis.

Secondly, the reviewer mentioned that "without any experimental evidence for the presence of 'other genes' under the studied genetic background, the possibility that the cloned gene was not the causal gene could not be excluded".

Please note that our claim that *G4* is *Pm57* is not merely based on functional validation using mutagenesis as it is being concluded. Our claim that *G4* is the causal gene of *Pm57* is based on a thorough investigation by four independent approaches, including fine mapping, susceptible mutant analysis, VIGS-induced gene silencing, and transgenic assays. Firstly, we fine-mapped *Pm57* into a 710 kb interval that harbored only two R genes (*G4* and *G5*), thus we

considered *G4* and *G5* as *Pm57* candidates. Secondly, we generated 15 susceptible mutant plants and found that 11 of 15 susceptible plants had mutations in *G4* gene encoding sequence, in contrast, no mutation existed in *G5* sequences. Thirdly, we performed VIGS-induced silencing of both *G4* and *G5*, and found silencing of *G4* led to susceptibility of resistant plants containing *Pm57*, whereas *G5* silencing couldn't cause resistance change of *Pm57*-containing plants. Based on these results, we concluded that gene *G4* was most likely the candidate for *Pm57*. Finally, we transferred *G4* gene into a susceptible common wheat and found 31 transgenic plants that contained and expressed *G4* gene conferred resistance with the same level of *Pm57*-carrying resistant line. By combining these results of fine mapping, susceptible mutant analysis, VIGS-induced gene silencing, and transgenic assays, we finally concluded that *G4* gene was the causal gene for *Pm57*.

Thus, analysis of the candidate gene sequence mutations using susceptible mutants is only one of the four approaches we used to clone *Pm57*, and, that 4 out of 15 susceptible plants had no mutation sites in *G4* (*Pm57*) encoding sequence cannot challenge the conclusion that *G4* gene, which encodes a resistant protein with kinase-pseudokinase domains, a vWA domain, and a putative Vwaint domain, is the causal gene for *Pm57*.

We understand the reviewer's concern about our conclusive claim, '..... four of the 15 susceptible mutants (Supplementary Fig. 16), indicating that mutations may have occurred in other genes or elements involved in the *Pm57* regulatory pathway". As mentioned above, a reasonable proportion of exceptional susceptible mutants has also been reported in cloning of other major resistance genes. Please note those authors have also suggested possible mutations in other unknown genes or elements involved in the resistance gene regulation pathway⁵⁻⁸. Nevertheless, we agree with the reviewer that this claim needs experimental evidence, thus we have revised the sentence as "Neither sequence variation nor changes in the expression level of *Pm57* were found in four of the 15 susceptible mutants (Supplementary Fig. 16), providing a resource to further study other genes or elements required for *Pm57* function." (see Lines 377-379)

We believe with the independent line of evidence, our results conclusively demonstrate that *G4* is *Pm57* and we respectfully disagree with the reviewer that further experiments are needed to validate the candidacy of *G4* as *Pm57*. However, based on reviewer's suggestion, we plan to conduct complementation tests as a follow-up study to elucidate the *Pm57* regulatory pathway where we will study the four exceptional susceptible mutants.

The following are parts of the referred publications.

1. Hewitt, T. et al. A highly differentiated region of wheat chromosome 7AL encodes a *Pm1a* immune receptor that recognizes its corresponding *AvrPm1a* effector from *Blumeria graminis*. *New Phytol.* **229**, 2812-2826 (2021).
2. Yahiaoui, N., Srichumpa, P., Dudler, R. & Keller, B. Genome analysis at different ploidy levels allows cloning of the powdery mildew resistance gene *Pm3b* from hexaploid wheat. *Plant J.* **37**, 528-38 (2004).
3. Lu, C. et al. Wheat *Pm55* alleles exhibit distinct interactions with an inhibitor to cause different powdery mildew resistance. *Nat. Commun.* **15**, 503 (2024).
4. Klymiuk, V. et al. Cloning of the wheat *Yr15* resistance gene sheds light on the plant tandem kinase-pseudokinase family. *Nat. Commun.* **9**, 3735 (2018).
5. Xie, J. et al. A rare single nucleotide variant in *Pm5e* confers powdery mildew resistance in common wheat. *New Phytol.* **228**, 1011-1026 (2020).
“..... No sequence variation was detected in the entire 13.5-kb genomic regions containing RXL, NLR and C2 at another 20 susceptible EMS mutants, suggesting possible mutations in other unknown genes or elements involved in the Pm5e regulation pathway.”
6. Lu, P. et al. A rare gain of function mutation in a wheat tandem kinase confers resistance to powdery mildew. *Nat. Commun.* **11**, 680 (2020).
“..... No sequence variation was detected in the WTK3 and CNL genes (including the putative promoters, exons and introns, and terminator regions) of the other 15 EMS mutants, suggesting possible mutations in other unknown genes or elements involved in the MIHLT regulation pathway.”
7. Yu, G. et al. *Aegilops sharonensis* genome-assisted identification of stem rust resistance gene *Sr62*. *Nat. Commun.* **13**, 1607 (2022).
“..... only seven of 14 susceptible mutants carried non-synonymous or missense mutations in the tandem kinase (*Sr62*). This suggests that we obtained second-site mutations in one or multiple genes required for *Sr62* function.”
8. Yan, X. et al. High-temperature wheat leaf rust resistance gene *Lr13* exhibits pleiotropic effects on hybrid necrosis. *Mol. Plant.* **14**, 1029-1032 (2021).
“..... The other two susceptible mutants (EM7 and EM8) had no SNP in the NLR compared with NLRZM, suggesting that mutations might have occurred in other regulators in the LrZH22-mediated defense pathway.”

Reviewers' Comments:

Reviewer #3:

Remarks to the Author:

The authors used four complementary and independent approaches (positional cloning, mutagenesis, virus-induced gene silencing, and transgenic gain-of-function) that prove that G4 is indeed Pm57. Thus, we agree with the conclusions of the authors and recommend the paper for publication.

Reviewer #5:

Remarks to the Author:

I have carefully reviewed the new revision and the rebuttal letter the authors submitted for their manuscript of Pm57.

Question 1. My point was that all 12 genes mapped in the 710 kb interval should be the candidate genes for Pm57. However, the authors insisted 'only G4 and G5 were annotated as disease resistance genes, which coincidentally encode putative tandem kinase domain proteins'. I do not understand why G4 and G5 encoding kinase proteins should be 'disease resistance genes.' In supplementary Table 2, the authors presented either G4 or G5 was not annotated as 'disease resistance genes.'

Questions 2. I suggested that the authors should experimentally address the four exceptional susceptible mutants. However, the authors did not do additional experiments but disagreed with the suggestion by providing several research articles that used a similar method to validate candidate genes.

When I reviewed the revised manuscript, I found some more significant issues (my opinion only). For example,

1. The transcript levels of G4 in the leaves of 16 T1 transgenic lines, presented in Fig. 3b, did not fit with the genotypes of the transgenic lines, presented in supplemental Table 6. Many T1 transgenic populations showed that all plants were positive and no negative plants (e.g., L2, L6, L15, L16, L19, L22...), but some T1 transgenic populations showed the segregation of positive and negative plants (e.g., 8:4 in L4 and L12, 9:3 in L3 and L18). The authors did not tell how positive and negative plants were segregated in the T1 transgenic populations, how many copies of the transgene were found in each T1 population, how the transgene copy number was associated with the transcript levels (Fig. 3b), or if no difference in the phenotypes was found between homozygous and heterozygous genes in transgenic plants. There were a lot of issues in science on this part.

2. The authors claimed that the Pm57-GFP fusion protein was fluorescent in both the nucleus and the cytosol (Fig. 4a) compared to the 35S:GFP control in wheat protoplasts. I suggest the authors check where the GFP signals were localized in the wheat protoplasts in published papers. Even if there were no problems in Fig. 4a, I do not think that the authors' claim that Pm57-GFP fusion protein was localized on the nucleus and the 'cytosol' was correct.

REVIEWER COMMENTS

Reviewer #5 (Remarks to the Author):

I have carefully reviewed the new revision and the rebuttal letter the authors submitted for their manuscript of Pm57.

Question 1. My point was that all 12 genes mapped in the 710 kb interval should be the candidate genes for Pm57. However, the authors insisted 'only G4 and G5 were annotated as disease resistance genes, which coincidentally encode putative tandem kinase domain proteins'. I do not understand why G4 and G5 encoding kinase proteins should be 'disease resistance genes.' In supplementary Table 2, the authors presented either G4 or G5 was not annotated as 'disease resistance genes.'

>Response: Thank you very much for your comments. We are very sorry that we have not corrected it clearly. The proteins encoded by disease resistance genes (R-genes) share common domains such as coiled-coil (CC), nucleotide binding region (NB), Toll-interleukin region (TIR), leucine rich region (LRR) and kinase domain (K). Therefore, the genes encoding proteins with one of these domains are generally considered putative disease resistance genes. In our study, both G4 and G5 proteins contain tandem kinase domains, thus genes G4 and G5 could be considered as putative disease resistance genes. To avoid confusion, we have modified the sentence as follows: "It is worth noting that G4 and G5 encode proteins containing putative tandem kinase domains and represent wheat tandem kinase (WTK) genes (Fig. 1c), which belong to a new family of disease-resistant genes discovered in recent years^{39,40}."

Questions 2. I suggested that the authors should experimentally address the four exceptional susceptible mutants. However, the authors did not do additional experiments but disagreed with the suggestion by providing several research articles that used a similar method to validate candidate genes.

>Response: Thank you for your valuable suggestions. We are sorry that we have not provided a satisfactory response to this question in the last reply. We mistakenly thought that you were concerned about the correctness of the gene we cloned. Therefore, we reiterate that we have verified G4, which conferred strong broad-spectrum powdery mildew resistance to wheat in transgenic wheat, is indeed Pm57 gene. In addition, we provided several articles to show that the method we used to clone Pm57 gene is commonly used in recent years for cloning important disease-resistant genes.

In fact, we totally agree with your suggestion of experimentally addressing the four exceptional susceptible mutants. We appreciate your valuable suggestion that will help us a lot for further understanding mechanism of Pm57-mediated resistance to powdery mildew. Following your suggestion, we have conducted experiments aiming to address the four exceptional susceptible mutants and

made some progresses. We amplified the cDNA sequences of all the other expressed genes in addition to *G4* and *G5* in the *Pm57*-harboring interval from the four exceptional susceptible mutants. However, none sequence variations was identified in those tested genes between the resistant wild-type parental line 89(5)69 and these four mutants (the primers used were listed below). This further supports that the mutated genes leading to susceptibility of those four exceptional susceptible mutants are not in *Pm57*-harboring region. In addition, we also amplified cDNA sequences of genes encoding *Pm57*-interacting proteins identified in the Yeast two-hybrid library of wheat leaves infected by the *Bgt* pathogen and still did not identify any mutation in the four exceptional susceptible mutants.

Next, we will cross those mutants with *Pm57* wild-type parental lines, develop different populations segregating for powdery mildew resistance, and then clone the genes based on classic positional cloning. Besides, we are continuing to identify proteins interacting with *Pm57* so as to address the mutants by comparing gene sequences of *Pm57* interacting proteins between wild type and those four exceptional susceptible mutants. However, cloning unknown genes in wheat currently are still very challenging, it may take several years to completely address the four exceptional susceptible mutants.

Table 1 List of primers and amplicon size of the genes in *Pm57*-harbored interval

Gene	Primer set sequences (5' to 3')	Tm	Amplicon size
G2	F: GGCGTGATCTGGTTTTGAT R: CAACGCTCCATGATCATAGC	57°C	620 bp
G6	F: AGGAGTTCCACGAGTCGCT R: CAACTCCCCTCTCTCATGCT	57°C	2432 bp
G7	F: CAAGCGCGCAAGTACAACAT R: TCATGGCCCCGAAATAGAGGT	57°C	1314 bp
G8	F: TTCGCCGTGGTATTTATGTG R: CTCAGGGTTCTTGGCTGTTC	57°C	2690 bp
G9	F: GCACACTATGGCTCCAGCCT R: ACGACGCCGATCATTAGTTG	56°C	668 bp
G10	F: GCACACTATGGCTCCAGCTTC R: GACTGACATGATGACGCTGG	57°C	669 bp
G11	F: CTGCTCGCACGATAAAGGAT R: TGACACGCTGACATGACGAC	56°C	682 bp
G12	F: GTTCAGGATGGCTCCAGCT R: CGCTGACTGACATGATGACG	57°C	697 bp

When I reviewed the revised manuscript, I found some more significant issues (my opinion only). For example,

1. The transcript levels of *G4* in the leaves of 16 T1 transgenic lines, presented in Fig. 3b, did not fit with the genotypes of the transgenic lines, presented in supplemental Table 6. Many T1 transgenic populations showed that all plants were positive and no negative plants (e.g., L2, L6, L15, L16, L19, L22...), but some T1 transgenic populations showed the segregation of positive and negative plants (e.g., 8:4 in L4 and L12, 9:3 in L3 and L18). The authors did not tell how positive and negative plants were segregated in the T1 transgenic populations, how many copies of the transgene were found in each T1

population, how the transgene copy number was associated with the transcript levels (Fig. 3b), or if no difference in the phenotypes was found between homozygous and heterozygous genes in transgenic plants. There were a lot of issues in science on this part.

>Response: Thank you for your suggestion to improve the section. At that time, we used about 12 seedlings for each of the T₁ transgenic lines to explore the relationship between disease resistance and the presence of *G4* gene in transgenic plants. We confirmed that all *G4*-positive plants were resistant, and all negative plants were susceptible except for transgenic lines L20 and L29, in which all positive and negative plants were susceptible. We further tested *G4* gene expression and found that *G4* was not expressed in transgenic lines L20 and L29. Whereas, only small samples of about 12 plants for each line were used, thus we didn't look into the copy numbers of transgene, and also did not investigate the influence of transgene copy number on transcript levels and disease resistance of *G4*.

Response to your comments “how positive and negative plants were segregated in the T₁ transgenic populations, how many copies of the transgene were found in each T₁ population”: We now estimated the *G4* gene copy number in each T₁ line based on resistance segregation ratios using the remaining seeds of the 16 T₁ lines. At the same time, we also determined the presence of *G4* genes in those tested plants. The results have been listed in the table below where red numbers represent the numbers of newly tested transgenic seedlings. The results confirmed that line L5 was *G4* transgene negative, and all tested plants of L5 were susceptible. Among the remaining 15 *G4* positive T₁ lines, all T₁ plants of lines L20 and L29 were susceptible, though the ratios of *G4* gene positive plants to negative plants fit 3:1. The results indicated that both L20 and L29 contained one copy of unexpressed *G4* transgene, which is in accordance with previous results that no expression of *G4* transgenes was detected in these two T₁ lines (Fig. 3b). In L16, all 55 T₁ plants are resistant and *G4* positive, suggesting that L16 might have more than 3 copies of *G4* transgene which should segregate 1/64 susceptible plants. In the remaining 12 lines, six lines (L4, L7, L8, L9, L18, L30) had single copy of expressed *G4* transgene, and the other six lines (L1, L2, L11, L15, L17, L22) had two copies of expressed *G4* transgene.

Table 2 The disease resistance and copy number of *G4* transgene determined in 16 transgenic T₁ lines

T ₁ transgenic family	No. of resistant plants	No. of susceptible plants	Chi-squared test χ^2 value	No. of putative transgene copy
L1	46(11+35)	3(1+2)	$\chi^2_{15:1}=0.001$	2
L2	51(12+39)	3(0+3)	$\chi^2_{15:1}=0.046$	2
L4	31(8+23)	12(4+8)	$\chi^2_{3:1}=0.194$	1
L5	0(0+0)	32(12+20)	-	-

L7	30(8+22)	11(4+7)	$\chi^2_{3:1}=0.073$	1
L8	45(9+36)	18(3+15)	$\chi^2_{3:1}=0.429$	1
L9	51(11+40)	9(0+9)	$\chi^2_{3:1}=3.200$	1
L11	53(10+43)	4(1+3)	$\chi^2_{15:1}=0.058$	2
L15	59(11+48)	3(0+3)	$\chi^2_{15:1}=0.213$	2
L16	55(12+43)	0(0+0)	$\chi^2_{15:1}=3.670$	≥ 3
L17	41(9+31)	4(2+2)	$\chi^2_{15:1}=0.538$	2
L18	35(9+26)	12(3+9)	$\chi^2_{3:1}=0.007$	1
L20	0(0+0)	65(11+54)	-	-
L22	50(12+38)	3(0+3)	$\chi^2_{15:1}=0.031$	2
L29	0(0+0)	37(12+25)	-	-
L30	44(10+34)	10(2+8)	$\chi^2_{3:1}=1.210$	1

Note: When $p=0.05$, $\chi^2=3.841$. The χ^2 obtained from the experiment are all less than 3.841. Statistically, the difference is not significant at the level of 0.05, and the difference between the observed number and the theoretical number belongs to random error.

Fig. 3b:

Response to your comments “how the transgene copy number was associated with the transcript levels (Fig. 3b)”: Combining the estimated copy number listed in Table 2 with the expression data in Fig. 3b, we found that the transgene copy number was generally positively correlated to the transcript levels of transgene in the examined transgenic lines except for line L22. Line L22 was estimated to contain two copies of the *G4* transgene, but the expression level of *G4* transgene was lower than that of line L7 which had a single copy of *G4* gene. It is reasonable to consider that the expression level of a transgene is affected not only by its copy number, but also by its inserted position in receptor chromosomes.

Response to your comments that “..... difference in the phenotypes was found between homozygous and heterozygous genes in transgenic plants”: We have assessed the resistance phenotype of each *G4* positive plants in these transgenic lines segregating for *G4* gene, and no resistance level differences were observed among positive plants, all of them exhibiting an immune response to *Bgt* E09. Therefore, it appears that no resistance level differences existed between homozygous and heterozygous transgenic plants.

2. The authors claimed that the Pm57-GFP fusion protein was fluorescent in both the nucleus and the cytosol (Fig. 4a) compared to the 35S:GFP control in wheat protoplasts. I suggest the authors check where the GFP signals were localized in the wheat protoplasts in published papers. Even if there were no problems in Fig. 4a, I do not think that the authors' claim that Pm57-GFP fusion protein was localized on the nucleus and the 'cytosol' was correct.

>Response: We sincerely thank you for pointing out this mistake as it had been overlooked throughout the revision process. As suggested, we have checked the locations of GFP signals in the wheat protoplasts in published papers and found that GFP signals were localized in the nucleus and 'cytoplasm' in wheat protoplasts (Zhu et al., 2015; Li et al., 2020). Thus, we have corrected the previous description and modified the sentence as follows: "As shown in Fig. 4a, green fluorescence of the GFP control was observed in the nucleus and cytoplasm. Similarly, the Pm57-GFP fusion protein was fluorescent in the nucleus and cytoplasm in wheat (Fig. 4a)."

Zhu, X. et al. The wheat AGC kinase TaAGC1 is a positive contributor to host resistance to the necrotrophic pathogen *Rhizoctonia cerealis*. *J. Exp. Bot.* 66(21): 6591-6603 (2015).

"the green fluorescent signal of the 35S:TaAGC1-GFP protein in wheat mesophyll protoplasts was detected in the cytoplasm and nucleus (Fig. 4C). The green fluorescent signal of the 35S:GFP protein was distributed in the nucleus and cytoplasm (Fig. 4A-C). Thus, the TaAGC1 protein localizes in the cytoplasm and nucleus in wheat."

Li, X. et al. A wheat WRKY transcription factor TaWRKY46 enhances tolerance to osmotic stress in transgenic *Arabidopsis* plants. *Int. J. Mol. Sci.* 21(4):1321 (2020).

"The GFP signal of 35S:TaWRKY46-GFP was exclusively observed in the nucleus, whereas GFP signal of the control was discovered in the cytoplasm and nucleus (Figure 3)."

Reviewers' Comments:

Reviewer #3:

Remarks to the Author:

The answers of the authors to the reviewers comments are satisfactory and we recommend this manuscript for publication.

REVIEWERS' COMMENTS

Reviewer #3 (Remarks to the Author):

The answers of the authors to the reviewers comments are satisfactory and we recommend this manuscript for publication.

>Response: We thank the Reviewer for the encouraging comments.